# Geometry Aware Operator Transformer As An Efficient And Accurate Neural Surrogate For PDEs On Arbitrary Domains

**Shizheng Wen**[1]         **Arsh Kumbhat**[1]         **Levi Lingsch**[1,2]

**Sepehr Mousavi**[1,3]   **Yizhou Zhao**[4]   **Praveen Chandrashekar**[5]   **Siddhartha Mishra**[1,2]

[1] Seminar for Applied Mathematics, ETH Zurich, Switzerland
[2] ETH AI Center, Zurich, Switzerland
[3] Department of Mechanical and Process Engineering, ETH Zurich, Switzerland
[4] School of Computer Science, CMU, USA
[5] Centre for Applicable Mathematics, TIFR, India

## Abstract

The very challenging task of learning solution operators of PDEs on arbitrary domains accurately and efficiently is of vital importance to engineering and industrial simulations. Despite the existence of many operator learning algorithms to approximate such PDEs, we find that accurate models are not necessarily computationally efficient and vice versa. We address this issue by proposing a geometry aware operator transformer (GAOT) for learning PDEs on arbitrary domains. GAOT combines novel multiscale attentional graph neural operator encoders and decoders, together with geometry embeddings and (vision) transformer processors to accurately map information about the domain and the inputs into a robust approximation of the PDE solution. Multiple innovations in the implementation of GAOT also ensure computational efficiency and scalability. We demonstrate this significant gain in both accuracy and efficiency of GAOT over several baselines on a large number of learning tasks from a diverse set of PDEs, including achieving state of the art performance on three large scale three-dimensional industrial CFD datasets. Our project page for accessing the source code is available at camlab-ethz.github.io/GAOT.

## 1   Introduction

Partial Differential Equations (PDEs) are widely used to mathematically model very diverse natural and engineering systems [17]. Currently, numerical algorithms, such as the finite element and finite difference methods, are the preferred framework for *simulating* PDEs [41]. However, these methods can be computationally very expensive, particularly for the so-called *many-query* problems such as uncertainty quantification (UQ), control, and inverse problems. Here, the solver must be called repeatedly, leading to prohibitive costs and providing the impetus for the design of fast and efficient surrogates for PDE solvers [34].

To this end, ML/AI based algorithms are increasingly being explored as *neural PDE surrogates*. In particular, *neural operators* [24, 6], including those proposed in [27, 28, 32, 42, 22], which learn the *PDE solution operator* from data, are widely used [4]. As many of these neural operators are restricted to PDEs on Cartesian (regular) grids, they cannot be directly applied to most engineering

39th Conference on Neural Information Processing Systems (NeurIPS 2025).

and industrial systems, which are set on domains with complex geometries, imposing a pressing need for neural operators for learning PDEs on arbitrary domains (point clouds).

In this context, a variety of options have recently been proposed, including domain masking for FNO and CNO [42], replacing FFT in FNO with direct spectral evaluations [30], augmenting FNO with learned diffeomorphisms [26] and mapping arbitrary point cloud data between the input domain and latent regular grids with learned encoders/decoders, while processing on the latent grid with FNO [29] or transformers [1, 50]. Alternatives such as end-to-end message-passing based graph neural networks [38, 18, 44, 43, 8, 16, 36] or end-to-end transformer based models [49, 33, 20, 48, 45] have also been proposed.

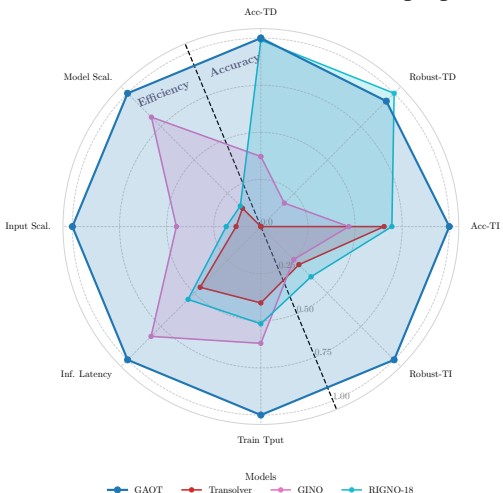

Figure 1: Normalized performance of GAOT and baselines across eight axes, covering accuracy (Acc.), robustness (Robust), throughput (Tput), scalability (Scal.) on time-dependent (TD) and time-independent (TI) tasks.

A thorough comparison of the existing models, not just in terms of accuracy but also computational efficiency and scalability, is necessary to evaluate whether these models are yet capable enough to act as surrogates for highly successful finite element methods for engineering simulations [41]. As one of the contributions in this paper, we performed such a comparison (see Sec. 3 for details) to find evidence for an *accuracy-efficiency trade-off*, i.e., accurate and robust models, such as the message passing based RIGNO of [36] are not necessarily computationally efficient nor scalable in terms of training throughput and inference latency. On the other hand, more efficient models such as GINO [29] are not accurate enough (see the accompanying Radar Chart in Fig. 1). Given this observation, our main goal in this paper is to propose an algorithm to learn PDE solution operators on arbitrary domains that is accurate, computationally efficient, and can be seamlessly scaled to real-world industrial simulations.

To this end, we propose *Geometry Aware Operator Transformer* (GAOT, pronounced goat) as a neural surrogate for PDEs on arbitrary domains. While being based on the well-established *encode-process-decode* paradigm [38], GAOT includes several novel features that are designed to ensure both computational efficiency and accuracy, such as

- Our proposed *multiscale attentional graph neural operator* (MAGNO) as the encoder between inputs on an arbitrary point cloud and a *coarser* latent grid, designed to enhance accuracy through its multiscale information processing and attention modules.

- Novel *Geometry embeddings* in the encoder (and decoder) that provide the model with access to information about the (local) domain geometry, greatly increasing accuracy.

- A *transformer processor* that utilizes patching (as in ViTs [12]) for computational efficiency.

- A MAGNO decoder, able to generate *neural fields*, with the ability to approximate the underlying solution at *any query point* in the domain.

- A set of novel implementation strategies to ensure that the computational realization of GAOT is efficient and highly scalable.

Combining these elements allows GAOT to treat PDEs on arbitrary domains in a robust, accurate and computationally efficient manner. We demonstrate these capabilities by,

- Extensively testing GAOT on 28 challenging benchmarks for both time-independent and time-dependent PDEs of various types, ranging from regular grids to random point clouds to highly unstructured adapted grids, and comparing it with widely-used baselines to show that GAOT is both highly accurate as well as computationally efficient and scalable, see Fig. 1.

- The efficiency and scalability of GAOT is further showcased by it achieving state of the art (SOTA) performance on the large scale three-dimensional industrial benchmark of *DrivAerNet++* dataset for automobile aerodynamics [14]. We also test GAOT and demonstrate its superior performance to the GINO baseline on two further 3D industrial scale datasets, i.e.,

DrivaerML dataset for automobile aerodynamics and a NASA-CRM dataset for aerospace applications.

- Through extensive ablations, we also highlight how the novel elements in the design of GAOT such as multiscale attentional encoders and geometry embeddings crucially contribute to the overall performance of our model.

## 2  Methods.

**Problem Formulation.** We start with a generic *time-independent PDE*,

$$\mathcal{D}(c, u) = f, \quad \forall x \in D \subset \mathbb{R}^d, \quad \mathcal{B}(u) = u_b, \quad x \in \partial D, \tag{1}$$

with $u : D \mapsto \mathbb{R}^m$, the PDE solution, $c$ is the coefficient (PDE parameters), $f$ is the forcing term, $u_b$ are boundary values and $\mathcal{D}$ and $\mathcal{B}$ are the underlying differential and boundary operators, respectively. Denoting as $\chi_D$, a function (e.g. indicator or signed distance) parameterizing the domain $D$, we combine all the *inputs* to the PDE (1) together into $a = (c, f, u_b, \chi_D)$, then the *solution operator* $\mathcal{S}$ maps inputs into PDE solution with $u = \mathcal{S}a$. The corresponding *operator learning task* is to learn the solution operator $\mathcal{S}$ from data. To this end, let $\mu$ be an underlying *data distribution*. We sample i.i.d. inputs $a^{(i)} \sim \mu$, for $1 \leq i \leq M$ and assume that we have access to *data pairs* $(a^{(i)}, u^{(i)})$ with $u^{(i)} = \mathcal{S}a^{(i)}$, Thus, the operator learning task is to approximate the distribution $\mathcal{S}_{\#\mu}$ from these data pairs. In practice, we can only access *discretized* versions of the data pairs, sampled on collocation points (which can vary over samples).

Similarly denoting a generic *time-dependent PDE* as,

$$u_t + \mathcal{D}(c, u) = 0, \quad \forall x \in D \subset \mathbb{R}^d, \ t \in [0, T] \quad u(0) = u_0, \quad x \in D, \tag{2}$$

with, $u : D \times [0, T] \mapsto \mathbb{R}^m$, $c$ the PDE coefficient and $u_0$ the initial datum and the underlying (spatial) differential operator $\mathcal{D}$. Clubbing the *inputs* to the PDE (2) into $a = (c, u_0, \chi_D)$, the corresponding *solution operator* $\mathcal{S}_t$, with $u(t) = \mathcal{S}_t(a)$ for all $t \in [0, T]$, maps the input into trajectory of the solution. The *operator learning task* consists of approximating $(\mathcal{S}_t)_{\#\mu}$ from data pairs $(a^{(i)}, u^{(i)}(t))$ for all $t \in [0, T^{(i)}]$ and $1 \leq i \leq M$ with samples $a_i$ drawn from the data distribution $\mu$. However in practice, we only have access to data, sampled on a discrete set of spatial points per sample as well as only on discrete time snapshots $t_n^{(i)} \in [0, T^{(i)}]$ and have to learn the solution operator from them.

**GAOT Model Architecture.** The overall architecture of GAOT is depicted in Fig. 2. For simplicity of exposition, we start with the time-independent case, where given inputs $a(x_j)$ on the point cloud $D_\Delta = \{x_j\} \subset D$, for $1 \leq j \leq J$, GAOT provides an approximation to solution $u$ of the PDE (1) at any query point $x \in D$. At a high level, GAOT follows the *encode-process-decode* paradigm of [38]. In the first step, an *encoder* transforms the input on the underlying point cloud $D_\Delta$ to a *latent point cloud* $\mathcal{D} \subset \mathbb{R}^{\tilde{d}}$. The resulting *spatial tokens* are then processed by a processor module to learn useful representations and its output is remapped to the original domain $D$ via the *decoder*, which allows evaluation at any query point $x \in D$.

**Choice of Latent Domain.** As depicted in **SM** Fig. B.1, the latent domain $\mathcal{D}$ (to which the encoder maps) can be chosen in three different ways, namely i) a regular (structured) grid stencil, consisting of equispaced points on a Cartesian domain (see also Fig. 2) ii) randomly downsampling the underlying point cloud $D_\Delta$ or iii) a projected low-dimensional representation, where a high-dimensional domain is projected to a lower dimension (for instance using tri-plane embeddings in 3-D [10]) and a regular grid is used in the lower-dimensional domain. GAOT is a general framework where any of these latent point cloud choices can be employed for $\mathcal{D}$.

**Encoder.** Given input values $a(x_j)$ on the underlying point cloud $D_\Delta$, the encoder aims to transform it into latent features $w_e(y)$ at any point $y \in \mathcal{D}$ on the latent point cloud. Using a graph-neural operator (GNO) encoder as in GINO [29] would lead to,

$$\widetilde{w}_e(y) = \sum_{k=1}^{n_y} \alpha_k K(y, x_k, a(x_k)) \varphi(a(x_k)), \tag{3}$$

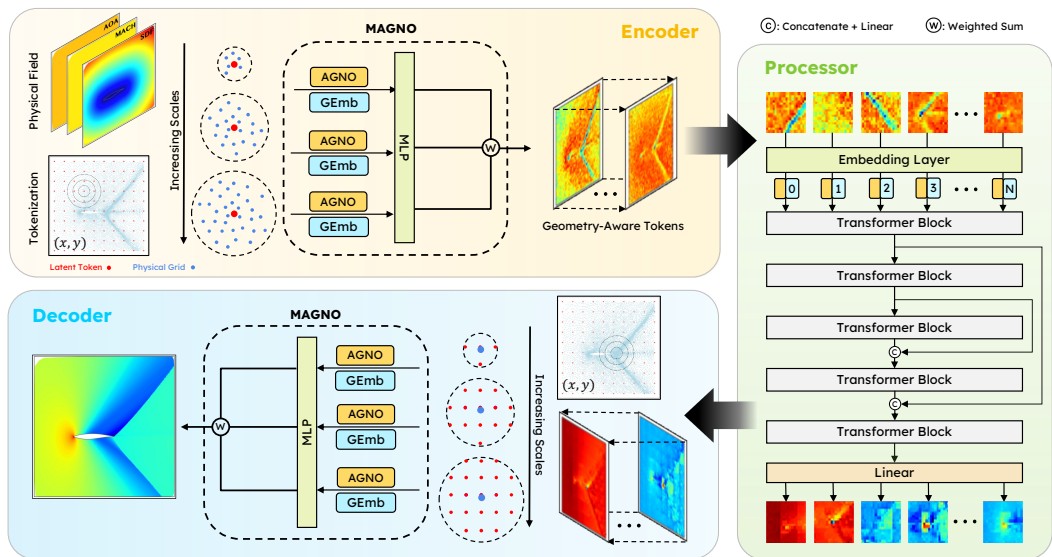

Figure 2: Schematic of the GAOT with an equispaced latent token grid. The encoder uses a multiscale attentional graph neural operator (MAGNO) to aggregate the input data into geometry-aware tokens. A vision transformer (ViT) block with residual connections processes tokens, enabling global exchange of information. A MAGNO decoder identifies the nearest tokens around a given query point to decode the final field.

with MLPs $K$ and $\varphi$ and the sum above taken over all the $n_y$ points $x_k \in D_\Delta$ such that $|y - x_k| \leq r$ for some hyperparameter $r > 0$, where $\alpha_k$ are some given quadrature weights. In other words, a GNO accumulates information from all the points in the original point cloud that lie inside a ball of radius $r$, centered at the given point $y$ in the latent point cloud, and processes them through a kernel integral.

Our first innovation is based on the realization that this *single-scale* approach might be limiting the overall accuracy. Instead, we would like to introduce a mechanism to integrate *multiscale* information into the encoder. To this end and as shown in Fig. 2, we choose $r_m = s_m r_0$, for some base radius $r_0$ and scale factors $s_m$, for $m = 1, \ldots, \bar{m}$ to modify GNO (3) by,

$$\widetilde{w}_e^m(y) = \sum_{k=1}^{n_y^m} \alpha_k^m K^m(y, x_k, a(x_k))\varphi(a(x_k)), \tag{4}$$

for any fixed scale $m$ and with MLPs $K^m, \varphi$. The above sum is taken over all the $n_y^m$ points $x_k \in D_\Delta$ such that $|y - x_k| \leq r_m$. To choose the quadrature weights $\alpha_k^m$, we propose an *attention based* choice,

$$\alpha_k^m = \frac{\exp(e_k^m)}{\sum_{k'=1}^{n_y^m} \exp(e_{k'}^m)}, \quad e_k^m = \frac{\langle \mathbf{W}_q^m y, \mathbf{W}_\kappa^m x_k \rangle}{\sqrt{\bar{d}}}, \tag{5}$$

with $\mathbf{W}_q^m, \mathbf{W}_\kappa^m \in \mathbb{R}^{\bar{d} \times m}$ are query and key matrices respectively, completing the description of the *attentional graph neural operator* (AGNO) (4) at each scale $m$.

**Geometry Embeddings.** The only geometric information in the afore-described encoder is provided by the coordinates of the underlying points. This alone does not convey the rich geometric information about the domain that can affect the solution of the underlying PDE (2). Hence, we need to embed further geometric information into the model. Deviating from the literature where geometric information is provided either by appending them as node and edge features on the underlying graphs [18] or by encoding a signed distance function [29], we propose to use novel *geometry embeddings* to encode this information. To this end and as described in **SM** Sec. B.3, we can rely on *local statistical embeddings* for each point $y \in \mathcal{D}$ as all the neighboring points $x_k$ in $D_\Delta$ with $|y - x_k| \leq r_m$ have already been computed in the AGNO encoder. From these points, we can readily compute statistical

descriptors such as i) number of neighbors $x_k \in D_\Delta$, in the ball $B_{r_m}(y)$, ii) the *average distance* $D_{\mathrm{avg}} = \frac{1}{n_y^m} \sum_{k=1}^{n_y^m} |y - x_k|$, iii) the variance of this distance $D_{\mathrm{var}}$, with respect to the average $D_{\mathrm{avg}}$, iv) the *centroid offset vector* $\Delta_y = \frac{1}{n_y^m} \sum_{k=1}^{n_x^y} (x_k - y)$ and v) a few principal component (PCA) features of the covariance matrix of $y - x_k$ to calculate the *local shape anisotropy*. These statistical descriptors, for each scale $m$ and each point $y \in \mathcal{D}$ are then concatenated into a single vector $z_y$, normalized across components to yield zero mean and unit variance and fed into an MLP to provide the embedding $g^m(y)$. Alternatively, geometry embedding using *PointNet* models [39] can also be considered.

**MAGNO.** As shown in Fig. 2, the scale-dependent AGNO $\widetilde{w}_e^m$ (4) and the geometry embedding $g^m$, at each scale $m$, can be concatenated together and passed through another MLP to yield a scale-specific latent features function $\widehat{w}^m(y)$. Next, we need to integrate these features across all $m$ scales. Instead of naively summing these scale contributions, we observe that different scales might contribute differently for every latent token to the encoding. To ascertain this relative contribution, we introduce a (small) MLP $\psi_m$ and weigh the relative contributions with a *softmax* and combine them into the *multiscale attentional graph neural operator* or MAGNO encoder by setting,

$$w_e(y) = \sum_{m=1}^{\bar{m}} \beta_m(y)\widehat{w}^m(y), \quad \forall y \in \mathcal{D}, \quad \beta_m(y) = \frac{\exp(\psi_m(y))}{\sum\limits_{m'=1}^{M} \exp(\psi_{m'}(y))} \tag{6}$$

**Transformer Processor.** The encoder provides a set of *geometry aware tokens* $w_e(y_\ell)$, for all points $y_\ell \in \mathcal{D}$, with $1 \leq \ell \leq L$, in the latent point cloud. These tokens are further transformed by a processor. As shown in Fig. 2, we choose a suitable transformer based processor. While postponing details on the processor architecture to **SM** Sec. B.4, we summarize our choices here. If the latent points $\{y_\ell\}$ lie on a regular grid (either through a structured stencil or a projected low-dimensional one), we use a patch-based *vision transformer* or ViT ([12] and [22, 35] for PDE operator learning) for computational efficiency. The equispaced latent points are combined into patches and the tokens in each patch are flattened into a single token embedding which serves as the input for a multi-head attention block, followed by a feed forward block. RMS normalization is applied to the tokens before processing. Either sinusoidal absolute position embeddings or rotary relative position embeddings are used to encode token positions. If the latent points $y_\ell$ are randomly downsampled from the original point cloud, there is no obvious way to patch them together. Hence, a standard transformer [47], but with RMS normalization, can be used. Additionally, we employ multiple skip connections across transformer blocks (see Fig. 2). The transformer processor transforms the tokens $w_e(y_\ell)$ into processed tokens, that we denote by $w_p(y_\ell)$, for all $1 \leq \ell \leq L$.

**Decoder.** Given any query point $x \in D$ in the original domain, the task of the decoder in GAOT is to provide $w(x)$, which approximates the solution $u$ of the PDE (1) at that point. To this end, we simply employ the MAGNO architecture in reverse. By choosing a base radius $\hat{r}_0$ and scale factors $\hat{s}_m$, a set of increasing radii $\hat{r}_m = \hat{s}_m \hat{r}_0$ are selected to define a set of increasing balls $B_{\hat{r}_m}(x)$ around the query point $x$ (Fig. 2). A corresponding AGNO model is defined by replacing $y \to x$, $x_k \to y_\ell$ and $a \to w_p$ in (4), with corresponding attentional weights computed via (5). In parallel, *geometry embeddings* over each ball $B_{\hat{r}_m}(x)$ are computed to provide statistical information about how the latent points $y_\ell$ are distributed in the neighborhood of the query point $x$. These AGNO features and geometry embeddings are concatenated and passed through a MLP to provide $w(x)$ which has the desired dimensions of the solution $u$ of the PDE (1). We denote the GAOT model as $\mathcal{S}_\theta$ with the output $w = \mathcal{S}_\theta(a)$, for the inputs $a$ to the PDE (1). It is trained to minimize the mismatch the underlying operator $\mathcal{S}$, i.e, the parameters $\theta$ are determined to minimize a loss $\mathcal{L}(\mathcal{S}(a), \mathcal{S}_\theta(a))$, over all input samples $a^i$, with $\mathcal{L}$ being either the absolute or mean-square errors.

**Extension to time-dependent problems.** To learn the solution operator $\mathcal{S}_t$ of the time-dependent PDE, we observe that the $\mathcal{S}_t$ can be used to update the solution forward in time, given the solution at any time point $u(t)$ by applying $u(t + \tau) = \mathcal{S}_\tau(u(t))$. Thus, for any time $t$, given the augmented input $a(t) = (c, u(t))$, with $c$ being the coefficient in the PDE (2), we need GAOT to output $u(t + \tau)$, for any $\tau \geq 0$. To this end, we retain the architecture of GAOT, as described for the time-independent case above, and simply add the current time $t$ and the *lead-time* $\tau$ as further inputs to the model.

Table 1: Benchmark results on time-dependent and time-independent datasets. Best and 2nd best models are shown in blue and orange fonts for each dataset.

| Dataset | Median relative $L^1$ error [%] | | | | | |
|---|---|---|---|---|---|---|
| **Time-Independent** | **GAOT** | **RIGNO-18** | **Transolver** | **GNOT** | **UPT** | **GINO** |
| Poisson-C-Sines | **3.10** | **6.83** | 77.3 | 100 | 100 | 20.0 |
| Poisson-Gauss | **0.83** | 2.26 | **2.02** | 88.9 | 48.4 | 7.57 |
| Elasticity | **1.34** | **4.31** | 4.92 | 10.4 | 12.6 | 4.38 |
| NACA0012 | **6.81** | **5.30** | 8.69 | 6.89 | 16.1 | 9.01 |
| NACA2412 | **6.66** | **6.72** | 8.51 | 8.82 | 17.9 | 9.39 |
| RAE2822 | 6.61 | **5.06** | **4.82** | 7.15 | 16.1 | 8.61 |
| Bluff-Body | **2.25** | 5.76 | **1.78** | 44.2 | 5.81 | 3.49 |
| **Time-Dependent** | **GAOT** | **RIGNO-18** | **GeoFNO** | **FNO DSE** | **UPT** | **GINO** |
| NS-Gauss | **2.91** | **2.29** | 41.1 | 38.4 | 92.5 | 13.1 |
| NS-PwC | **1.50** | **1.58** | 26.0 | 56.7 | 100 | 5.85 |
| NS-SL | **1.21** | **1.28** | 13.7 | 22.6 | 51.5 | 4.48 |
| NS-SVS | **0.46** | **0.56** | 9.75 | 26.0 | 4.2 | 1.19 |
| CE-Gauss | **6.40** | **6.90** | 42.1 | 30.8 | 64.2 | 25.1 |
| CE-RP | **5.97** | **3.98** | 18.4 | 27.7 | 26.8 | 12.3 |
| Wave-Layer | **5.78** | **6.77** | 11.1 | 28.3 | 19.6 | 19.2 |
| Wave-C-Sines | **4.65** | **5.35** | 13.1 | 5.52 | 12.7 | 5.82 |

More precisely, the time-dependent version of GAOT is of the form $\widehat{\mathcal{S}}_\theta(x, t, \tau, a(t))$, where $a(t)$ takes values at points sampled in $D$. Following [36], the map $\hat{S}_\theta$ can be used to update an approximate solution of PDE (2) in time by following a very general time-stepping strategy:

$$\mathcal{S}_\theta(t, \tau, a(t)) = \gamma u(t) + \delta \widehat{\mathcal{S}}_\theta(x, t, \tau, a(t)). \qquad (7)$$

Here, choosing the parameters $(\gamma, \delta)$ appropriately leads to different strategies for time stepping: $\gamma = 0, \delta = 1$ directly approximates the *output* of the solution operator at time $t + \tau$; $\gamma = 1, \delta = 1$ yields the *residual* of the solution at the later time, with respect to the solution at current time; $\gamma = 1, \delta = \tau$ is equivalent to approximating the *time-derivative* of the solution. GAOT provides the flexibility to use any of these time-stepping strategies. We also use the *all2all* training strategy [22] to leverage trajectory data for time-dependent PDEs.

**Efficient implementation.** As our goal is to ensure accuracy and computational efficiency, we have designed GAOT with ability for large-scale computations in mind. We started with the realization that the heaviest burden of the computation should fall on the processor. The encoder and decoder are often responsible for memory overheads as these modules entail sparse computations on graphs with far more edges than nodes, making the computations largely edge-based and leading to high (and inefficient) memory usage. Moreover, in many PDE learning tasks on arbitrary geometries, the underlying domain (and the resulting graph) varies significantly between data samples, making load balancing very difficult.

To address these computational challenges, we resorted to i) moving the graph construction outside the model evaluation by either storing the graph, representing the input point cloud, in memory for small graphs or on disk for large graphs and loading them during training with efficient data loaders ii) sequentially processing each input in a given batch for the encoder and decoder, while still batch processing in the transformer processor, allowing us to reduce memory usage while retaining efficiency and iii) if needed for very large-scale datasets, we use an edge-dropping strategy to further increase the memory usage of the encoder and decoder. These innovations are essential to ensure batch training and underpin the efficiency of GAOT, even when input geometries vary significantly. A more detailed discussion on these *novel implementation tricks* is provided in SM E.2.

## 3 Results.

**Datasets and Baselines.** We start by testing GAOT of a challenging suite of 15 datasets for PDEs with input/output data on arbitrary point clouds in two space dimensions, see **SM** Secs D and G

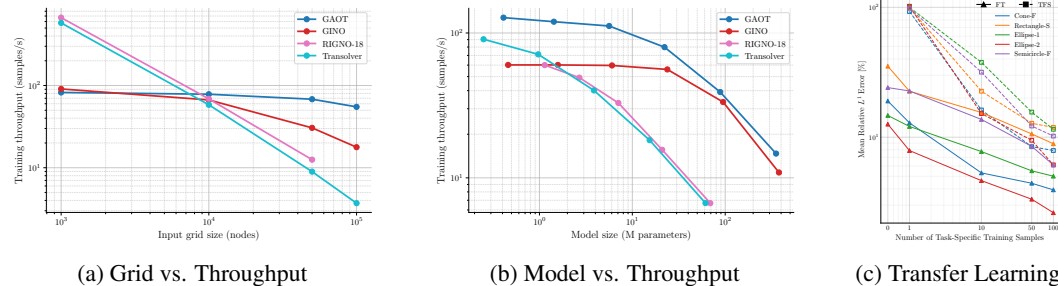

| (a) Grid vs. Throughput | (b) Model vs. Throughput | (c) Transfer Learning |

Figure 3: Training throughput (samples/s) with increasing input grid size (a) and model size (b) for proposed GAOT, GINO, RIGNO and Transolver. (c) Transfer learning performance of GAOT on unseen bluff body shapes (See SM Sec. E.10 for dataset details). FT (fine-tuning) adapts a pretrained GAOT model from Table 1, while TFS denotes training from scratch. FT consistently outperforms TFS across varying numbers of task-specific training samples.

for a detailed description of the datasets and for visualizations, respectively. For time-independent PDEs, in addition to the elasticity benchmark of [26], we consider two Poisson equation datasets: Poisson-Gauss, defined on random points in a square domain, and Poisson-C-Sines, a new dataset we propose, containing rich multiscale solutions on a circular domain. In addition, we propose 4 new datasets comprising compressible flows past objects, both airfoils as well as bluff bodies. These datasets have significant variation in domain geometry and flow conditions (Mach numbers ranging from subsonic to supersonic, varying angles of attack etc.) and are tailor-made for testing neural PDE surrogates on arbitrary domains in two space dimensions. For time-dependent PDEs, we test on the challenging datasets considered recently in [36], composed of 8 operators corresponding to the compressible Euler (2), incompressible Navier-Stokes (4), and acoustic wave equations (2). These time-dependent operators include complex multiscale solutions with shocks and other sharp traveling waves which can interact, reflect and diffract making them hard to learn. We test GAOT on these datasets and compare them with several widely used neural operators for PDEs on arbitrary domains including those based on message passing (RIGNO [36]), Fourier Layers (GINO [29], GeoFNO [26], FNO DSE[30]) and Transformers (Transolver [49], UPT [1] and GNOT [20]).

**Accuracy and Robustness.** In Table 1, we present the relative test errors for the above datasets to observe that GAOT is very accurate on all of them, being either the best (10) or second-best (4) model on 14 of them. On average, over the time-independent datasets, GAOT is almost 50% more accurate than the second-best performing model (RIGNO-18) while on time-dependent datasets, it is slightly more accurate than the second-best performing model (RIGNO-18). What is even more noteworthy is the *robustness* of the performance of GAOT over all the datasets. As seen from Table 1, the accuracy of GAOT is uniformly good over all the datasets and does not deteriorate on any of them. On the other hand, all the baselines show significantly poor performance on outlier datasets. This robustness can be quantified in terms of a *robustness score* (see **SM** Sec. E.3) to find that GAOT is almost three times more robust on the time-independent datasets as the second-best model (RIGNO-18), while GAOT and RIGNO-18 are as robust as each other on the time-dependent datasets.

**Computational Efficiency and Scalability.** It is worth reiterating that the computational efficiency of an ML model is a significant marker of overall performance. We test efficiency in terms of two critical quantities, the *training throughput* and the *inference latency*. For a given input and model size, training throughput measures the number of samples that a model can process during training (forward pass, backward pass and gradient update) per unit time (in seconds) on a given compute system (GPU or CPU). The higher the training throughput, the faster the

Table 2: Comparison of model size (Params.), throughput (samples/s), and latency (ms) across GAOT and representative baselines.

| Model | Params. (M) | Tput. | Latency |
|---|---|---|---|
| GAOT | 5.62 | 97.5 | 6.966 |
| GINO | 6.04 | 60.4 | 8.455 |
| RIGNO-18 | 2.69 | 50.3 | 12.74 |
| Transolver | 3.86 | 39.5 | 15.29 |

model can be trained. On the other hand, the inference latency is the amount of time it takes for a model to infer a single input. We present the throughput and latency for GAOT and three selected baselines (RIGNO-18 for Graph-based, GINO for FNO-based and Transolver for Transformer-based

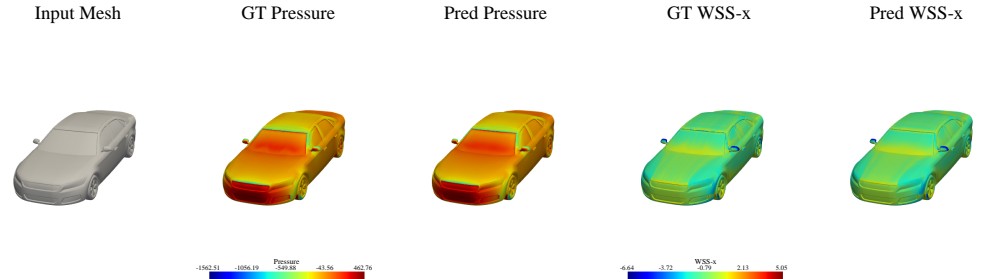

| Input Mesh | GT Pressure | Pred Pressure | GT WSS-x | Pred WSS-x |

Figure 4: Comparison of predicted and ground-truth (GT) results for the pressure and wall shear stress in the x-direction (WSS-x) on the DrivAerNet++ test sample `N_S_WWS_WM_172`.

models) in Table 2 for learning tasks (such as the bluff-body dataset for compressible flow) where the domain geometry varies throughout the dataset. These experiments are conducted on one NVIDIA GeForce RTX 4090 GPU with float32 precision. We see from this table that GAOT has the highest training throughput and the fastest inference latency, being almost $50\%$ and $15\%$, respectively better than the second-most efficient model (GINO).

How the training throughput of a model scales with increasing input and model size, is absolutely crucial for evaluating whether it can be used to process large-scale datasets (input scalability) or whether it can serve as a backbone of foundation models (model scalability) which require large model sizes [22]. To evaluate the scalability of different models, we plot how the training throughput changes as input size and model size (Fig. 3 (a, b)) are increased to find that GAOT scales much more favorably than the baselines with respect to both input and model size. In fact, models like Transolver and GNOT scale very poorly, making it impossible for us to train them for the large-scale time-dependent datasets with all2all training, which requires handling large volumes of data for large input sizes. Hence, we omit them in the accuracy results for time-dependent datasets in Tab. 1. The results for both accuracy and efficiency across a range of metrics for GAOT, RIGNO, GINO and Transolver are summarized in **SM** Tab. E.4 and visualized in the Radar chart Fig. 1. This demonstrates that GAOT ensures both accuracy (robustness) and computational efficiency (scalability) at the same time, while being the best model performing model on both sets of metrics.

**Industrial scale 3D datasets.** Given the high accuracy and excellent computational efficiency and scalability of GAOT, we showcase its abilities further on three challenging three-dimensional large-scale benchmarks for industrial simulations. We start with the DrivAerNet++ dataset of [14]. In this benchmark, the data consists of high-fidelity CFD simulations across $8K$ different car shapes which span the entire range of conventional car design. The underlying task is to learn steady-state surface fields (See Fig. 4) such as the pressure

Table 3: Error metrics of MSE ($\times 10^{-2}$) and Mean AE ($\times 10^{-1}$) for Pressure and Wall Shear Stress on the DrivAerNet++ dataset.

| Model | Pressure | | Wall Shear Stress | |
|---|---|---|---|---|
| | MSE | Mean AE | MSE | Mean AE |
| GAOT | 4.2694 | 1.0699 | 8.6878 | 1.5429 |
| FIGConvNet | 4.9900 | 1.2200 | 9.8600 | 2.2200 |
| TripNet | 5.1400 | 1.2500 | 9.5200 | 2.1500 |
| RegDGCNN | 8.2900 | 1.6100 | 13.8200 | 3.6400 |
| GAOT (NeurField) | 12.0786 | 1.7826 | 22.9160 | 2.5099 |

and wall shear stress, given the input car shape and flow conditions. The data has approximately $500K$ points per shape, making the overall training extremely compute intensive. Thus, only scalable models can currently process this learning task. We test GAOT on this challenging 3D benchmark and report the RMSE and MAE test errors for the pressure and wall shear stress in Tab. 3. Compared to baselines results taken from the leaderboard of the DrivAerNet++ challenge [10], we see that GAOT significantly improves on the state-of-the-art (see also Fig. 4). This improvement is most visible in the MAE for wall shear stress where GAOT is ca. $30\%$ more accurate than the second-best model (TripNet), which currently sits atop the leaderboard for wall shear stress predictions. We recall that GAOT's decoder endows it with *neural field* properties. We showcase it for the DrivAerNet++ dataset by training GAOT on a randomly selected set of less than $10\%$ of the total input points (per batch) and then testing on the original car surface point cloud by querying the desired points through GAOT's decoder. Although not as accurate as training GAOT with full input, we observe from Tab. 3 that this neural field version of GAOT has comparable accuracy to some of the baselines which have been trained with 10x more input points, further demonstrating the flexibility and accuracy of

Table 4: Comparison of GAOT and GINO across two benchmarks with MSE ($\times 10^{-2}$) and Mean AE ($\times 10^{-1}$). **Cp**: Pressure Coefficient, **WSS**: Wall Shear Stress, **P**: Pressure, **Cf**: Skin Friction Coefficient. **DML**: DrivaerML dataset, **CRM**: NASA CRM dataset.

| Model | Cp (DML) | | WSS (DML) | | P (CRM) | | Cf (CRM) | |
|-------|------|---------|------|---------|------|---------|------|---------|
| | MSE | Mean AE | MSE | Mean AE | MSE | Mean AE | MSE | Mean AE |
| GAOT | 5.1729 | 1.2352 | 16.9818 | 2.1640 | 7.7170 | 1.6014 | 16.1091 | 2.2305 |
| GINO | 8.8124 | 1.5238 | 28.4832 | 2.7330 | 10.5688 | 1.7450 | 21.1789 | 2.4240 |

Figure 5: Comparison of predicted and ground-truth (GT) results for the pressure on the test sample of DrivAerML and NASA-CRM.

GAOT. Further assessments of the neural field property of GAOT are provided in SM E.8. A natural follow-up question in this regard is whether training at full (high) resolution provides a significant advantage over training at low resolution and inferring at high resolution ? Recent works such as [15] have investigated this question in the context of the DrivAerNet++ dataset. We explore this issue in SM E.9 and demonstrate that GOAT trained at full resolution is significantly (almost twice) more accurate than competing models trained at low resolution and inferred at full resolution, see SM Table E.8.

Next, we consider the very challenging DrivAerML dataset of [3] (see Fig. 5), where the learning task is exactly the same as in DrivAernet++, i.e., predicting surface fields on the car, given its shape as the input. However, unlike DrivAernet++ which was based on coarse RANS simulations, DrivAerML's ground truth is based on highly accurate LES simulations. This enables the incorporation of much more fine-scale physical effects in this dataset. The learning problem becomes harder, not just in terms of the challenging underlying physics, but also the fact that the number of points on the car surface is now 9M, instead of 500K for DrivAernet++. Thus, only highly scalable models can deal with this extreme resolution. Consequently, we are only able to test GAOT and GINO for this dataset and report the results in Table 4 to observe that GAOT is significantly (almost twice on wall shear-stress) as accurate as GINO on this dataset.

Finally, we consider an industrial-scale dataset, recently proposed in [7], where the learning task (see Fig. 5), is to predict surface pressure and the skin friction coefficient, given the shape of a full aircraft. The ground truth is generated with RANS simulations using a Spalart-Allmaras turbulence model, and the results with GAOT and GINO are reported in Table 4, showing that GAOT significantly outperforms GINO on this large-scale industrial dataset.

**Generality, Generalization and Scaling.** We highlight GAOT's flexibility with respect to the point distributions that it can handle by testing it on PDEs with regular grid inputs, as suggested in [36]. To this end, we considered 7 additional datasets and present the test errors in **SM** Sec. E.5. to find that GAOT is highly accurate even on regular grids and is either more accurate or comparable to the highly expressive GNN-based RIGNO, while being more accurate than widely used neural operators such as FNO and CNO. A key requirement in operator learning [24, 6] is the ability of the model to generalize (at test time) to input and output resolutions that are different from the training resolution. As GAOT can be readily evaluated at any query point, we showcase this aspect of GAOT in **SM** Sec. E.7. by plotting the test errors for a sequence of resolutions, different from the training resolution, for the Poisson-Gauss benchmark, to find that GAOT generalizes very well in both the sub- and super-resolution settings, even to grids with 10x more input points than the training resolution. Another test of the generalizability of a model is its ability to perform well *out-of-distribution*, either zero-shot or when it is fine-tuned with a few in-distribution samples for the new learning task. To

test this aspect, we consider the datasets for compressible flow past bluff bodies and train a GAOT model on a set of bluff body shapes and then test it on shapes that were not in the training set. Then, the model is fine-tuned with a few task-specific samples and the results are shown in Fig. 3 (c). We observe that our model performs very well in a *few-shot transfer learning* scenario, with the fine-tuned model providing an almost order of magnitude gain in accuracy over the model, trained from scratch. Finally, in **SM** Sec. E.6, we demonstrate that GAOT scales with both model and dataset size, with scaling with respect to dataset size, also illustrated in Fig. 3 (c)

**Why does GAOT work so well ?**    To answer this question, we have performed extensive ablation studies in **SM** Sec.F to observe that i) the MAGNO encoder/decoder is clearly superior to message-passing based encoders/decoders, ii) choosing a regular equispaced latent point cloud performs significantly better than either downsampling on the original point cloud or using a projected low-dimensional regular grid, iii) GAOT is highly robust to the size of its latent grid, iv) a time-derivative marching strategy, i.e, setting $\gamma = 1, \delta = \tau$ in (7) is superior to other choices of $\gamma, \delta$, v) using a statistical geometric embedding performs significantly better than either not using additional geometric information or using a pointnet to process geometric information vi) incorporating *multiscale* features in the MAGNO encoder/decoder provides a significant gain in accuracy when compared to using just a single scale GNO encoder/decoder as in GINO and vii) The power of GAOT does not just stem from its VIT processor, but also from its MAGNO encoder/decoder (SM E.5), acting in tandem. These results justify the choices that we have made in designing GAOT and selecting the relevant model components, while also revealing how these innovative features underpin GAOT's accuracy. However, as argued before, this accuracy might come at the price of computational inefficiency. But, as we have demonstrated above, GAOT is also the most efficient model and does not have to pay the accuracy-efficiency trade-off. The reasons behind this boil down to the tricks used in efficiently implementing GAOT, which are discussed at length in SM E.2 and E.1.

## 4    Discussion

**Summary.** We present GAOT, a new neural operator for learning PDE solutions on arbitrary domains. It is based on a novel multiscale GNO encoder/decoder, combined with geometric embeddings that convey statistical information about the local domain geometry, and a (vision) transformer based processor. The model is designed to handle any point cloud input and provide the output at any query point in the underlying domain. Several innovative strategies have been used to make the implementation of GAOT computationally efficient and scalable. We test GAOT on a large number of challenging datasets for a variety of time-dependent and time-independent PDEs over diverse two-dimensional domain geometries to find that GAOT is significantly more accurate, robust and computationally efficient in terms of training throughput and inference latency, over a large set of baselines. We further demonstrate the potential of GAOT by presenting its SOTA performance on three large-scale three-dimensional datasets of industrial simulations in the automobile and aerospace sectors. These results demonstrate that GAOT can be a powerful and scalable neural operator with wide-spread applications. They also showcase the main advantage with an efficient and accurate neural operator such as GAOT, i.e, its inference time is many orders of magnitude faster than the runtime of a classical numerical PDE solver. We quantify this speedup in SM E.12 to find that GAOT can be anywhere between 4 to 9 orders of magnitude faster to run than classical PDE solvers, while retaining accuracy.

**Related Work.** As discussed before, there are 3 broad classes of models for learning PDEs on arbitrary domains namely i) end-to-end message-passing based frameworks exemplified here with RIGNO, which significantly improves upon models such as (multiscale) MeshGraphNets [38, 18] ii) Transformer based frameworks such as Transolver [49], GNOT [20] and UPT [1] and iii) frameworks, based on GNO encoders/decoders and FNO processors as in GINO [29]. GAOT differs from all these approaches by a) not using graph-based message passing, b) only employing transformers in the processor c) using a transformer, rather than FNO as a processor and significantly augmenting the GNO encoder/decoder by multiscale features, attention based-quadrature and geometry embeddings. It is precisely these choices, along with a highly efficient implementation, that allows GAOT to significantly surpass GINO, RIGNO, and Transolver in both accuracy and efficiency.

## Acknowledgements

The contribution of Siddhartha Mishra to this work was supported in part by the DOE SEA-CROGS project (DE-SC-0023191). The authors thank Mohamed Elrefaie and Prof. Faez Ahmed (MIT) for their inputs regarding the Carbench datasets and results.

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

# A Details of Problem Formulation.

Here, we introduce the core concepts of *operator learning* for both time-dependent and time-independent partial differential equations (PDEs). We begin by defining the general forms of PDEs under consideration and then discuss the associated operator-learning tasks.

## A.1 General Forms of PDEs

We focus on two broad classes of PDEs: time-dependent and time-independent.

**Time-Dependent PDE.** Let $D \subset \mathbb{R}^d$ be a $d$-dimensional spatial domain, and let $(0, T)$ denote the time interval. A general time-dependent PDE (see Main Text Eqn. (2)) can be written as

$$
\begin{aligned}
\frac{\partial}{\partial t} u(t, x) + \mathcal{D}\big(c, t, u, \nabla_x u, \nabla_x^2 u, \ldots\big) &= 0, & \forall (t, x) &\in (0, T) \times D, \\
\mathcal{B}\big(u, \nabla_x u, \nabla_x^2 u, \ldots\big) &= u_b, & \forall (t, x) &\in (0, T) \times \partial D, \\
u(0, x) &= u_0(x), & \forall x &\in D,
\end{aligned}
\tag{A.1}
$$

where

- $u(t, x)$ is the PDE solution in $(0, T) \times D$;
- $c(t, x)$ is a known, possibly spatio-temporal parameter (e.g., a material coefficient or source term);
- $\mathcal{D}$ is a (spatial) differential operator;
- $\mathcal{B}$ is a boundary operator acting on $\partial D$;
- $u_b(x)$ are boundary values;
- $u_0(x)$ is the initial condition at $t = 0$.

We assume $u(t, \cdot) \in \mathcal{X} \subset L^p(D; \mathbb{R}^m)$ for some $1 \leq p < \infty$ and integer $m \geq 1$. Likewise, $u_0(x) \in \mathcal{X}^0 \subset \mathcal{X}$ is an element of the initial-condition space, and $c \in \mathcal{Q} \subset L^p(\bar{D}; \mathbb{R}^m)$ is taken from a parameter space.

**Time-Independent PDE.** A time-independent (steady-state) PDE of the general form (Main Text Eqn. (1)) can be written as

$$
\begin{aligned}
\mathcal{D}\big(c, \bar{u}, \nabla_x \bar{u}, \nabla_x^2 \bar{u}, \ldots\big) &= f, & \forall x &\in D, \\
\mathcal{B}\big(\bar{u}, \nabla_x \bar{u}, \nabla_x^2 \bar{u}, \ldots\big) &= u_b, & \forall x &\in \partial D,
\end{aligned}
\tag{A.2}
$$

where $\bar{u}(x) \in \mathcal{X}$ and $c(x) \in \mathcal{Q}$ are now independent of $t$ and $f$ is a source term. In certain scenarios, one may view (A.2) as the long-time limit of (A.1), i.e.,

$$
\bar{u}(x) = \lim_{t \to \infty} u(t, x).
\tag{A.3}
$$

Hence, much of the theory for time-dependent PDEs can be adapted to time-independent problems by recognizing steady-state solutions as limiting cases.

## A.2 Solution Operators for PDEs

Let us denote the solution to the time-dependent PDE (A.1) by

$$
u(t, \cdot) = \mathcal{S}\big(a, c, t\big),
\tag{A.4}
$$

where $\mathcal{S} : \mathcal{X}^0 \times \mathcal{Q} \times (0, T) \to \mathcal{X}$ is the *solution operator*, mapping any initial datum $u_0 \in \mathcal{X}^0$ (and parameter functions $c \in \mathcal{Q}$) to the solution $u(t)$ at time $t$.

**Time-Shifted Operator.** In many operator-learning strategies, it is useful to consider a *time-shifted operator* that predicts solutions at a future time from a current snapshot. Specifically, define

$$\mathcal{S}^\dagger : \ \mathcal{X} \times \mathcal{Q} \times (0,T) \times \mathbb{R}^+ \ \longrightarrow \ \mathcal{X},$$

such that

$$\mathcal{S}^\dagger\big(u^t, \ c^t, \ t, \ \tau\big) \ = \ \mathcal{S}^t\big(u^t, \ c^t, \ \tau\big) \ = \ u^{t+\tau}. \tag{A.5}$$

Here, $u^t = u(t, \cdot)$ is the solution snapshot at time $t$, which now serves as an initial condition on the restricted time interval $(t, T)$. Likewise, $c^t$ is the corresponding parameter snapshot at time $t$.

**Steady-State Operator.** For the time-independent PDE (A.2), we define

$$\overline{\mathcal{S}} : \ \mathcal{Q} \ \longrightarrow \ \mathcal{X} \tag{A.6}$$

to be the analogous solution operator, such that $\bar{u} = \overline{\mathcal{S}}(c)$ solves the boundary-value problem for any parameter/boundary data $c$. Although many operator-learning methods primarily focus on the time-dependent form $\mathcal{S}$, the same ideas apply to steady-state problems by treating $\bar{u}$ as a limiting case.

**Operator Learning Task (OLT).** A central goal is to approximate these solution operators without repeatedly resorting to expensive, high-fidelity numerical solvers. Formally, the OLT can be stated as:

> *Given a data distribution $\mu \in \mathrm{Prob}(\mathcal{X}^0) \times \mathcal{Q}$ for initial/boundary conditions and parameters $c \in \mathcal{Q}$), learn an approximation $\mathcal{S}^* \approx \mathcal{S}$ to the true solution operator $\mathcal{S}$. That is, for any $a \sim \mu$, we want $\mathcal{S}^*(t, a)$ to closely approximate $u(t)$ for all $t \in [0,T]$. For time-independent problems, this goal changes accordingly to learning $\overline{\mathcal{S}}^* \approx \overline{\mathcal{S}}$.*

### A.3 Discretizations.

In practice, we only have access to a *discretized form* of the data as the labelled data is generated either through experiments/observations or numerical simulations. In both cases, we can only evaluate the inputs and outputs to the underlying solution operator at discrete points.

We start by describing these discretizations for the time-independent PDE (A.2). To this end, fix the $i$-th sample and let $D_{\Delta^{(i)}} = \{x_j^{(i)} \in D^{(i)}\}$, for $1 \leq j \leq J^{(i)}$ denote a set of *sampling points* on the underlying domain $D^{(i)}$. Observe that the underlying domain itself can be an input to the solution operator $\overline{\mathcal{S}}$ of (A.2). We assume access to the functions $\big(c^{(i)}(x_j), f^{(i)}(x_j), u^{(i)}(x_j)\big)$ and the corresponding discretized boundary values. Denoting these discretized inputs and outputs as $a_{\Delta^{(i)}}^{(i)}$ (where $a = (c, f, u_b)$) and $u_{\Delta^{(i)}}^{(i)}$, respectively, the underlying learning task boils down to approximating $\overline{\mathcal{S}}_{\#\mu}$ from the discretized data-pairs $\big(a_{\Delta^{(i)}}^{(i)}, u_{\Delta^{(i)}}^{(i)}\big)$. Note that although the data is given in a discretized form, we still require that our operator learning algorithm can provide values of the output function $u$ at any *query point* $x \in D$.

For the time-dependent PDE (A.1), in addition to the spatial discretization $D_{\Delta^{(i)}} = \{x_j^{(i)} \in D^{(i)}\}$, for $1 \leq j \leq J^{(i)}$, we only have access to data at time snapshots $t_n^{(i)} \in [0, T^{(i)}]$. Thus, the data to the time-dependent operator learning task consists of inputs $(c^{(i)}(x_j), u_0^{(i)}(x_j))$ and outputs $u(x_j^{(i)}, t_n^{(i)})$, from which the space- and time-continuous solution operator $\mathcal{S}_t$ has to be learned at every query point $x \in D$ and time point $t \in [0, T]$.

Summarizing, for both time-independent and time-dependent PDEs, the operator learning task amounts to approximating the underlying (space-time) continuous solution operators, given discretized data-pairs.

## B  Details of GAOT Architecture

This appendix provides a detailed explanation of the core components of the GAOT model architecture, including the choice of latent token grid, the Multiscale Attentional Graph Neural Operator (MAGNO) used in the encoder and decoder, the geometry embedding mechanisms, and the transformer-based processor.

### B.1 Choice of Latent Grid

Given the input point cloud, denoted by $D_\triangle$ above, the first step in our design is to select a *latent* point cloud, consisting of points at which our spatial tokens are going to be specified. Here, we explore three distinct ways of choosing spatial tokens, each offering different trade-offs in terms of computational cost, geometric coverage, and ease of patching for efficient attention. Figure B.1 schematically illustrates these strategies.

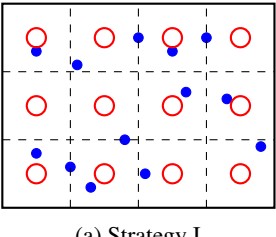 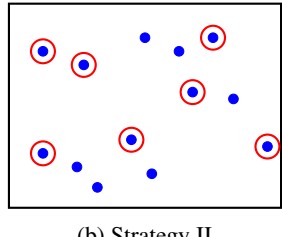 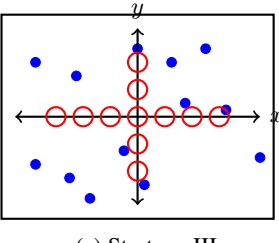

(a) Strategy I            (b) Strategy II           (c) Strategy III

Figure B.1: Schematic illustration of the three tokenization strategies in a 2D domain. Blue points and red circles corresponding to physical grid and latent token grid, respectively. **(a)** A structured stencil grid overlaying the domain. **(b)** Downsampled unstructured points used directly as tokens. **(c)** Projecting the 2D domain onto low-dimensional planes.

**Structured Stencil Grid (Strategy I).** In this approach, we overlay a structured grid of tokens across $D_\triangle$. For 2D domains, this may be a uniform mesh of cells; for 3D domains, an analogous dense grid can be used. GINO [29] is a good example for the use of such latent grids.

- *Advantages.* This grid can be quite fine if needed, ensuring adequate coverage. Moreover, we can group tokens into patches before feeding them into the transformer processor (see below), effectively reducing the token count (number of latent points). Our following experiments in **SM** Sec. F.2 suggest that patch size has a negligible effect on performance; hence, one can choose large patches to speed up training.

- *Limitations.* The main drawback is that token count grows exponentially with the dimension; for 3D, the number of grid cells can be prohibitively large. Also, if the input data lie on a low-dimensional manifold embedded in $D_\triangle$, some tokens may remain underutilized. Nonetheless, we find in practice that even empty tokens (those with no neighboring input points) can still contribute to better global encoding and improved convergence.

**Downsampled Unstructured Points (Strategy II).** This method directly downsamples the input unstructured point cloud and treat each sampled point as a token, RIGNO [36] and UPT [1] are typical methods based on this strategy. If the data are denser in some regions, naturally more tokens appear there.

- *Advantages.* This method avoids the pitfalls of having many tokens in empty regions, as might happen if the data indeed lie on a lower-dimensional manifold. By adaptive sampling, it can allocate tokens more efficiently.

- *Limitations.* Unstructured tokens are harder to patch effectively for attention mechanisms in the processor. In a following experiment, we observed that this strategy can be less effective than Strategy I even when the domain is indeed partially low-dimensional.

**Projected Low-Dimensional Grid (Strategy III).** Inspired by certain 3D computer vision models [9, 23], one can project the 3D domain (or higher-dimensional space) into a lower-dimensional representation and then place a structured stencil grid in the reduced coordinates—for example, using triplane embeddings in 3D [10].

- *Advantages.* Such a projection drastically reduces the token count in 3D, and avoids the purely *low-dimensional manifold* disadvantage of Strategy I. Moreover, one can still apply patching on the structured plane.

- *Limitations.* Decomposing $d$-dimensional coordinates into disjoint projections (e.g., splitting $x, y, z$ axes) can introduce additional approximation errors. Some local neighborhood

information is inevitably lost during the projection. This trade-off can degrade the final accuracy compared to direct methods (Strategy I or II).

In this paper, we mainly adopt Strategy I, i.e. a *structured stencil grid*, for all experiments due to its robustness and simplicity. While using a stencil grid indeed creates some empty tokens in low-density regions, we consistently observe fast convergence and strong generalization across various PDE datasets, see the ablation studies in **SM** Sec. F.2.

## B.2 Multiscale Attentional Graph Neural Operator

Both the encoder and decoder in GAOT (see Figure 2 in the main text) employ the proposed Multiscale Attentional Graph Neural Operator (MAGNO). MAGNO is designed to augment classical Graph Neural Operators (GNOs) by incorporating multiscale information processing and attention-based weighting. A traditional GNO constructs a local graph for each query point (or token) by collecting all neighboring nodes within a specified radius, approximating a (kernel) integral operator over this local neighborhood. Below, we first recap the standard single-scale GNO scheme, then extend it to a multiscale version, and finally incorporate attention mechanisms for adaptive weighting.

**Recap of Single-Scale Local Integration (GNO Basis)**   For any point $y$ in the latent space $\mathcal{D}$ (in the encoder) or a query point $x$ in the original domain $D_\Delta$ (in the decoder), a GNO layer aims to aggregate information from its neighborhood via a kernel integral. For the encoder, given input data $a(x_j)$ on the original point cloud $D_\Delta = \{x_j\}$, the GNO transforms it into latent features $w_e(y)$. The fundamental GNO computation is given by Eq. (3) from the main text:

$$\widetilde{w}_e(y) = \sum_{k=1}^{n_y} \alpha_k K(y, x_k, a(x_k))\varphi(a(x_k)) \tag{B.1}$$

where the sum is over $n_y$ points $x_k$ in the original point cloud $D_\Delta$ such that $|y - x_k| \leq r$. $K$ and $\varphi$ are MLPs, $a(x_k)$ is the input feature at point $x_k$, and $\alpha_k$ are given quadrature weights. This form can be seen as a discrete approximation of an integral operator:

$$\int_{B_r(y)\cap D_\Delta} K\big(y, x', a(x')\big)\varphi\big(a(x')\big)\mathrm{d}x' \tag{B.2}$$

where $B_r(y)$ is a ball of radius $r$ centered at $y$.

**Multiscale Neighborhood Construction**   The single-scale approach, while effective for capturing local interactions within a fixed radius $r$, may not efficiently perceive multiscale information crucial for many PDE problems. To address this, we introduce multiple radii. As described in the main text, we choose $r_m = s_m r_0$, where $r_0$ is a base radius and $s_m$ are scale factors ($m = 1, \ldots, \bar{m}$). For each scale $m$, we gather points $x_k$ from the original point cloud $D_\Delta$ within the ball $B_{r_m}(y)$ centered at $y \in \mathcal{D}$ (for the encoder) with radius $r_m$. A GNO-like local integration is then performed for each scale $m$, as shown in Eq. (4) from the main text:

$$\widetilde{w}_e^m(y) = \sum_{k=1}^{n_y^m} \alpha_k^m K^m(y, x_k, a(x_k))\varphi(a(x_k)) \tag{B.3}$$

Here, $n_y^m$ is the number of neighbors $x_k$ within radius $r_m$. The MLPs $K^m$ and $\varphi$ can be scale-specific or share parameters across scales. This paper chooses the shared parameters across all scales.

### B.2.1 Attentional Weighting in Local Integration (AGNO)

In the main text, we further propose an attention-based choice for the quadrature weights $\alpha_k^m$, as given by Eq. (5):

$$\alpha_k^m = \frac{\exp(e_k^m)}{\sum\limits_{k'=1}^{n_y^m} \exp(e_{k'}^m)}, \quad e_k^m = \frac{\langle \mathbf{W}_q^m y, \mathbf{W}_\kappa^m x_k \rangle}{\sqrt{\bar{d}}} \tag{B.4}$$

where $\mathbf{W}_q^m, \mathbf{W}_\kappa^m \in \mathbb{R}^{\bar{d}\times d}$ (assuming original and latent coordinate dimension $d$, and attention dimension $\bar{d}$) are learnable query and key matrices. This mechanism allows the model to dynamically assign contribution weights to each neighbor $x_k$ based on the relationship between $y$ and $x_k$. This forms the final form of our Attentional Graph Neural Operator or AGNO at each scale $m$.

### B.2.2 Attentional Fusion of Multiscale Features

After computing the AGNO features $\widetilde{w}_e^m(y)$ for each scale (which are then fused with geometry embeddings, detailed in Sec. B.3, to form $\widehat{w}^m(y)$), we need to integrate this information from different scales. As described in the main text (Fig. 2 and Eq. (6)), instead of simple summation or concatenation, we introduce a small MLP $\psi_m$ to learn the relative contribution of each scale to the final encoded feature $w_e(y)$:

$$w_e(y) = \sum_{m=1}^{\bar{m}} \beta_m(y)\widehat{w}^m(y), \quad \beta_m(y) = \frac{\exp(\psi_m(y))}{\sum\limits_{m'=1}^{\bar{m}} \exp(\psi_{m'}(y))} \tag{B.5}$$

Here, $\psi_m(y)$ is typically computed based on coordinates of $y$. $\beta_m(y)$ is the attention weight for the $m$-th scale at point $y$.

The final output of the MAGNO encoder, $w_e(y)$, is thus a feature representation that adaptively weights and fuses multiscale local information with attention mechanisms. The MAGNO in the decoder follows the exact same structure, with different inputs, outputs, and operating objects, as described in the Main Text.

### B.3 Geometry Embeddings

While the Multiscale Attentional GNO already leverages geometric structure via local neighborhoods, one often needs to incorporate more explicit shape or domain information in practical PDE scenarios. For instance, when the geometry of the domain itself (e.g., the shape of an airfoil) plays a critical role in the solution operator, coordinates alone may be insufficient to encode all the necessary geometric priors. Therefore, we introduce geometry embeddings to enhance the model's geometric awareness. These embeddings work in tandem with the MAGNO encoder (and decoder), providing a rich geometric description for each token (latent point $y$) and its neighborhood at various scales $m$.

Prior work on including geometric information in neural PDE solvers typically resorts to two major approaches: (i) appending geometry features directly into node/edge attributes [36], or (ii) using a signed distance function (SDF) [29]. However, we argue that:

- Simply merging geometry and physical features at the node level may entangle them prematurely, potentially hurting performance when the geometry is complex or when additional modalities (e.g. material properties) must be fused.

- Computing SDF to represent geometry is often cumbersome, especially for unstructured datasets or when the boundary is only partially known. Each new shape would require re-computation, and the SDF values may be inaccurate if the surface is not well-defined.

Instead, we advocate two more direct and flexible mechanisms for extracting geometric descriptors: local *Statistical embedding* and *PointNet-based embedding*, shown in Figure B.2.

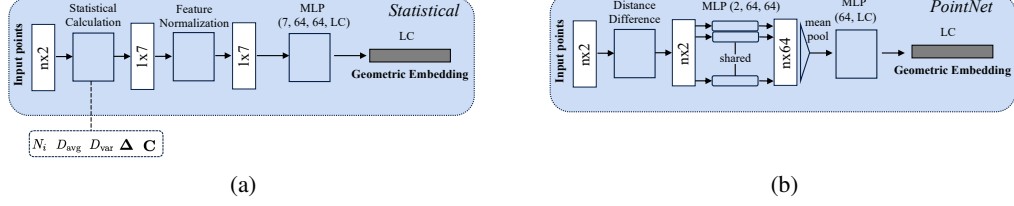

(a)                                               (b)

Figure B.2: Schematic of the Geometric Embedding for Statistical Embedding (a) and PointNet-based Embedding (b). LC denotes the lifting channels for MAGNO output.

**Local Statistical Embedding**   The core idea is to extract statistical descriptors from the neighborhood $B_{r_m}(y)$ (or $B_{\hat{r}_m}(x)$ for the decoder) of original point cloud points $x_k$ (or latent points $y_\ell$ for the decoder) for each latent point $y$ (for the encoder) or query point $x$ (for the decoder) at each scale $m$. Taking the encoder as an example, for a latent point $y \in \mathcal{D}$ and scale $m$, its neighborhood is $N_m(y) = \{x_k \in D_\Delta : |y - x_k| \le r_m\}$, containing $n_y^m$ points. We compute the following statistics:

- Number of Neighbors $n_y^m$: Measures local density around point $y$.

- Average Distance $D_{\text{avg}}^m(y)$:

$$D_{\text{avg}}^m(y) = \frac{1}{n_y^m} \sum_{k=1}^{n_y^m} |y - x_k| \tag{B.6}$$

Describes the average spatial extent of the neighborhood.

- Distance Variance $D_{\text{var}}^m(y)$:

$$D_{\text{var}}^m(y) = \frac{1}{n_y^m} \sum_{k=1}^{n_y^m} (|y - x_k| - D_{\text{avg}}^m(y))^2 \tag{B.7}$$

Reflects the dispersion of points within the neighborhood.

- Neighbor Centroid Offset Vector $\Delta_y^m$:

$$\Delta_y^m = \left( \frac{1}{n_y^m} \sum_{k=1}^{n_y^m} x_k \right) - y \tag{B.8}$$

The vector from $y$ to the centroid of its neighbors $x_k$.

- PCA Features: These features aim to capture the local shape anisotropy of the distribution of the $n_y^m$ neighbor points $\{x_k\}$ within the $m$-th scale ball $B_{r_m}(y)$. This is achieved by performing PCA on the set of these neighbor coordinates $\{x_k\}$. Using the centroid of the neighbors $\bar{x}_{\text{nbrs},y}^m = \left( \frac{1}{n_y^m} \sum_{k=1}^{n_y^m} x_k \right)$, the $d \times d$ covariance matrix of the neighbor coordinates is calculated as:

$$\mathbf{C}_y^m = \frac{1}{n_y^m} \sum_{k=1}^{n_y^m} (x_k - \bar{x}_{\text{nbrs},y}^m)(x_k - \bar{x}_{\text{nbrs},y}^m)^\top \tag{B.9}$$

If $n_y^m = 0$ (or too few points for a meaningful covariance, e.g. $n_y^m < d$), the covariance matrix $\mathbf{C}_y^m$ is treated as a zero matrix, leading to zero eigenvalues. Otherwise, the $d$ real eigenvalues of this symmetric, positive semi-definite covariance matrix, sorted in descending order ($\lambda_1^m \geq \lambda_2^m \geq \cdots \geq \lambda_d^m \geq 0$), are used as the PCA features. These eigenvalues represent the variance of the neighbor data along the principal component directions, thus describing the extent and orientation of the local point cloud cluster.

These statistical descriptors, computed for each scale $m$ and each point $y \in \mathcal{D}$, are concatenated into a vector $z_y^m$, normalized (e.g., to have zero mean and unit variance for each component), and then fed into an MLP to yield the geometry embedding $g^m(y)$ for that scale:

$$g^m(y) = \text{MLP}_{\text{geo}}(\text{Normalize}(z_y^m)) \tag{B.10}$$

This $\text{MLP}_{\text{geo}}$ is typically shared across all points and scales.

**Point-Based Embedding** As an alternative, we can train a PointNet-style network [40] to derive a compact geometric descriptor from each token's neighborhood. Classical PointNet architectures typically include:

- Input Transformer: aligns input points to a canonical space (optional),
- Shared MLP: processes each point individually,
- Symmetric Pooling: aggregates per-point features into a global descriptor, ensuring permutation invariance.

In our PDE setting, we do not necessarily need an input transformer; the local coordinates can directly serve as input features. We replace the typical max-pooling with mean-pooling to produce smoother local embeddings (though other pooling strategies are also possible). For a point $y$ and scale $m$, we collect the relative coordinates of its neighbors $\{\delta_k^m = x_k - y\}_{k=1}^{n_y^m}$. These relative coordinates are fed into a shared MLP (point-wise MLP):

$$h_k^m = \text{MLP}_{\text{pt}}(\delta_k^m) \tag{B.11}$$

Then, a symmetric pooling operation (e.g., mean pooling or max pooling) aggregates these per-point features into a global geometric feature:

$$\bar{h}^m(y) = \text{MeanPool}(\{h_k^m\}_{k=1}^{n_y^m}) = \frac{1}{n_y^m} \sum_{k=1}^{n_y^m} h_k^m \tag{B.12}$$

This aggregated feature $\bar{h}^m(y)$ can optionally be passed through another small MLP to produce the final geometry embedding $g^m(y)$.

### B.3.1 Integration of Geometry Embeddings in MAGNO

As depicted in Fig. 2 of the main text and described in the MAGNO paragraph, in the MAGNO component of the encoder (or decoder), the geometry embedding is fused with the AGNO output at each scale. The specific workflow (for the encoder) is as follows:

1. **Scale-Specific AGNO Features**: For latent point $y$ and scale $m$, compute the AGNO output $\widetilde{w}_e^m(y)$ (as described in Sec. B.2.1).

2. **Scale-Specific Geometry Embedding**: In parallel, using methods from Sec. B.3, compute the geometry embedding $g^m(y)$ from the same neighborhood $N_m(y)$.

3. **Feature Fusion**: Concatenate the AGNO features $\widetilde{w}_e^m(y)$ and the geometry embedding $g^m(y)$, and pass them through an MLP for fusion, yielding the scale-specific latent feature function $\widehat{w}^m(y)$:
$$\widehat{w}^m(y) = \text{MLP}_{\text{fuse}}^m([\widetilde{w}_e^m(y) \,\|\, g^m(y)]) \tag{B.13}$$
where $\|$ denotes concatenation. And $\text{MLP}_{\text{fuse}}^m$ is shared across scales.

4. **Multiscale Aggregation**: Finally, as described in Sec. B.2.2, an attention mechanism is used to perform a weighted sum of the fused features $\{\widehat{w}^m(y)\}_{m=1}^{\bar{m}}$ from all scales, yielding the final encoder output $w_e(y)$ (Eq. (B.5)).

Compared to merging geometry and PDE features at the node level before any operator updates, this *per-scale* integration offers several benefits:

- Scale-Adapted Geometry. Each scale has a correspondingly sized neighborhood, allowing the geometric embedding to reflect local shape details at the appropriate radius. Small radii capture fine-grained features (e.g. sharp corners), while large radii convey coarse global context.

- Modular Flexibility. Both MAGNO and Geometric Embeddings act as distinct modules. One can upgrade either component (e.g. adopting a more customized local aggregator or geometry encoder) without changing the overall pipeline.

- Unified Per-Token Fusion. The final aggregated feature $w_e(y)$ collects information from all relevant scales and from geometric descriptors, leading to a richer token representation. This is particularly advantageous in settings with complex boundaries (e.g. airfoils, porous media) where multiple length scales and shape cues matter.

This design preserves the encode-process-decode philosophy: each token gains geometry-aware, multiscale PDE features during the encoder stage, facilitating global attention and final decoding later in the pipeline.

### B.4 Processor

After constructing geometry-aware tokens, we employ a Transformer-based processor to enable global message passing among all tokens. Depending on the chosen tokenization strategy (**SM** B.1), we can choose the following strategies, respectively:

- Regular Grid (Strategy I or III): If the latent points $\{y_\ell \in \mathcal{D}\}$ lie on a regular grid (e.g., via a structured stencil or a projected low-dimensional regular grid), we adopt a strategy similar to vision transformers (ViTs) [12]. The latent points are grouped into non-overlapping "patches." All token features $w_e(y)$ within each patch are flattened and linearly projected into a single patch token embedding. These patch tokens then serve as the input sequence to the Transformer.

- Randomly Downsampled Points (Strategy II): If the latent points $\{y_\ell\}$ are randomly downsampled from the original point cloud $D_\Delta$, they lack a regular grid structure. In this case, there is no obvious "patching" method, and each latent token $w_e(y_\ell)$ directly serves as an element in the Transformer's input sequence.

**Positional Encoding**    Transformers themselves are permutation-invariant and do not inherently process sequential order or spatial position. Thus, positional information must be injected. In GAOT, we use the Relative Positional Embeddings (RoPE) [46], which is a method that integrates relative positional information directly into the self-attention mechanism. It achieves this by applying rotations, dependent on their relative positions, to the Query and Key vectors. This has shown strong performance in many Transformer models.

**Transformer Block Structure**    For the transformer Blocks, we adopt an RMS norm $\mathrm{RMSNorm}(\cdot)$ at the beginning of attention and feedforward layers:

$$\mathbf{z} = \mathrm{RMSNorm}(\mathbf{x}), \quad \mathrm{RMSNorm}(\mathbf{x}) = \frac{\mathbf{x}}{\sqrt{\mathrm{mean}(\mathbf{x}^2)}} \odot \boldsymbol{\alpha} \qquad (B.14)$$

where $\boldsymbol{\alpha}$ is a learned scaling parameter. This approach is akin to LayerNorm but uses the root mean square of feature magnitudes rather than computing mean-and-variance separately. This prenorm design helps stabilize the gradient flow compared to the conventions in [47]. Each block has the structure,

$$\mathbf{Z}_{\mathrm{attn}} = \mathbf{X} + \mathrm{MultiHeadAttn}(\mathrm{RMSNorm}(\mathbf{X})), \quad \mathbf{Z}_{\mathrm{ffn}} = \mathbf{Z}_{\mathrm{attn}} + \mathrm{FFN}(\mathrm{RMSNorm}(\mathbf{Z}_{\mathrm{attn}})).$$
$$(B.15)$$

Furthermore, we use *Group Query* and *Flash Attention* in the code for efficient multi-head self-attention.

**Long-Range Skip Connections**    In addition to the intra-block residual connections, we also introduce long-range skip connections across multiple Transformer blocks, as suggested in works like [5]. For instance, the Transformer blocks can be divided into an earlier part and a later part, and layers can be symmetrically connected (e.g., the first with the last, the second with the second-to-last, etc.), allowing later blocks to directly receive information from earlier blocks, further improving information flow.

By stacking these blocks, the Transformer processor learns complex global dependencies among tokens, transforming the locally geometry-aware tokens $w_e(y_\ell)$ from the encoder into processed tokens $w_p(y_\ell)$ that incorporate richer contextual information. These processed tokens are then converted by the MAGNO decoder to the desired approximation of the output of the underlying solution operator (see Main Text and Fig. 2).

### B.5   Graph Building Tricks

For 2D datasets, we construct neighborhoods for each latent token using a fixed-radius rule, following the Ref. [29]. This works well because local density variations are moderate and the total edge count remains manageable. However, this strategy is not scalable for large 3D industrial datasets, where meshes could be highly adaptive. A single global radius $r$ will lead to a sharp trade-off: if $r$ is large, the number of edges explodes in dense regions and quickly exceeds GPU memory; if $r$ is small, the Overlap of the local neighborhoods fails to cover the whole physical domain and induces information loss. To simultaneously control edge count and guarantee the whole domain coverage, we introduce a *bidirectional* graph-building strategy that merges two complementary graphs:

1. **Radius graph centered at latent tokens.** For each latent token $y_\ell$, we connect all physical points $x_j$ within a radius $r$, producing edges that encode local aggregation around latent tokens. This yields good locality, but may not be able to cover the physical regions if $r$ is too small.

2. **KNN graph centered at physical points.** For each physical grid point (node) $x_j$, we connect it to its $k$ nearest latent tokens, adding at least one attachment per physical node. This forms a lightweight backbone that ensures every physical location is represented in the latent space, even where radius neighborhoods are empty.

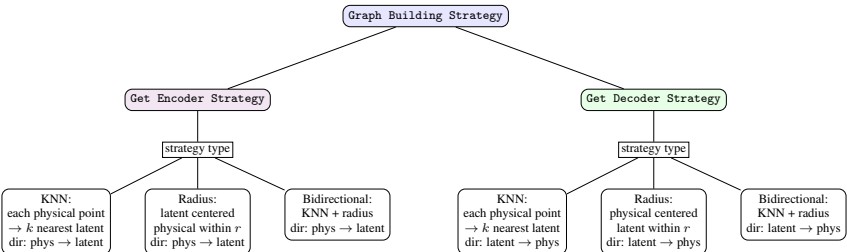

Figure B.3: Neighbor strategy dispatcher used in GAOT. For 2D datasets, a radius-based graph is adopted; for large 3D datasets, the bidirectional strategy (radius + KNN) is used to ensure domain coverage without edge explosion.

We then merge the two edge sets (remove the repeated pairs). The resulting bidirectional graph not only ensures all the physical points are considered, but control the total edge count through the tunable hyperparameter $k$ (typically $k \in [1, 4]$) and radius $r$. In practice, we use the encoder direction *physical $\rightarrow$ latent* and the decoder direction *latent $\rightarrow$ physical*; the same bidirectional recipe applies to both by swapping source/target roles, shown in Fig. B.3. Although the bidirectional build is more expensive than a pure-radius build, we precompute graphs offline and cache them, so the extra cost does not affect training throughput.

On all 2D benchmarks, we retain the pure-radius construction. On 3D industrial-scale datasets, we switch to the bidirectional construction to avoid radius-induced edge explosion while preserving coverage. To further control memory and improve robustness, we also apply *edge masking* during training.

## B.6    Training Details.

This section discusses the details of how the GAOT models were trained, including the loss functions, data normalization procedures, general training hyperparameters, and default model configurations. In our experiments, we address both time-independent and time-dependent PDEs. The application of GAOT to time-independent PDEs is straightforward. For time-dependent PDEs, we employ three different time-stepping methods and all2all training [22], as discussed in the main text. More details on these methods can be found in [36].

### B.6.1    Loss Function

The loss function used for training GAOT is the Mean Squared Error (MSE), computed between the model's final predictions and the true physical quantities. For a set of $N_s$ samples and $N_p$ spatial points, the loss is:

$$\mathcal{L}_{\text{MSE}} = \frac{1}{N_s N_p} \sum_{i=1}^{N_s} \sum_{j=1}^{N_p} \|\mathcal{S}_\theta(\cdot)_i(x_j) - \mathbf{u}_{\text{true},i}(x_j)\|_2^2 \tag{B.16}$$

where $\mathcal{S}_\theta(\cdot)_i(x_j)$ is the model's prediction for sample $i$ at point $x_j$, and $\mathbf{u}_{\text{true},i}(x_j)$ is the corresponding ground truth. The exact form of $\mathcal{S}_\theta(\cdot)$ depends on whether the problem is time-independent or time-dependent.

**Time-Independent PDEs.**    For time-independent PDEs, given an input $a(x_j)$ (e.g., boundary conditions, coefficients $c$), GAOT directly predicts the solution $u(x_j)$. Thus, $\mathcal{S}_\theta(a)(x_j)$ is the direct output of the GAOT architecture, and the MSE loss is computed between $\mathcal{S}_\theta(a)(x_j)$ and the true steady-state solution $u_{\text{true}}(x_j)$ of PDE (A.2).

**Time-Dependent PDEs.**    To learn the solution operator for time-dependent PDEs, GAOT is used to update the solution forward in time. Given the solution $u(t)$ at time $t$ and coefficients $c$ (forming the augmented input $a(t) = (c, u(t))$), the model predicts the solution at $t + \tau$. The GAOT architecture produces an output $\widehat{\mathcal{S}}_\theta(x, t, \tau, a(t))$. The final prediction for $u(t + \tau)$, denoted $\mathcal{S}_\theta(t, \tau, a(t))$, is constructed using a general time-stepping strategy as per Eq. (7) from the main text, see also [36]:

$$\mathcal{S}_\theta(t, \tau, a(t)) = \gamma u(t) + \delta \widehat{\mathcal{S}}_\theta(x, t, \tau, a(t)) \tag{B.17}$$

The MSE loss is then computed between this $\mathcal{S}_\theta(t, \tau, a(t))$ and the true solution $u_{\text{true}}(t + \tau)$. The choice of parameters $(\gamma, \delta)$ determines the time-stepping strategy and what the network output $\widehat{\mathcal{S}}_\theta$ effectively learns:

- **Output Stepping** ($\gamma = 0, \delta = 1$): The final prediction is $\mathcal{S}_\theta(t, \tau, a(t)) = \widehat{\mathcal{S}}_\theta(x, t, \tau, a(t))$. The network output $\widehat{\mathcal{S}}_\theta$ directly learns to approximate $u(t + \tau)$.

- **Residual Stepping** ($\gamma = 1, \delta = 1$): The final prediction is $\mathcal{S}_\theta(t, \tau, a(t)) = u(t) + \widehat{\mathcal{S}}_\theta(x, t, \tau, a(t))$. The network output $\widehat{\mathcal{S}}_\theta$ learns to approximate the residual, $u(t + \tau) - u(t)$.

- **Time-Derivative Stepping** ($\gamma = 1, \delta = \tau$): The final prediction is $\mathcal{S}_\theta(t, \tau, a(t)) = u(t) + \tau \cdot \widehat{\mathcal{S}}_\theta(x, t, \tau, a(t))$. The network output $\widehat{\mathcal{S}}_\theta$ learns to approximate the time-derivative, $(u(t + \tau) - u(t))/\tau$.

GAOT offers the flexibility to use any of these strategies. A detailed ablation of their comparative performance is described in **SM** Sec. F.

### B.6.2 Data Normalization

Data normalization is applied to stabilize training. We typically use Z-score normalization, where for a quantity $X$, its normalized version $\widehat{X}$ is $(X - \mu_X)/\sigma_X$. The mean $\mu_X$ and standard deviation $\sigma_X$ are computed over the training dataset.

**Time-Independent PDEs.** For input features $a(x_j)$ and output solution fields $u(x_j)$, normalization parameters are computed across all samples and spatial points in the training set for each channel independently. The model is trained on normalized inputs to predict normalized outputs.

**Time-Dependent PDEs.** The input $u(t)$ is normalized using its global mean and standard deviation computed over all time steps and samples in the training set. The normalization of the target for the network output $\widehat{\mathcal{S}}_\theta(x, t, \tau, a(t))$ depends on the chosen time-stepping strategy, as $\widehat{\mathcal{S}}_\theta$ learns a different physical quantity in each case:

- Output Stepping: The network $\widehat{\mathcal{S}}_\theta$ aims to predict $u(t + \tau)$. Thus, the ground truth values $u(t + \tau)$ are normalized, and $\widehat{\mathcal{S}}_\theta$ is trained to predict these normalized values. Statistics $\mu_u$ and $\sigma_u$ are computed from all values $u(t')$ in the training set. The normalized target for $\widehat{\mathcal{S}}_\theta$ is $\widehat{u}(t + \tau) = (u(t + \tau) - \mu_u)/\sigma_u$.

- Residual Stepping: The network $\widehat{\mathcal{S}}_\theta$ aims to predict the residual $R(t, \tau) = u(t + \tau) - u(t)$. Thus, these true residual values are computed from the training data, and their statistics $(\mu_R, \sigma_R)$ are used for normalization. The normalized target for $\widehat{\mathcal{S}}_\theta$ is $\widehat{R}(t, \tau) = (R(t, \tau) - \mu_R)/\sigma_R$.

- Time-Derivative Stepping: The network $\widehat{\mathcal{S}}_\theta$ aims to predict the time-derivative $D(t, \tau) = (u(t + \tau) - u(t))/\tau$. These true derivative values are computed, and their statistics $(\mu_D, \sigma_D)$ are used for normalization. The normalized target for $\widehat{\mathcal{S}}_\theta$ is $\widehat{D}(t, \tau) = (D(t, \tau) - \mu_D)/\sigma_D$.

Time $t$ and lead-time $\tau$ inputs are also typically scaled or normalized. Further details on these normalizations can be found in [36].

### B.6.3 General Training Setup

This section provides an overview of our training hyperparameters. Unless otherwise noted, all experiments follow these settings. Table B.1 summarizes the primary hyperparameters and training schedules. In particular, we distinguish between time-dependent and time-independent PDE tasks in terms of epoch count and highlight the differences in hardware usage for the industrial 3D dataset. All models except industrial 3D dataset run on a single GeForce RTX 4090 GPU. For the DrivAerNet++ dataset, we use 4 GeForce A100 GPUs in data parallel mode for 50 epochs, and each GPU holds a batch size of 1. For DrivAerML and NASA-CRM dataset, we use 1 GeForce A100 GPUs for 200 epochs. For the scheduler, we warm up to mitigate instability at early epochs, then adopt a cosine schedule for gradual decay, and finalize with a step-based drop for fine-tuning the last epoch range.

Table B.1: Key training hyperparameters and schedulers used for all models, unless otherwise specified.

| | |
|---|---|
| Hardware | • Single-GPU: All models for 2D dataset are trained on a single GeForce RTX 4090 with batch size $= 64$. |
| | • Single-GPU: Models for DrivAerML and NASA-CRM are trained on single GeForce A100 GPUs with batch size $= 1$. |
| | • Four-GPU: For the DrivaerNet++ dataset, we use four GeForce A100 GPUs, each with batch size $= 1$. |
| Optimizer | • Algorithm: AdamW |
| | • Weight Decay: $1 \times 10^{-5}$ |
| Epochs | • 2D Time-Dependent PDEs: 500 epochs |
| | • 2D Time-Independent PDEs: 1000 epochs |
| | • 3D DrivAerNet++: 50 epochs. |
| | • 3D NASA-CRM and DrivAerML: 200 epochs. |
| Learning Rate Scheduler | • Warmup (first 10% epochs): LR increases linearly from $8 \times 10^{-4}$ to $1 \times 10^{-3}$. |
| | • Cosine Decay (next 85% epochs): LR decays from $1 \times 10^{-3}$ to $1 \times 10^{-4}$. |
| | • StepLR (final 5% epochs): LR drops from $1 \times 10^{-4}$ to $5 \times 10^{-5}$. |

### B.6.4 GAOT Model Configuration

Table B.2 outlines the default configuration of the GAOT framework. This includes the MAGNO used in both the encoder and decoder stages, as well as the Transformer-based global processor. MAGNO converts node features into geometry-aware tokens (encoder) and reconstructs continuous fields (decoder). By default, the coordinates will be rescaled in the domain $[-1, 1]^d$, and we use a single aggregation radius $0.033$ for adequate coverage. If multiscale is enabled, we adopt radii $\{0.022, 0.033, 0.044\}$. The default geometric embedding method is local *Statistical Embedding* (e.g. as **SM** Sec. B.3), and will typically be implemented for unstructured datasets. The Transformer processes geometry-aware tokens globally via multi-head self-attention. We set the hidden dimension to 256 with a 1024-dim feed-forward layer, residual connections, RMSNorm, and RoPE for positional embeddings. By adjusting the patch size and the number of tokens, we can trade off computational cost and model resolution, discussed in **SM** Sec. F.2.

The configuration slightly differs for the industrial 3D datasets to accommodate large-scale adaptive meshes. We employ the following number of latent tokens for each dataset:

- **NASA-CRM:** [64, 64, 32],
- **DrivAerML:** [64, 32, 32],
- **DrivAerNet++:** [64, 64, 32].

In these setups, the graph construction follows the *bidirectional* strategy described in Appendix B.5, using a single-scale radius of $0.033$ and a KNN size of $k = 1$. The Transformer processor adopts a patch size of 2 and consists of 10 layers. All other hyperparameters remain consistent with the default configuration in Table B.2.

### B.7 Inference

When predicting solutions for *time-independent* PDEs, we simply feed the input parameters $a$ (e.g. boundary conditions, coefficients, or geometric shape) into our learned operator $\mathcal{S}_\theta(a)$ and obtain the steady-state output $u(x)$ directly. However, for *time-dependent* PDEs, there are two different strategies for forecasting the solution at a future time, using the learned one-step advancement operator $\mathcal{S}_\theta(t, \tau, a(t))$ which, as defined in Eq. (B.17), takes the current time $t$, a lead-time $\tau$, and the augmented input $a(t) = (c, u(t))$ to predict the solution at $t + \tau$. The two main inference strategies are *direct inference* and *autoregressive inference*.

Table B.2: Default architectural hyperparameters for GAOT.

| Abbreviation | Default Value | Description |
|---|---|---|
| **MAGNO (Encoder / Decoder)** | | |
| PJC | 256 | Dimensionality for MAGNO's internal hidden layers. |
| ENC-MLP | [64, 64, 64] | Hidden layers of the encoder MLP in MAGNO. |
| DEC-MLP | [64, 64] | Hidden layers of the decoder MLP in MAGNO. |
| LC | 32 | Output/Lifting channels after MAGNO (both encoder and decoder). |
| TS | I | Tokenization Strategy I. |
| NT | [64, 64] | Number of tokens for Strategy I (e.g., a $64 \times 64$ stencil grid). |
| GR | 0.033 | Aggregation radius (single-scale) for every token. If multiscale: $\{0.022, 0.033, 0.044\}$. |
| GeoEmb | statistical | Geometric Embedding for the encoder and decoder. |
| EM | 0.3 | Edge masking ratio for the MAGNO, used for 3D drivaernet++ dataset. |
| **Transformer** | | |
| PS | 2 | Default patch size for token grouping if Strategy I or III is used. |
| Norm | RMSNorm | Normalization used in attention and MLP layers. Pre-norm configuration. |
| PE | RoPE | Positional embedding used in the Transformer. Rotary positional embeddings. |
| RES-CON | True | Residual connections between transformer blocks. |
| TL | 5 | Number of Transformer blocks. |
| THS | 256 | Hidden dimension per self-attention block. |
| HEAD | 8 | Number of attention heads. |
| Dropout | 0.2 | Dropout ratio in the attention module. |
| FFN | 1024 | $4\times$ hidden size (THS) for the feedforward layer. |

**Direct Inference (DR)**    Recall from the main text that our learned operator $\mathcal{S}_\theta$ takes the lead-time $\tau$ as an explicit input, allowing for predictions over variable time steps. Given a snapshot of the solution $u(t_n)$ (which is part of $a(t_n)$), the network can directly predict the solution at any later time $t_n + \tau_{target}$, up to a maximum trained horizon $t_{\max}$, by evaluating $\mathcal{S}_\theta(t_n, \tau_{target}, a(t_n))$. Hence, for each possible time increment $\tau_{target} = k \cdot \Delta t$ (where $\Delta t$ is a base time step in the dataset, and $1 \leq k \leq k_{\max}$), we can produce the model's estimate $u(t_n + \tau_{target})$ from $u(t_n)$ in a single step, without iterating through intermediate time steps. Concretely, if our dataset is discretized at times $\Omega_t^\Delta = \{t_0, t_1, \ldots, t_N\}$, we can directly evaluate $\mathcal{S}_\theta(t_n, \tau_{target}, a(t_n))$ for various $\tau_{target}$ values originating from any $t_n \in \Omega_t^\Delta$. This provides a sequence of direct predictions at each possible time offset $\tau_{target}$ from any initial time $t_n$.

**Autoregressive Inference (AR)**    While direct inference estimates the solution at a single future time, an alternative is to iterate the operator in multiple, typically smaller, sub-steps to reach the final time. This approach is called *autoregressive* (AR) inference. Formally, given an initial snapshot $u(t_0)$, we repeatedly apply the learned operator $\mathcal{S}_\theta$ with a chosen fixed time increment for each step, $\Delta t_{AR}$, to advance the solution:

$$u(t_{k+1}) = \mathcal{S}_\theta(t_k, \Delta t_{AR}, a(t_k)) \tag{B.18}$$

where $t_{k+1} = t_k + \Delta t_{AR}$, and $a(t_k) = (c, u(t_k))$ uses the solution $u(t_k)$ from the previous step (or the initial condition if $k = 0$). This process is repeated until the desired final time $t_{\text{final}}$ is reached.

We examine two types of autoregressive step sizes in our experiments:

- **AR-2**: Use an autoregressive time increment of $\Delta t_{AR} = 2$ (assuming time units are consistent with the dataset). Starting from $u(t_0)$, we compute $u(t_0 + 2) = \mathcal{S}_\theta(t_0, 2, a(t_0))$. Then, using $u(t_0 + 2)$, we compute $u(t_0 + 4) = \mathcal{S}_\theta(t_0 + 2, 2, a(t_0 + 2))$, and so on, up to $t_{14}$. In total, we perform 7 consecutive evaluations of $\mathcal{S}_\theta$.

- **AR-4**: Use an autoregressive time increment of $\Delta t_{AR} = 4$. In this scenario, we predict $u(t_0 + 4) = \mathcal{S}_\theta(t_0, 4, a(t_0))$, then from $u(t_0 + 4)$, $u(t_0 + 8) = \mathcal{S}_\theta(t_0 + 4, 4, a(t_0 + 4))$, and so on, eventually reaching $t_{14}$ in just 4 iterations (assuming $t_0 = 0$ and $t_{final} = 16$ for this example, or if $t_{14}$ is the target after some steps).

Generally, the choice of the AR step size $\Delta t_{AR}$ is flexible. One could select any valid $\Delta t_{AR} \leq \tau_{\max}$ (where $\tau_{\max}$ is the maximum lead-time the model was reliably trained for in a single step) at each sub-step. Note that using fewer, larger time steps (e.g., $\Delta t_{AR} = 4$) can reduce computational cost but potentially compounds prediction errors more quickly if the operator $\mathcal{S}_\theta$ is less accurate for larger single-step lead-times. Conversely, smaller increments (e.g. $\Delta t_{AR} = 2$) tend to accumulate errors more gradually but require more iterations (and thus more computation) to reach the final time. Details can be found in [22, 36].

## C    Baselines

For the time-dependent benchmarks (including those on unstructured grids detailed in Table 1 and regular grids in Table E.5, the corresponding baseline results are primarily obtained from the work by [36]. These baseline models include RIGNO-12, RIGNO-18, CNO, scOT, FNO, GeoFNO, FNO DSE, and GINO. For further details on these methods, please refer to the paper [36]. In this work, we have additionally included three more recent models for a comprehensive comparison: Transolver [49], GNOT [20], and UPT [1]. Brief descriptions of these newly added models and the specific hyperparameters adopted in our experiments are provided below.

### C.1    UPT

*Universal Physics Transformers* (UPT) [1] form a neural-operator framework that fits into the canonical encode–process–decode pipeline:

$$\mathcal{U} = \mathcal{D} \circ \mathcal{A} \circ \mathcal{E}, \tag{C.1}$$

where $\mathcal{E}$ (Encoder) compresses $k$ input points—coming from an arbitrary Eulerian mesh or Lagrangian particle cloud—into a fixed set of $n_{\text{latent}}$ tokens. It first embeds the features and coordinates through a radius-graph message-passing layer that aggregates information into $n_s$ supernodes, and finally employs transformer and perceiver pooling blocks to obtain the latent representation $z_t \in \mathbb{R}^{n_{\text{latent}} \times h}$.

$\mathcal{A}$ (Approximator) is a stack of transformer blocks that advances the latent state in time, $\mathcal{A} : z_t \mapsto z_{t+\Delta t}$, enabling fast *latent roll-outs* without repeatedly decoding to the spatial domain.

$\mathcal{D}$ (Decoder) is a Perceiver-style cross-attention module that evaluates the latent field at any set of query positions $\{y_i\}_{i=1}^{k'}$, yielding $u_{t+\Delta t}(y_i) = \mathcal{D}(z_{t+\Delta t}, y_i)$ with $\mathcal{O}(n_{\text{latent}})$ complexity independent of $k'$.

In the setting of time-independent problems, we bypass the latent roll-out stage, and adopt a lightweight configuration in our experiments. Specifically, we use latent tokens $= 64$ and embedding dimensions $= 64$, which results in a model size of $0.74M$. A large variant with 256 latent tokens and 192 embedding dimension as setup in [1] was found to suffer from optimization difficulties and was not adopted in our baseline results. The same optimizer setup as GAOT is used here.

**Considered Hyperparameters**

| Architecture | |
|---|---|
| Trainable parameters | 0.74M |
| Number of supernodes $n_s$ | 2048 |
| Radius for message passing | 0.033 |
| Embedding (feature) channels | 64 |
| Encoder transformer blocks | 4 |
| Encoder attention heads | 4 |
| Latent tokens $n_{\text{latent}}$ | 64 |
| Latent dimension $h$ | 64 |
| Approximator transformer blocks | 4 |
| Approximator attention heads | 4 |
| Decoder attention heads | 4 |
| Training | |
| Optimizer | AdamW |
| Scheduler | same as in B.2 |
| Initial learning rate | $1 \cdot 10^{-3}$ |
| Weight decay | $10^{-5}$ |
| Number of epochs | 500 |
| Batch size | 64 |

## C.2 Transolver

*Transolver* [49] is a transformer-based operator learning model designed for PDEs on unstructured grids. It follows an encode-process-decode paradigm by stacking multiple Transolver blocks. The core of each block is the Physics-Attention mechanism.

Given input features $X_{\text{phys}} \in \mathbb{R}^{N \times C}$ for $N$ mesh points:

1. Encoding to Tokens: First, for each mesh point feature $x_i \in X_{\text{phys}}$, $M$ slice weights $w_i \in \mathbb{R}^{1 \times M}$ are learned, typically via a projection followed by a Softmax function: $w_i = \text{Softmax}(\text{Project}(x_i))$. These weights determine the assignment of mesh points to $M$ learnable "slices". The $j$-th physics-aware token $z_j \in \mathbb{R}^{1 \times C}$ is then encoded by a weighted aggregation of all mesh point features, using the slice weights:

$$z_j = \frac{\sum_{i=1}^{N} w_{i,j} x_i}{\sum_{i=1}^{N} w_{i,j}} \tag{C.2}$$

   This results in $M$ tokens $Z = \{z_j\}_{j=1}^{M} \in \mathbb{R}^{M \times C}$.

2. Token Processing: These $M$ tokens $Z$ are processed by a standard attention mechanism (e.g., multi-head self-attention) to capture correlations between different physical states represented by the tokens:

$$Z'_{\text{proc}} = \text{Attention}(Z) \tag{C.3}$$

   The processed tokens are $Z'_{\text{proc}} = \{z'_j\}_{j=1}^{M} \in \mathbb{R}^{M \times C}$.

3. Decoding to Physical Grid (Deslicing): The updated token features $Z'_{\text{proc}}$ are then broadcast back and recomposed onto the $N$ physical mesh points using the original slice weights $w$:

$$x'_i = \sum_{j=1}^{M} w_{i,j} z'_j \tag{C.4}$$

   This yields the output features for the Physics-Attention block, $X'_{\text{phys}} = \{x'_i\}_{i=1}^{N} \in \mathbb{R}^{N \times C}$.

A full Transolver layer typically incorporates this Physics-Attention mechanism within a standard Transformer layer structure, including Layer Normalization and Feed-Forward Networks.

While the Physics-Attention mechanism itself is designed to have a computational complexity linear with respect to the number of mesh points, it is important to note that the slicing (Eq. C.2) and deslicing (Eq. C.4) operations, which involve all $N$ points, are performed within each of the $L$ Transolver layers. This repeated mapping can lead to significant computational costs and memory overhead, especially for large $N$. This contrasts with architectures like GAOT and UPT, which perform the encoding to a latent space and decoding from it only once, with intermediate processing happening entirely in the latent token domain. In our experiments with time-independent partial differential equations, we followed the settings from the original Transolver paper [49];

**Considered Hyperparameters**

| Architecture | |
|---|---|
| Trainable parameters | 3.85 M |
| Hidden channels | 256 |
| Attention heads | 8 |
| Number of Layers | 8 |
| MLP ratio | 2 |
| number of slice | 32 |
| *Training* | |
| Optimizer | AdamW |
| Scheduler | same as in B.2 |
| Initial learning rate | $1 \cdot 10^{-3}$ |
| Weight decay | $10^{-5}$ |
| Number of epochs | 500 |
| Batch size | 20 |

### C.3  GNOT

*General Neural Operator Transformer* (GNOT) [20] is a Transformer-based framework designed for operator learning, particularly addressing challenges such as irregular meshes, multiple heterogeneous input functions, and multiscale problems. Its overall architecture can be represented as:

$$\mathcal{G} = \mathcal{F} \circ \underbrace{(\mathcal{B})^L}_{\text{processor}} \circ \mathcal{E}, \tag{C.5}$$

where $\mathcal{E}$ is the encoder, $\mathcal{B}$ represents a GNOT Transformer block (repeated $L$ times), and $\mathcal{F}$ is the output decoder.

1. Encoder ($\mathcal{E}$): The encoder maps diverse input sources (geometry, fields, parameters, edges) to embeddings using dedicated MLPs. This yields query embeddings $Q \in \mathbb{R}^{N_q \times d}$ for target points and a set of $m$ conditional embeddings $\{Y^{(\ell)} \in \mathbb{R}^{N_\ell \times d}\}_{\ell=1}^m$ from other input functions.

2. GNOT Transformer Block ($\mathcal{B}$): Each block refines query embeddings $Q$ using conditional embeddings $Y^{(\ell)}$ via:

   - Heterogeneous Normalized linear Cross-Attention (HNA): Fuses $Q$ with each $Y^{(\ell)}$ using separate MLPs for keys/values from different $Y^{(\ell)}$, followed by normalization and averaging.

$$Q'_{\text{cross}} = Q + \frac{1}{L_c} \sum_{\ell=1}^{L_c} \text{NormLinearCrossAttn}(Q, Y^{(\ell)}) \tag{C.6}$$

   - Normalized Self-Attention: Applies normalized linear self-attention to $Q'_{\text{cross}}$ for further refinement.

$$Q'_{\text{self}} = \text{NormLinearSelfAttn}(Q'_{\text{cross}}) \tag{C.7}$$

- Geometric Gating FFN: A Mixture-of-Experts (MoE) FFN where expert FFNs ($E_k$) are weighted by $p_k(x_{\text{coord}})$. These weights are predicted by a gating network $G(\cdot)$ using query point coordinates $x_{\text{coord}}$, enabling soft domain decomposition for multiscale problems.

$$\text{FFN}_{\text{Gated}}(X) = \sum_{k=1}^{K} p_k(x_{\text{coord}}) \cdot E_k(X), \quad p_k(x_{\text{coord}}) = \text{Softmax}(G_k(x_{\text{coord}})) \quad \text{(C.8)}$$

These components, with Layer Normalization and residual connections, form the block.

3. Decoder ($\mathcal{F}$): After $L$ blocks, a final decoder (typically an MLP) maps processed query features to the output solution.

**Considered Hyperparameters**

| Architecture | |
|---|---|
| Trainable parameters | 4.87 M |
| Hidden channels | 128 |
| Attention heads | 8 |
| Number of Layers | 8 |
| MLP ratio | 2 |
| Training | |
| Optimizer | AdamW |
| Scheduler | same as in B.2 |
| Initial learning rate | $1 \cdot 10^{-3}$ |
| Weight decay | $10^{-5}$ |
| Number of epochs | 500 |
| Batch size | 20 |

## D   Datasets

In this work, we test GAOT on 28 benchmarks for both time-independent and time-dependent PDEs of various types, ranging from regular grids to random point clouds to highly unstructured adapted grids. The time-dependent and Poisson-Gauss datasets are sourced from [22] and [36], respectively. The static elasticity dataset is from [26]. The DrivAerNet++ and DrivAerML datasets are taken from [14] and [3], respectively, where the objective is to predict the surface pressure and wall shear stress. The NASA-CRM dataset is obtained from [7], where we similarly predict the pressure field and surface friction coefficient on the NASA Common Research Model (CRM) surface. For 3D car and wing aerodynamics, the full-resolution surface field is directly predicted without any downsampling. Furthermore, we have also generated five additional challenging datasets: a Poisson-C-Sines dataset exhibiting multiscale properties, and four datasets for compressible fluid dynamics with highly unstructured adapted grids. Detailed information regarding these datasets can be found in the Tab D.1.

We focus on the following PDE Types simulated for various initial/boundary conditions and domain geometries:

**Hyper-Elastic Equation (HEE)**   :

$$\rho^s \frac{\partial^2 \mathbf{u}}{\partial t^2} + \nabla \cdot \boldsymbol{\sigma} = 0, \tag{D.1}$$

where $\rho^s$ is the mass density, $\mathbf{u}$ is the displacement vector, and $\boldsymbol{\sigma}$ is the stress tensor. A constitutive model links the strain tensor $\boldsymbol{\epsilon}$ to the stress tensor. The material is the incompressible Rivlin-Saunders type, characterized by $\boldsymbol{\sigma} = \frac{\partial w(\boldsymbol{\epsilon})}{\partial \boldsymbol{\epsilon}}$ with $w(\boldsymbol{\epsilon}) = C_1(I_1 - 3) + C_2(I_2 - 3)$.

Table D.1: Overview of the datasets used in this work. Datasets listed above the line are time-independent, while those below are time-dependent. Geometry variation (GeoVar) describes whether every data sample in the dataset has a different geometry. Characteristics briefly describe each dataset's geometry or PDE setup. The PDE Type column indicates the corresponding class. Visualization (Vis.) provides references to visual examples; for time-dependent datasets, this may include visualizations for both unstructured partially ones and original regular grid ones. Datasets marked with * are newly proposed in this work.

| Abbreviation | GeoVar | Characteristic | PDE Type | Vis. |
|---|---|---|---|---|
| Poisson-C-Sines* | F | Circular domain with sines $f$ | PE | G.1 |
| Poisson-Gauss | F | Gaussian source | PE | G.2 |
| Elasticity | T | Hole boundary distance | HEE | G.3 |
| NACA0012* | T | Flow past NACA0012 airfoil | CE | G.4 |
| NACA2412* | T | Flow past NACA2412 airfoil | CE | G.5 |
| RAE2822* | T | Flow past RAE2822 airfoil | CE | G.6 |
| Bluff-Body* | T | Flow past bluff-bodies | CE | G.7 |
| DrivAerNet++(p) | T | Surface pressure | INS | G.8 |
| DrivAerNet++(wss) | T | Surface wall shear stress | INS | G.9 |
| DrivAerML(p) | T | Surface pressure coefficient | INS | G.10 |
| DrivAerML(wss) | T | Surface wall shear stress | INS | G.11 |
| NASA-CRM (p) | T | Surface pressure | INS | G.12 |
| NASA-CRM (sfc) | T | Surface friction coefficient | INS | G.13 |
| NS-Gauss | F | Gaussian vorticity IC | INS | G.14, G.22 |
| NS-PwC | F | Piecewise const. IC | INS | G.15, G.23 |
| NS-SL | F | Shear layer IC | INS | G.16, G.24 |
| NS-SVS | F | Sinusoidal vortex sheet IC | INS | G.17, G.25 |
| CE-Gauss | F | Gaussian vorticity IC | CE | G.18, G.26 |
| CE-RP | F | 4-quadrant RP | CE | G.19, G.27 |
| Wave-Layer | F | Layered wave medium | WE | G.20, G.28 |
| Wave-C-Sines | F | Circular domain with sines IC | WE | G.21 |

**Poisson Equation (PE)** :

$$-\Delta u = f, \quad \text{in } (0,1)^2, \tag{D.2}$$

with homogeneous Dirichlet boundary conditions. The dataset related to the Poisson equation uses either sinusoidal or Gaussian-like source terms on square or circular domains (see Table D.1).

**Incompressible Navier–Stokes (INS)** :

$$\nabla \cdot \mathbf{v} = 0,$$
$$\partial_t \mathbf{v} + (\mathbf{v} \cdot \nabla)\mathbf{v} = -\nabla p + \nu \nabla^2 \mathbf{v},$$

where $\mathbf{v}$ is the velocity field, $p$ is the pressure, and viscosity is $\nu$. It assumes periodic boundary conditions and samples various initial conditions (e.g., Gaussian, piecewise-constant, sinusoidal vortex sheets).

**Compressible Euler (CE)** :

$$\partial_t \mathbf{u} + \nabla \cdot \mathbf{F} = 0, \qquad \mathbf{u} = (\rho, \rho\mathbf{v}, E)^\top, \quad E = \tfrac{1}{2}\rho\|\mathbf{v}\|^2 + \tfrac{p}{\gamma-1}, \tag{D.3}$$

with $\gamma = 1.4$. Showing in [22], it imposes periodic boundary conditions and ignore gravity effects. Data are generated using random initial/boundary conditions such as Gaussian or Riemann problem (RP) setups.

**Wave Equation (WE)** :

$$\partial_{tt} u \; - \; c^2(x, y) \, \nabla^2 u \; = \; 0, \tag{D.4}$$

with a spatially varying propagation speed $c(x, y)$ in an inhomogeneous medium. The dataset employs absorbing or homogeneous Dirichlet boundaries. Initial conditions (e.g., sinusoidal or layered) are drawn from parameterized distributions.

All time-dependent problems are numerically integrated up to $T = 1$ (except for the Wave-C-Sines where T=0.005), collected (up to) $N_t = 21$ uniform snapshots per sample at $t \in \{0, 2, 4, 6, 8, 10, 12, 14\}$. For time-dependent PDE, as mentioned before, we use the same all2all training strategy proposed in Poseidon [22]. This means that each trajectory can generate 28 pairs for training.

## D.1 Poisson-C-Sines

This dataset contains solutions to the two-dimensional Poisson equation with a circular domain. The Poisson equation is a fundamental linear elliptic partial differential equation (PDE) given by Eq. D.2. Unlike standard benchmarks that map point-wise evaluations of the source term to the solution, this dataset defines the operator learning task within the context of the finite element discretization. Specifically, we aim to learn the mapping from the *projection coefficients* of the source term $f$ onto the finite element basis functions to the nodal solution values $u$.

The continuous source term is constructed via a spectral expansion:

$$f(x, y) = \frac{\pi}{K^2} \sum_{i,j=1}^{K} a_{ij} \cdot (i^2 + j^2)^{-r} \sin(\pi i x) \sin(\pi j y), \quad \forall (x, y) \in D, \tag{D.5}$$

where $r = -0.5$ and $K = 16$. The coefficients $a_{ij}$ are sampled i.i.d. uniformly from $[-1, 1]$. To generate the input data, this source term is projected onto the finite element test space spanned by the nodal basis functions $\{\phi_k\}$. The input to the model is the resulting load vector $\mathbf{b}$, where each entry corresponds to the integral $b_k = \int_{\Omega} f(x, y) \phi_k(x, y) d\Omega$. Consequently, the operator learning task is defined as $\mathcal{G}^{\dagger} : \mathbf{b} \mapsto \mathbf{u}$, representing the inverse of the discretized stiffness operator. The mesh is generated using the Delaunay algorithm with 16,431 points and 32,441 elements, and zero Dirichlet boundary conditions are enforced during the assembly.

## D.2 Compressible Flow Past Airfoils & Bluff Bodies

A classical benchmark for compressible flow physics used for testing the accuracy of neural operators and PDE foundation models is the case of flow past airfoils [22, 26]. The datasets used in these papers are limited to transonic flow past perturbations of a single airfoil. To capture a broader range of rich flow phenomena, it is essential to explore the parameter space spanned by the Mach number $Ma$, angle of attack $\alpha$ and the shape function. To address this issue, this new dataset introduces samples comprising a range of flow phenomena from subsonic to supersonic flow for varying angles of attack across classical airfoils and various bluff body geometries. The steady-state compressible Euler equations govern the flow phenomena in this dataset. The equations have been solved using the finite-volume EULER solver of the open-source software SU2 [13] on an unstructured grid generated by Delaundo [37]. Convective flux discretization is done using the Jameson-Schmidt-Turkel (JST) scheme that is designed especially for achieving quick convergence to steady-state solutions of the compressible Euler equations. Figure D.1 represents an O-type unstructured mesh generated using Delaundo for the RAE2822 airfoil. Similar O-type unstructured meshes have been generated for all airfoils and bluff-bodies considered. The free-stream pressure and temperature conditions for all simulations in this dataset are $p_\infty = 1$ atm and $T_\infty = 288.15$ K.

### D.2.1 Airfoils

Flow past airfoils is considered for $0.5 \leq Ma \leq 1.4, 0.5° \leq \alpha \leq 5.0°$ and for 500 unique perturbations applied to shape functions of the NACA2412, NACA0012, and RAE2822 airfoils. Anisotropic adaptive mesh refinement for highly accurate shock resolution (oblique and bow shocks) is performed using INRIA's pyAMG library coupled with SU2 [31] . The anisotropic mesh refinement is done using a Mach sensor that generates refined meshes based on the simulation on a coarse grid such as in Figure D.1. The final simulations are then performed by the SU2 EULER solver on the

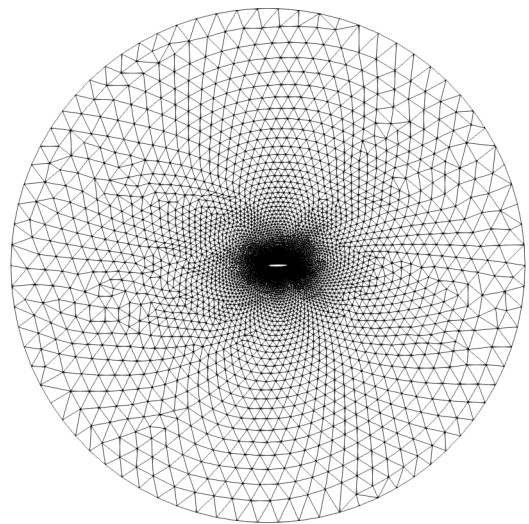

Figure D.1: O-type unstructured mesh - RAE2822 airfoil

new refined mesh. Figure D.2 represents the highly unstructured adapted grids for transonic and supersonic flow past the RAE2822 airfoil.

We consider the reference airfoil shapes with the upper and lower surface coordinates located at $(x, y_{\text{ref}}^{\text{U}}(\xi))$ and $(x, y_{\text{ref}}^{\text{L}}(\xi))$ where $\xi = \frac{x}{c}$, $c$ is the chord length. We use the Class Function/Shape Function Transformation (CST) Method [25] for parameterizing the airfoil surfaces in terms of a class function $C(\xi)$ and shape functions $S^{\text{U}}(\xi), S^{\text{L}}(\xi)$ using an in-house MATLAB code. The airfoil upper surface function $\eta^{\text{U}}(\xi)$ and lower surface function $\eta^{\text{L}}(\xi)$ are parametrized as follows:

$$\eta^{\text{U}}(\xi) = C(\xi)S^{\text{U}}(\xi), \quad \eta^{\text{L}}(\xi) = C(\xi)S^{\text{L}}(\xi) \tag{D.6}$$

where the class function for airfoils is given as:

$$C(\xi) = \sqrt{\xi}(1 - \xi) \tag{D.7}$$

and the upper and lower surface shape functions are respectively

$$S^{\text{U}}(\xi) = \sum_{i=0}^{n} A_i \frac{n!}{i!(n-i)!} \xi^i (1-\xi)^{n-i}, \quad S^{\text{L}}(\xi) = \sum_{i=0}^{n} B_i \frac{n!}{i!(n-i)!} \xi^i (1-\xi)^{n-i} \tag{D.8}$$

The polynomials $S_{i,n} = \frac{n!}{i!(n-i)!} \xi^i (1-\xi)^{n-i}$ associated with the coefficients $(A_i, B_i)$ are Bernstein polynomials and $n = 7$ is chosen. The parameters $(A_i, B_i)$ directly influence key airfoil design variables such as the leading edge radius, trailing edge boattail angle, maximum airfoil thickness and

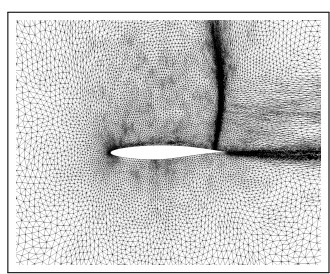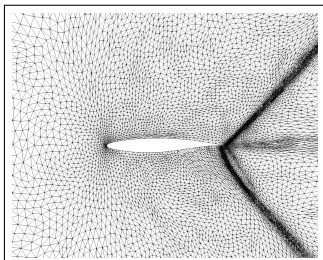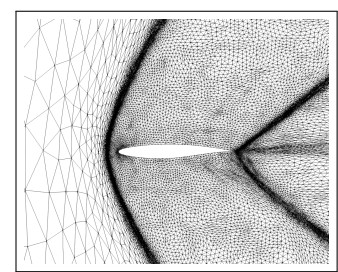

Figure D.2: Adaptively refined meshes for flow past the RAE2822 airfoil at $\alpha = 2.0$ and at different $Ma = 0.8$ (left), 1.0 (center), 1.4 (right).

maximum thickness location. The parameters $(A_0, B_0)$ are linked to the leading edge radius $R_{\text{LE}}$ as follows,

$$A_0 = -B_0 = \sqrt{2r}, \quad r = \frac{R_{\text{LE}}}{c} \tag{D.9}$$

and the parameters $(A_n, B_n)$ are linked to the upper boattail angle $\beta_{\text{U}}$ and lower boattail angle $\beta_{\text{L}}$ :

$$A_n = \tan(\beta_{\text{U}}), \quad B_n = \tan(\beta_{\text{L}}) \tag{D.10}$$

To generate perturbed variations of the airfoils, minor random perturbations are made to the CST parameters $(A_i, B_i)$ keeping in mind the constraints $A_0 = -B_0$, $A_i > B_i$ and $\beta_{\text{U}} > \beta_{\text{L}}$. We have randomly sampled 5384 solutions from our dataset for each classical airfoil shape with a train/validation/test split of 5000/128/256. For each data, we sub-samples 8000 points for training, validation and testing.

### D.2.2 Bluff-Body

Flow past bluff-bodies is considered at $0.3 \leq Ma \leq 1.3, 0.5° \leq \alpha \leq 15.0°$ for a wide variety of simple bluff-body geometries. Steady-state solutions for the compressible Euler equation for flow past bluff-bodies may not exist or are often unstable making it difficult to attain convergence. This bluff-body aerodynamics dataset comprises of samples that are at pseudo-steady state in a large finite-time limit. Figure E.6a describes all the bluff body geometries taken into consideration in this dataset.

We sample 4384 solutions from our dataset with a train/validation/test split of 4000/128/256 for all bluff-body geometries used for "Training and Testing" . For each data, we sub-sample 14000 points for training, validation and testing.

## E   Additional Results

### E.1   Asymptotic Complexity

Table E.1 summarizes the asymptotic complexities of the compared models in Fig. 1. In practice, the number of edges $E$ is typically $3-5\times$ the number of nodes $N$. For consistency, we denote the number of latent tokens by $T$ across all models; however, ViT-style patchifying in GAOT effectively reduces $T$ and thereby improves scalability. Furthermore, different models employ distinct hidden dimensions $C$ for their edge update functions, which introduces further variation in the total computational cost.

Table E.1: Asymptotic complexity. $N$: number of nodes, $E$: number of edges, $T$: latent tokens, $C$: hidden channels, $p$: patch size, $L$: number of layers/blocks.

| Model | Total Complexity |
|---|---|
| GAOT | $O(E + L(T/p)^2 C)$ |
| GINO | $O(E + LT \log T)C$ |
| RIGNO | $O(LEC)$ |
| Transolver | $O\big(L(NTC + T^2C)\big)$ |

The main computational burden in graph-based models arises from message passing, and memory usage is dominated by storing edge activations, scaling linearly with both $E$ and $C$. GAOT, RIGNO, and GINO compress information into latent tokens and then perform $L$ layers of latent processing. In contrast, Transolver performs compression and decoding within each block, operating directly on nodes of $N$ rather than edge $E$, but accumulating higher total computation across layers.

It is worth noting that these are theoretical costs only. On modern hardware, actual runtime can deviate significantly because graph-based operations rely on sparse and irregular memory access patterns, which are inefficient on GPUs. By contrast, transformer-style dense computations are highly optimized and throughput-efficient on current accelerators. The following Sec E.2 provides a more practical profiling comparison.

### E.2   Runtime Profiling

To further analyze the computational efficiency of our implementation, we compare the detailed runtime breakdowns of GINO [29] and GAOT on the NACA0012 dataset using a single NVIDIA

GeForce RTX 4090 GPU. Table E.2 presents the per-component execution times for both models at a batch size of 1.

Table E.2: Comparison of time breakdown between GINO (6.07M parameters) and GAOT (5.62M parameters) with batch size = 1 on the NACA0012 dataset.

| Component | GINO | | GAOT | |
| --- | --- | --- | --- | --- |
| | Sub-Operation | Time (ms) | Sub-Operation | Time (ms) |
| Encoder | Graph building | 1.984 | AGNO | 2.296 |
| | GNO | 2.793 | GeoEmb | 3.412 |
| | **Total Encoder** | **4.777** | **Total Encoder** | **5.708** |
| Processor | Per-layer FNO | 1.178 | Per-layer Transformer (ps=2) | 2.02 |
| | $1.178 \times 5$ layers | **5.89** | $2.02 \times 5$ layers | **10.1** |
| Decoder | Graph building | 1.984 | AGNO | 1.356 |
| | GNO | 1.140 | GeoEmb | 1.865 |
| | **Total Decoder** | **3.124** | **Total Decoder** | **3.221** |
| **Total Time** | 13.791 | | 19.029 | |

For GINO, the total runtime per sample is 13.79 ms, where the encoder and decoder each spend almost 2 ms on graph building, comparable to or even exceeding the actual GNO operations. Also, the official implementation of GINO does not support batch processing when the underlying graphs change within a batch, as is typical for PDEs defined on arbitrary domains. This limitation arises because batch-processing the encoder and decoder often leads to out-of-memory (OOM) errors, making GINO scale poorly with batch size.

In contrast, GAOT eliminates the runtime overhead of graph building by precomputing graphs and loading them efficiently from cache. More importantly, to overcome the memory bottleneck in batch processing, we adopt a hybrid execution strategy: the encoder and decoder are processed sequentially through a lightweight for-loop, which avoids memory overflow without introducing noticeable computational overhead, while the processor (Transformer) is fully batch-processed to maximize throughput. This design choice is based on the observation that, for larger models, the processor typically dominates the compute cost, making it the optimal module to parallelize. As shown in Table E.3, GAOT achieves a total runtime of 263.1 ms for batch size 32, while GINO, lacking batching support, must loop 32 times, resulting in $13.791 \times 32 = 441.3$ ms. The Transformer processor scales efficiently (from 10.1 ms for batch size 1 to only 30.1 ms for batch size 32), while maintaining a manageable total memory usage of 7.74 GB. Consequently, GAOT achieves significantly better scalability than GINO in both input size and model size, without sacrificing computational efficiency. For very large graphs, e.g. DrivAerML dataset [3], edge masking is used to control memory usage ensuring stable training even on limited-memory GPUs.

Table E.3: Breakdown of time and memory for GAOT (5.62M parameters) with batch size = 32 on the NACA0012 dataset.

| Component | Sub-Operation | Time (ms) | Memory (GB) |
| --- | --- | --- | --- |
| Encoder | AGNO + GeoEmb | 126 | 1.836 |
| Processor | 5-layer Transformer, ps=2 | 30.1 | 3.592 |
| Decoder | AGNO + GeoEmb | 107 | 2.310 |
| **Total** | | **263.1** | **7.738** |

## E.3 Accuracy, Robustness and Computational Efficiency Metrics

**Accuracy** For benchmarks in Table 1 and Table E.5, we adopt the relative $L^1$ error metric, following the manner of CNO [42], to measure the discrepancy between the ground-truth operator output $\mathcal{S}(a)$ and the model's prediction $\mathcal{S}_\theta(a)$ over a discrete set of points. Suppose a given sample is discretized into $N$ points (either on a regular grid or an unstructured mesh). For a single-component solution

field, the discretized relative $L^1$ error $\varepsilon$ is defined as

$$\varepsilon = \frac{1}{N} \sum_{i=1}^{N} \frac{\left| \left(\mathcal{S}(a)\right)_i - \left(\mathcal{S}_\theta(a)\right)_i \right|}{\left| \left(\mathcal{S}(a)\right)_i \right|}. \tag{E.1}$$

Because the test set contains multiple input–output pairs $\{(a, \mathcal{S}(a))\}$, we obtain a distribution of errors. We report the median of these errors—rather than the mean—to mitigate the influence of strong outliers. For multi-component PDE solutions (e.g., velocity and pressure fields), we compute the median error per component, then average these medians to obtain a single scalar metric. In time-dependent tasks, we specifically report the relative $L^1$ error at the final time snapshot, as errors usually accumulate over time and thus the last snapshot often poses the greatest challenge.

For the pressure and wall shear stress (WSS) in the DrivAerNet++ dataset, we evaluated the model on 1154 samples according to the official leaderboard. The errors are calculated based on normalized pressure and WSS. For pressure, the mean and standard deviation (std) for normalization were obtained from the open-source code of [14]. However, for WSS, as the normalization statistics were not open-sourced, we calculated the mean and variance for the x, y, and z components over 8000 samples to be used for normalization. The Mean Squared Error (MSE) and Mean Absolute Error (MAE) are first computed for each individual sample. Then, the average of these errors across the 1154 test samples is reported as the final result. This entire procedure strictly follows their open-source code methodology of [14].

In Table E.4, we further provide an aggregate performance comparison of GAOT and three representative baseline models (Transolver, GINO, RIGNO-18) on both time-dependent and time-independent dataset categories, described in Table 1. Specifically, for each individual dataset, we calculate the normalized scores for every model. The best-performing model is assigned a score of 1. The scores for other models are calculated as the ratio of the best model's error to their respective errors:

$$S_{\text{norm}} = \frac{\text{error}_{\text{best}}}{\text{error}_{\text{model}}} \tag{E.2}$$

These dataset scores are then summed for each model to derive total scores for the time-dependent and time-independent dataset categories, respectively, offering a complete view of model performance.

**Robustness**    To evaluate the consistency of model performance across different datasets, we introduce a Robustness Score. This score is calculated for both the time-dependent and time-independent categories of datasets. Leveraging the normalized scores obtained by each model on the individual datasets within these categories, the Robustness Score for a model is defined as:

$$\text{Robustness Score} = \bar{S}_{\text{norm}} \times (1 - \text{CV}), \tag{E.3}$$

where $\bar{S}_{\text{norm}}$ is the mean of the model's normalized scores across all datasets in a specific category (either time-dependent or time-independent). The term CV represents the Coefficient of Variation of these normalized scores, calculated as:

$$\text{CV} = \frac{\sigma_{S_{\text{norm}}}}{\bar{S}_{\text{norm}}}, \tag{E.4}$$

where $\sigma_{S_{\text{norm}}}$ is the standard deviation of the model's normalized scores within that same category. A higher Robustness Score suggests that a model not only achieves high average performance (high mean normalized score) but also exhibits less variability in its performance across the different datasets within the category (low CV), indicating greater reliability. The robustness scores for GAOT, Transolver, GINO and RIGNO-18 are shown in Table E.4.

**Computational Efficiency**    To provide a comprehensive characterization of model performance and analyze the inherent accuracy-efficiency trade-off, we further evaluate the computational efficiency of the models during both training and inference phases.

Training efficiency is quantified by the *training throughput*, defined as the number of samples the model can process per second during training, encompassing the forward pass, backward pass, and gradient update. A high training throughput is indicative of a model's ability to learn quickly from data. This is essential for handling large-scale datasets or developing large foundation models where training time can be a significant bottleneck. For measuring throughput, the batch size for each

model was determined first by identifying the maximum value that could be run without encountering Out-of-Memory (OOM) errors on the target hardware. The actual batch size used for the throughput measurement was then set to approximately half of this maximum. This heuristic is based on the observation that peak throughput is often not achieved at the absolute maximum batch size, but rather at a point (frequently around half the maximum) where GPU resources, such as shared memory bandwidth, are optimally utilized, leading to the highest processing rates.

Inference efficiency is measured by the *inference latency*, which is the time taken for the model to perform a single forward pass on an individual sample (i.e., batch size of 1). Low inference latency is a critical attribute for the practical deployment of models, particularly in applications requiring real-time or near real-time predictions, such as in engineering simulations, interactive design tools, or control systems.

All computational efficiency metrics were benchmarked on the Bluff-Body dataset with one NVIDIA-4090 hardware. To ensure reliable and stable measurements, the GPU was warmed up prior to data collection, and each reported metric is the average of 100 repeated measurements.

### E.4 Results for Radar Chart in Main Text.

Table E.4 presents the raw data used to generate the radar chart in Figure 1 of the main text. The metrics depicted in the radar chart include:

- **Accuracy (Acc. and Acc.(t)):** Overall accuracy on time-independent (Acc.) and time-dependent (Acc.(t)) datasets.
- **Robustness (Robust. and Robust.(t)):** Robustness on time-independent (Robust.) and time-dependent (Robust.(t)) datasets.
- **Training Throughput (Tput.(train)):** The number of samples processed per second during training.
- **Inference Latency (Infer. Latency):** The time (ms) taken for a single forward pass on one sample during inference.

The precise definitions and calculation methodologies for these metrics are detailed in Sec. E.3

Table E.4: Data for Radar Chart

| Model | acc.(t) . | acc. | Tput(train) | Infer Latency | Peak memory | InputScal. | ModelScal | robust(t) | robust |
|-------|-----------|------|-------------|---------------|-------------|------------|-----------|-----------|--------|
| GAOT | 7.45 | 6.30 | 97.5 | 6.966 | 101.7 | 68.12 | 48.7 | 0.80 | 0.77 |
| Transolver | 0 | 4.12 | 39.5 | 15.295 | 144.0 | 8.96 | 6.69 | 0 | 0.22 |
| GINO | 2.77 | 2.94 | 60.4 | 8.455 | 556.8 | 30.53 | 40.00 | 0.15 | 0.19 |
| RIGNO-18 | 7.37 | 4.38 | 50.3 | 12.749 | 188.8 | 12.52 | 7.51 | 0.85 | 0.29 |

Table E.4 also includes **Peak Memory (MB)**, which records the peak GPU memory consumption of each model during inference with a batch size of 1. Although not visualized in the radar chart (Figure 1), the data indicates that GAOT exhibits the lowest peak memory usage among the compared models. Furthermore, Figure E.1 (a, c) illustrates the scaling of peak memory with increasing input grid size and model size, respectively. These plots demonstrate GAOT's superior memory utilization capabilities.

The **Input Scalability** and **Model Scalability** scores presented in the radar chart are derived from the training throughput measured under specific conditions:

- Input Scalability is based on throughput at an input grid size of 50,000 points.
- Model Scalability is based on throughput for a model size of approximately 70 million parameters.

These particular evaluation points were chosen due to the performance limitations encountered with models like RIGNO-18 and Transolver on a single NVIDIA 4090 GPU, which prevented us from benchmarking them at larger scales. It is important to note that GAOT's architecture allows it to scale significantly beyond these tested limits.

SM Figure E.1 provides detailed scaling curves for both peak memory and training throughput as functions of input grid size and model size. To vary the model size for these comparisons, we systematically adjusted key architectural width parameters for each model:

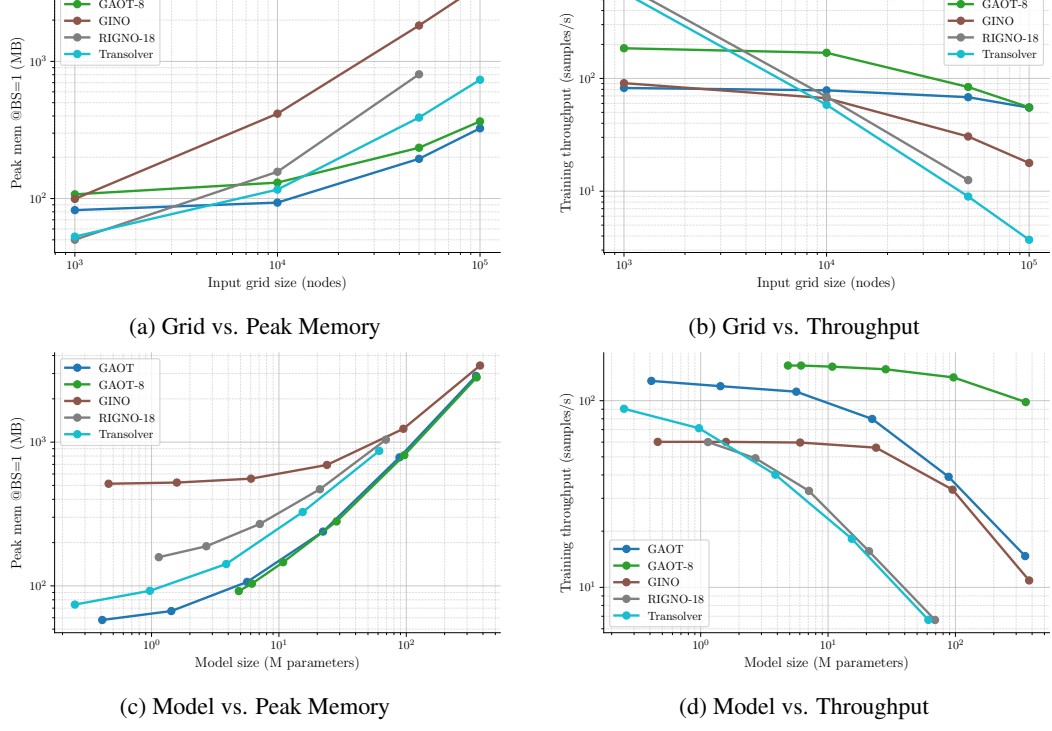

Figure E.1: Performance scaling comparisons across different metrics.

- For Transolver, we scaled its hidden channel dimension through [64, 128, 256, 512, 1024].

- For RIGNO, the hidden channel dimensions of its node and edge functions were varied across [64, 128, 256, 512, 1024].

- For GINO, the hidden channel dimension of its FNO processor layers was selected from [16, 32, 64, 128, 256, 512].

- For our GAOT model, we scaled the hidden channel dimension of its attention layers using values from [64, 128, 256, 512, 1024, 2048], while the hidden dimension of its FFN layers was maintained at four times the attention layer's hidden dimension.

This figure also introduces results for GAOT-8, a variant of GAOT where the patch size in the transformer processor is set to 8 (the default GAOT employs a patch size of 2). As shown, GAOT-8 can achieve enhanced computational performance. Furthermore, as detailed in our ablation studies (Section F.2), this improvement in efficiency with GAOT-8 does not give rise to the substantial accuracy degradation.

## E.5 Regular Grid Dataset

In addition to datasets with arbitrary point cloud geometries, we also evaluated the performance of our GAOT model on time-dependent PDE datasets where the inputs are provided on regular (structured) grids. The results for GAOT are compared against several baselines, including RIGNO (RIGNO-18 and RIGNO-12), CNO, scOT, and FNO. The performance data for these baseline models are sourced from the original RIGNO paper [36].

As demonstrated in Table E.5, GAOT also performs well on these structured grid datasets. Our model consistently ranks within the top two across six of the seven benchmark datasets, achieving the leading (first place) performance on five of them. This highlights GAOT's robustness and strong generalization capabilities across different input discretizations.

Furthermore, we ablated GAOT further by just running the ViT processor (without the encoder/decoder) on these Cartesian datasets. From these results, it is clear that in some cases, ViT is comparable to or slightly better than GAOT, indicating that the power of GAOT in these cases stemmed from a good processor. However in many more cases, GAOT is significantly superior to ViT also showing that the encoder/decoder contribute significantly to expressivity and together, this combination achieves SOTA performance. This interesting experiment clearly delineates the relative contributions of encoder/decoder vs. processor.

Table E.5: Benchmarks with time-dependent datasets with regular grid inputs. Best and 2nd best models are shown in blue and orange fonts for each dataset.

| Dataset | Median relative $L^1$ error [%] | | | | | | |
|---|---|---|---|---|---|---|---|
| Structured | GAOT | RIGNO-18 | RIGNO-12 | CNO | ViT | scOT | FNO |
| NS-Gauss | 2.29 | 2.74 | 3.78 | 10.9 | 3.16 | 2.92 | 14.41 |
| NS-PwC | 1.23 | 1.12 | 1.82 | 5.03 | 3.89 | 7.12 | 12.55 |
| NS-SL | 0.98 | 1.13 | 1.82 | 2.12 | 0.73 | 2.49 | 2.08 |
| NS-SVS | 0.46 | 0.56 | 0.75 | 0.70 | 0.39 | 1.01 | 7.52 |
| CE-Gauss | 5.28 | 5.47 | 7.56 | 22.0 | 6.81 | 9.44 | 28.69 |
| CE-RP | 4.98 | 3.49 | 4.43 | 18.4 | 4.30 | 9.74 | 38.48 |
| Wave-Layer | 5.40 | 6.75 | 8.97 | 8.28 | 5.48 | 13.44 | 28.13 |

## E.6 Model and Dataset Scaling

**Model Size**   To further investigate the scalability of our approach, we conduct an ablation study on how different model sizes affect performance. We focus on the two compressible Euler datasets, CE-Gauss and CE-RP, each with $1,024$ training trajectories. We measure the final-time relative $L^1$ error ($t = t_{14}$) and record the total number of parameters and per-epoch training time under various hyperparameter configurations.

We vary the following components of our GAOT architecture as explained in the Tab B.2:

- LC (Lifting Channels): The number of channels used during the encoder stage to project from the unstructured node features to latent tokens. Intuitively, a larger LC can preserve more local features when mapping from the input domain to the latent space.

- TL (Transformer Layers): The depth of the transformer-based processor. Increasing TL typically increases modeling capacity for global interactions.

- THS (Transformer Hidden Size): The hidden dimension of each self-attention block. A larger THS can capture richer representations.

- FFN (Feed-Forward Network Size): The hidden dimension inside the FFN sub-layer, which we set to $4 \times$ THS following standard vision transformer practice.

Table E.6 summarizes the performance across a range of these hyperparameters. We also record the total number of trainable parameters (in millions) and the approximate epoch time (in seconds) on one NVIDIA 4090 GPU with a batch size of 64. Here, all experiments are done with patch size equal to 2.

From the top block of Table E.6 (rows 1–4), we observe that as we increase THS from 32 to 256 (keeping TL=5 and LC=32), the final-time errors on both CE-Gauss and CE-RP decrease significantly. For example, on CE-Gauss, the error drops from $48.4\%$ down to $6.88\%$. This trend reflects the transformer's ability to scale with hidden dimension. The training time per epoch grows from roughly $84$ seconds to $143$ seconds. While performance improves, larger THS demands more computational resources.

Next, we fix (TL,THS)=(5,256) and vary LC in the middle block (rows 5–7). Setting LC=32 consistently achieves strong results. Lowering LC to 16 slightly degrades performance, while pushing LC to 64 or 128 yields only marginal gains. Hence, LC=32 appears sufficient to capture the encoder-level geometry information.

Finally, the bottom block (rows 8–10) examines the effect of transformer layers TL from 1 up to 10. With TL=1, errors remain quite high (25.0% on CE-Gauss); adding layers substantially reduces error

Table E.6: Relative $L^1$ test errors at $t = t_{14}$ with different architectural hyperparameters. Time refers to training time with batch size equals to 64 on 1 NVIDIA-4090 GPU, and the patch size is set to 2. The size of training trajectories is 1024.

| Model size | | Hyperparameters | | | | Median relative $L^1$ error [%] | |
| --- | --- | --- | --- | --- | --- | --- | --- |
| Parameters [M] | Time [s] | LC | TL | THS | FFN | CE-Gauss | CE-RP |
| 0.14 | 84 | 32 | 5 | 32 | 128 | 48.4 | 26.5 |
| 0.41 | 89 | 32 | 5 | 64 | 256 | 13.2 | 12.0 |
| 1.42 | 100 | 32 | 5 | 128 | 512 | 9.17 | 7.90 |
| 5.6 | 143 | 32 | 5 | 256 | 1024 | 6.88 | 5.28 |
| 5.5 | 142 | 16 | 5 | 256 | 1024 | 7.97 | 5.94 |
| 5.6 | 154 | 64 | 5 | 256 | 1024 | 6.94 | 5.18 |
| 6.1 | 181 | 128 | 5 | 256 | 1024 | 7.33 | 5.20 |
| 1.16 | 50 | 32 | 1 | 256 | 1024 | 25.0 | 14.5 |
| 3.39 | 98 | 32 | 3 | 256 | 1024 | 9.00 | 6.80 |
| 11.2 | 260 | 32 | 10 | 256 | 1024 | 5.28 | 5.35 |

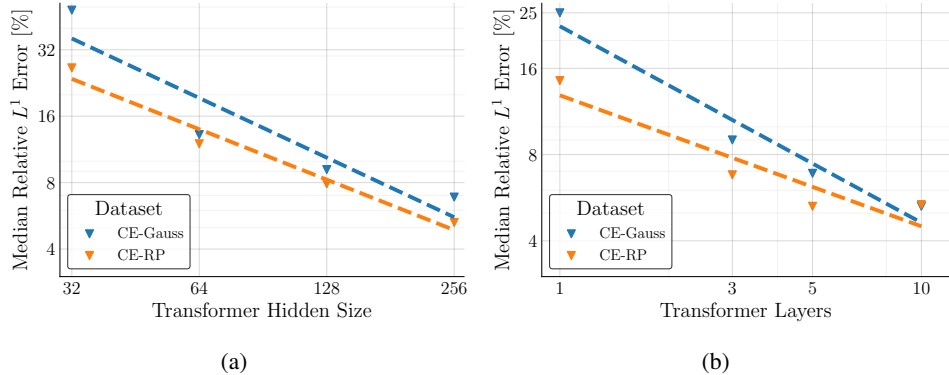

(a)  (b)

Figure E.2: Relative median $L^1$ test errors at $t = t_{14}$ with different strategies for scaling model sizes. The x-axis in the left plot corresponds to all the following hyperparameters: THS and TL.

to $9.0\%$ at $TL = 3$, and ultimately down to $5.28\%$ at TL=10 on CE-Gauss. Increasing TL to 10 also expands the parameter count to 11.2M, nearly doubling the training time per epoch (260s).

Figure E.2 illustrates how errors decrease as we scale THS or TL, while Figure E.3 shows the effect of changing LC or the total parameter count. The largest model tested reaches 11.2M parameters and attains around $5\%$ error on CE-Gauss, demonstrating the potential to improve accuracy by investing in more computational resources.

**Data Size** So far, we have discussed how increasing model size affects accuracy. In this subsection, we turn our attention to data scaling: we examine how the learned operator's performance changes as the number of training samples (trajectories or static solutions) grows. Figure E.4 illustrates two sets of experiments:

**(a)** Time-Dependent (Fluid) Datasets. We plot the final-time ($t = t_{14}$) error on multiple fluid PDE benchmarks as a function of the training set size $\{128, 256, 512, 1024\}$. All of these datasets use partial-grid subsampling. The results confirm that as we increase the number of training trajectories, errors consistently drop across all fluid datasets, often in a near-linear fashion with respect to the number of training trajectories. This trend highlights the model's capacity to benefit from additional time-series diversity.

**(b)** Time-Independent (Static) Datasets. We similarly measure how the final solution error decreases when expanding the dataset size to $\{128, 256, 512, 1024, 2048\}$ for three static PDE tasks: Poisson-Gauss, Poisson-C-Sines, and Elasticity. Note that elasticity is limited to at most 1024 samples due to

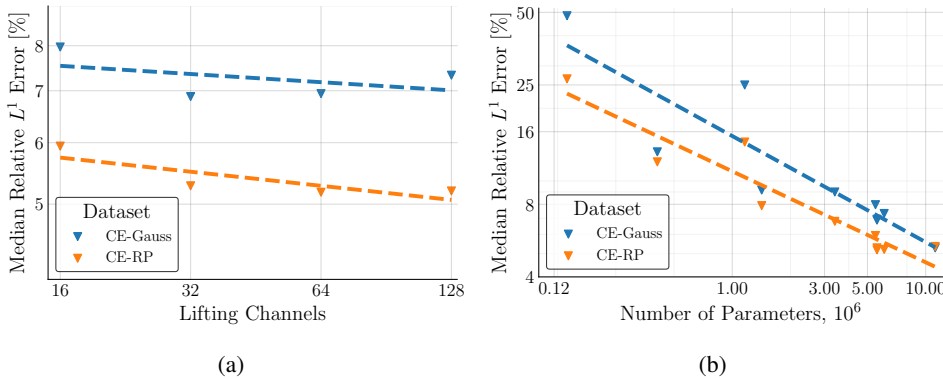

|     |     |
| --- | --- |
| (a) | (b) |

Figure E.3: Relative $L^1$ test errors at $t = t_{14}$ with different strategies for scaling model sizes. The x-axis in the left plot corresponds to all the following hyperparameters: LC, and the total number of trainable parameters.

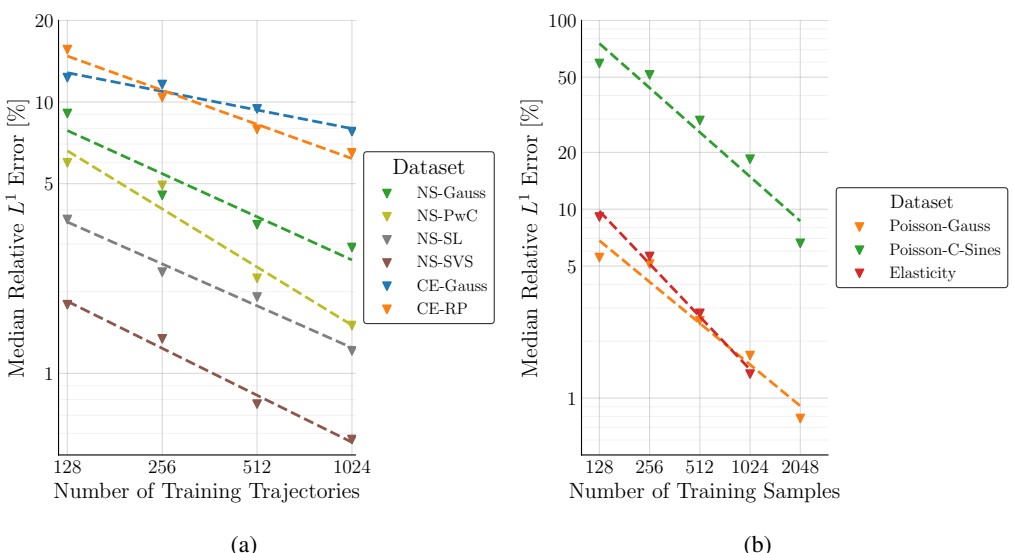

|     |     |
| --- | --- |
| (a) | (b) |

Figure E.4: Relative L1 test errors against the size of training dataset. The left plot shows these results for the time-dependent fluid datasets, and the right plot for time-independent datasets. The lines show linear regression slopes for each dataset.

data availability. Across all these static problems, we observe a consistent downward slope in error as the number of samples increases, again underscoring the advantage of larger training sets.

Overall, in both dynamic (time-dependent) and static (time-independent) scenarios, GAOT exhibits a scalable relationship between training set size and error reduction. As the training data grows, the learned operator converges more reliably to the underlying PDE solution. This robust data scaling property supports our premise that GAOT can serve as a strong foundation model backbone for PDE tasks, becoming increasingly accurate with more extensive datasets.

### E.7 Resolution Invariance

One of the core properties for operator learning is resolution invariance–the ability to train on a specific discretization yet accurately predict solutions at higher/lower resolutions. To validate this property in our GAOT framework, we conduct experiments on a time-independent PDE, Poisson-Gauss.

Figure E.5 illustrates the resolution invariance capabilities of GAOT. In this experiment, GAOT was trained using data discretized by 2048 points. Its performance was then evaluated across a spectrum

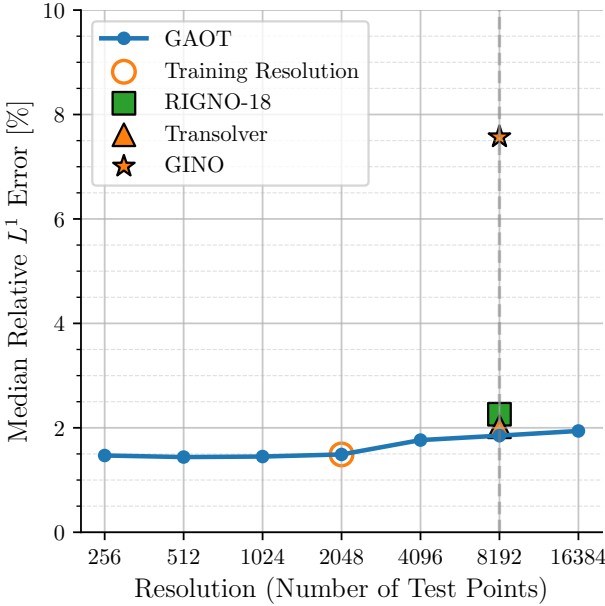

Figure E.5: The GAOT model is trained at a resolution of 2048 and evaluate at various test resolutions. The results for RIGNO-18, Transolver and GINO correspond to models trained and tested at a resolution of 8192.

of seven distinct resolutions: three sub-resolution settings (256, 512, and 1024 points), the training resolution itself (2048 points), and three super-resolution settings (4096, 8192, and 16384 points). For comparative purposes, Figure E.5 also displays the performance of the top three baseline models from Table 1 (excluding GAOT itself), namely RIGNO-18, Transolver, and GINO. These baseline results were obtained from models that were both trained and tested at a resolution of 8192 points.

The results demonstrate that GAOT possesses excellent resolution invariance. Notably, even when trained at a resolution of 2048 points, GAOT not only generalizes well to higher resolutions but also achieves the best performance when tested at 8192 points. It outperforms the baseline models which were specifically trained for and tested at this higher resolution (8192 points), underscoring GAOT's efficiency and robustness in learning resolution-independent solution operators.

### E.8   Comparison of Neural Field with UPT

The *neural field* property enables continuous evaluation of the learned mapping at arbitrary spatial points, allowing models to train and test at different resolutions-a highly desirable capability for PDE operator learning [1, 45]. This feature allows a model trained on a coarser resolution for efficiency to be tested at finer resolutions for higher accuracy.

In Table 3 of the main paper, we demonstrated this property for GAOT on the industrial-scale 3D Drivaernet++ dataset. However, as noted in this result, possessing the neural field property does not necessarily imply higher accuracy. Indeed, GAOT trained using only 10% of input points but tested on the full resolution (500K surface points) is less accurate than the fully trained version, though still outperforming several baselines.

To further examine this, we compared GAOT with the neural-field-based UPT model [1] on the 2D NACA0012 dataset under two settings: (i) both trained and tested at full resolution, and (ii) trained at one-fourth resolution but tested at full resolution. The results are shown in Table E.7.

Table E.7: Neural field training with 1/4 sub-sampling on the NACA0012 dataset.

| Model | Error (Full Resolution) | Error (Neural Field) |
|-------|-------------------------|----------------------|
| GAOT  | 6.81                    | 9.54                 |
| UPT   | 16.1                    | 16.3                 |

These results reveal that: (a) training at full resolution yields better accuracy than coarse training; (b) GAOT consistently outperforms UPT under both configurations; and (c) UPT's limited improvement with resolution suggests poor convergence, further highlighting the robustness of GAOT as a neural surrogate with flexible spatial generalization.

Table E.8: Quantitative comparison on the DrivAerNet++ dataset. The performance on both subsampled point clouds (10k points) and full-resolution meshes (487k points) is taken from the CarBench paper [15]. While baseline models struggle with the high-resolution input, GAOT is trained directly on the full-resolution mesh, enabling it to capture fine-grained geometric details. We report results for a representative unseen test sample (E_S_WW_WM_648) and the dataset-wide average across all 1154 test samples.

| Model | MAE ($m^2/s^2$) | | RMSE ($m^2/s^2$) | | Rel L2 | | $R^2_{test}$ | |
|---|---|---|---|---|---|---|---|---|
| | 10k | Full | 10k | Full | 10k | Full | 10k | Full |
| PointNet | 32.2 | 32.5 | 62.4 | 63.3 | 0.395 | 0.403 | 0.757 | 0.747 |
| NeuralOperator | 25.9 | 26.3 | 51.2 | 54.4 | 0.324 | 0.347 | 0.837 | 0.813 |
| PointMAE | 23.0 | 24.7 | 46.2 | 52.0 | 0.292 | 0.331 | 0.867 | 0.829 |
| PointNetLarge | 22.0 | 25.1 | 42.1 | 55.3 | 0.266 | 0.352 | 0.890 | 0.807 |
| RegDGCNN | 16.5 | 19.9 | 32.4 | 46.1 | 0.205 | 0.293 | 0.935 | 0.866 |
| PointTransformer | 16.2 | 20.6 | 29.5 | 47.0 | 0.187 | 0.299 | 0.946 | 0.861 |
| TransolverLarge | 12.2 | 18.5 | 25.5 | 47.2 | 0.161 | 0.300 | 0.959 | 0.860 |
| TripNet | 13.6 | 19.8 | 25.1 | 44.0 | 0.158 | 0.297 | 0.961 | 0.863 |
| Transolver++ | 12.3 | 18.3 | 24.6 | 46.0 | 0.156 | 0.293 | 0.962 | 0.867 |
| Transolver | 11.7 | 17.7 | 23.7 | 46.1 | 0.150 | 0.294 | 0.965 | 0.866 |
| AB-UPT | 10.2 | 17.2 | 21.5 | 45.7 | 0.136 | 0.291 | 0.971 | 0.869 |
| **GAOT** (Sample 648) | — | **11.4** | — | **22.8** | — | **0.145** | — | **0.967** |
| **GAOT** (Avg. 1154 samples) | — | 12.9 | — | 23.9 | — | 0.157 | — | 0.958 |

## E.9 Training at full vs. low Resolution for the DrivAernet++ Dataset.

Given these results, we turn our attention to the recent investigation of the question of whether training at full resolution confers any benefits over training at low resolution and simply inferring at full resolution. This question was recently explored in the context of the industrial scale DrivAernet++ dataset within the Carbench framework in [15]. In their study, the authors of [15] trained various AI surrogates at a subsampled resolution of $10K$ points from the full surface point cloud of $487$ K points. Then, they tested the models on an unseen sample at the low-resolution of $10K$ points as well as the full resolution of $487K$ points. To ensure uniformity, they interpolated the results from testing at low-resolution to the full resolution. We reproduce their results verbatim in Table E.8 where all the rows except the last two are identical to Table 3 of [15]. As seen from the Table, the performance of models trained at low-resolution degrades severely when interpolated to the full resolution. In fact, all models suffer from this issue and models such as Transolver and AB-UPT [2] which perform very well at low-resolution suffer from so much degradation in accuracy that their performance is at par with the models such as RegDGCNN, which were also inaccurate at low-resolution.

On the other hand, we trained GAOT on the full resolution and present the results of the single test sample as well as the average for all test samples in the last two rows of Table E.8 to observe that GOAT, trained at full resolution is twice more accurate than all the other models. Moreover, the performance of GAOT trained on full resolution is on par with the models that been trained and even tested on the low-resolution. Thus, this experiment further indicates that training at full resolution, if possible, is the best strategy for yielding accurate neural surrogates.

Finally, we would like to point out that models such as AB-UPT [2] also possess the neural field property and can be tested at full resolution, when trained at low resolution, using the neural field property, rather than just interpolating to high resolution. However, as shown in Table 3, we recall that the neural field property does not necessarily translate to performance retention and one can expect a possible degradation of accuracy, even with this paradigm, while training at low resolutions. Nevertheless, this issue needs to investigated further.

## E.10    Transfer Learning

The set of geometries illustrated in Figure E.6a, represent the varying bluff-body geometries in the Bluff-Body dataset, which is one of the benchmarks presented in Tab. 1 of the main text. This dataset is constructed by simulating compressible flow across diverse bluff-body geometries at varying $Ma$ and $\alpha$, as described in Sec. D.2. The distinct shapes depicted in Figure E.6b were specifically employed for the fine-tuning stage of our transfer learning experiments. The corresponding transfer learning performance, demonstrating the model's ability to adapt from the shapes used for pretraining to the novel ones indicated in Figure E.6b, is presented in Figure 3(c) in the main text.

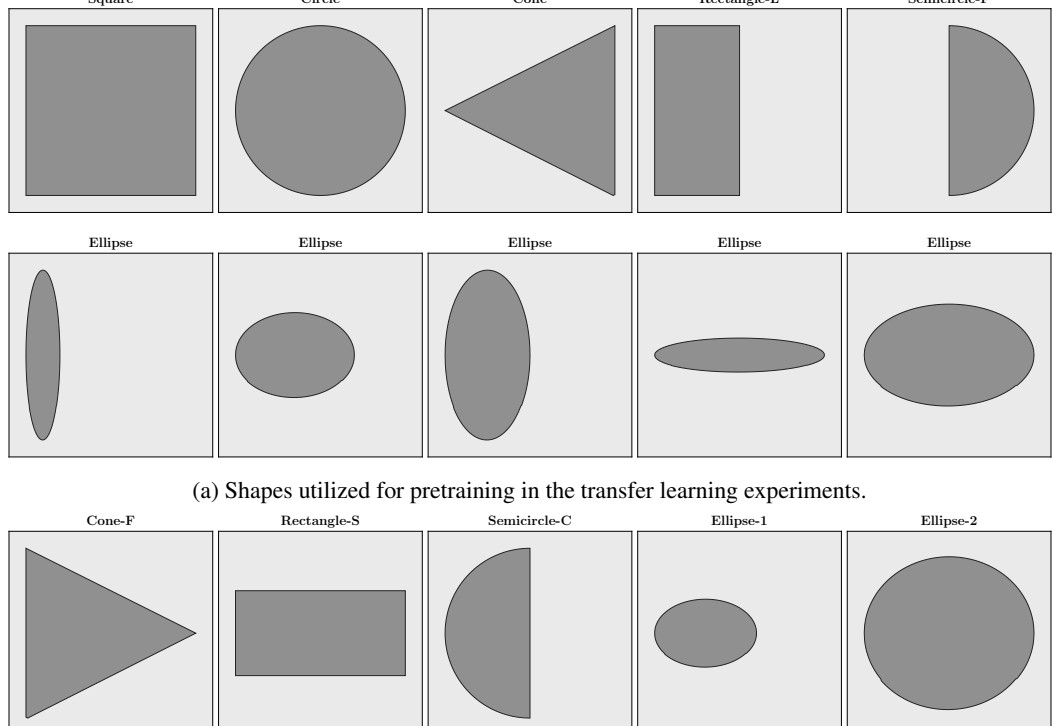

(a) Shapes utilized for pretraining in the transfer learning experiments.

(b) Bluff body shapes employed for the fine-tuning (FT) phase of the transfer learning task.

Figure E.6: The geometries in (a) are included in the Bluff-Body dataset in Tab. 1. Shape-* (*: C, F, S, L; Shape: Semicircle, Cone, Rectangle) indicates shapes and their contact surfaces (*) with respect to the flow. Here, F - flat surface, C - curved surface, L - larger side, S - smaller side.

## E.11    Training Randomness

In order to quantify the dependence of the final model performance on the inherent randomness in the training process, such as in weight initialization, we trained the GAOT model six independent times. Each training run utilized a different seed for the pseudo-random number generator. These experiments were conducted on the Bluff-Body dataset. The statistics of the resulting relative $L^1$ test errors across these six runs are summarized in Table E.9. The standard deviation of these errors is $0.12$. This relatively small standard deviation suggests that the GAOT model exhibits good stability with respect to the random aspects of the training procedure.

Table E.9: Statistics of relative $L^1$ test errors with different random seeds. We train GAOT on Bluff-Body dataset, which is repeated 6 times.

| Dataset | Error [%] |
|---|---|
| | Mean $\pm$ Standard deviation |
| Bluff-Body | $2.39 \pm 0.12$ |

## E.12 Runtime Comparison between GAOT and Classical Solvers

The main rationale for the design of efficient neural surrogates such as GAOT lies in the fact that classical numerical PDE solvers are slow, particularly in industrial 3D problems. On the other hand, neural operators are ultra-fast to infer. We discuss this issue from the perspective of GAOT in this section. To begin with, Table E.10 summarizes the inference times of GAOT and compares them with the runtimes of classical numerical solvers across several CFD datasets. For fairness, all GAOT inference times include graph construction overhead.

Table E.10: Comparison of GAOT inference time with traditional CFD solvers.

| Dataset | Mesh Points | Inference Time (ms) | Traditional Solver | Speedup |
|---------|-------------|---------------------|--------------------|---------|
| Bluff-body | 14,000 | 9.77 | $283 \sim 419$ s | $\sim 3 \times 10^4$ |
| NS-SL | 16,384 | 10.14 | 0.1 s | $\sim 10^1$ |
| DrivaerNet++ | 500,000 | 365.36 | 375 core hours | $\sim 3.7 \times 10^7$ |
| DrivaerML | 9,000,000 | 14,091.08 | 61,440 core-hours | $\sim 1.5 \times 10^9$ |

The NS-SL dataset is taken from Ref. [22], following the setup of Ref. [36] with random point-cloud inputs. Their optimized GPU-based spectral viscosity solver achieves approximately 0.1s per sample. For our new datasets-NACA0012, NACA2412, RAE2822, and Bluff-body-which simulate compressible flow around airfoils and bluff bodies, the traditional CPU-based solver required between 283s and 419s per sample due to adaptive grids and iterative refinement (see SM D for details). For the large-scale 3D benchmark DrivaerNet++ [14], the original authors reported an average cost of approximately 375 CPU hours per sample. For DrivAerML, [3] indicates that each scale-resolving CFD simulation using the hybrid RANS–LES approach required around 40 hours on 1536 cores (61,440 core-hours) on AWS HPC clusters, which represents an industrial-grade high-fidelity CFD workflow that includes near-wall resolution and statistical convergence monitoring.

Overall, traditional numerical solvers span a runtime range from 0.1s to a few minutes for 2D problems discussed in this work, and up to tens of thousands of core-hours for industrial 3D simulations. In contrast, GAOT performs inference within 8.95-10.14 ms for 2D datasets and around 365ms for 3D DrivaerNet++, achieving a remarkable speedup of 1–5 orders of magnitude for 2D problems and up to 10 orders of magnitude for 3D industrial-scale benchmarks. The reason why DrivAerML shows a significantly higher runtime compared to other datasets is mainly due to the excessive time spent on graph building. In fact, if we exclude the graph building process, its inference speed is 446.06 ms. The relatively long graph building time can be attributed to two factors: first, the large scale of the input mesh points; and second, to control memory usage while covering the entire solution domain, we employed more complex graph building techniques (see Section B.5). Nevertheless, even under these conditions, the efficiency improvement compared to traditional solvers remains at the billion order of magnitude level as its runtime of approximately 15 secs is completely dwarfed by the more than $60K$ core hours run time of the LES based ground truth simulations.

One can argue that the runtime of classical numerical PDE solvers can be reduced by coarsening the resolution. This does not hold for the industrial scale datasets as there is a minimum mesh resolution that is essential for the underlying physics to be resolved. However, on the academic 2D datasets, we can perform such a coarsening and examine if the cost-accuracy pay-off for GAOT still holds. To this end, we consider the NACA0012 dataset and coarsen the resolution of the underlying finite-volume solver by two levels of refinement. The underlying error is now $4.5\%$, when compared to the ground truth but the run-time reduces from 7 mins to approximately 1 minute. This error is comparable to the error of GAOT ($6.8\%$), while the runtime of GAOT for this dataset is approximately 10 milli-seconds, leading to a speedup of 6000. This further demonstrates the enormous advantage that an efficient and accurate neural operator such as GAOT can provide, when compared to classical numerical PDE solvers.

# F  Ablation Studies

## F.1  Encode-Process-Decode

In this subsection, we investigate the performance of four different *encode–process–decode* architectures on both time-dependent and time-independent PDE benchmarks. The four models considered

are GAOT (ours), Regional Attentional Neural Operator (RANO), Regional Fourier Neural Operator (RFNO) and GINO [29], with components are *Message-Passing (MP)* graph neural network [19], *Transformer* [47], *Fourier Neural Operator (FNO)* [27], *Graph Neural Operator (GNO)* [28] and proposed *Multi-scal Attentional GNO (MAGNO)*. Table F.1 summarizes the components of each model. All variants follow an *encode–process–decode* pipeline but differ in how graph, Fourier, or transformer-based mechanisms are deployed.

Table F.1: Components for different encode–process–decode designs.

| Model | Encode | Process | Decode |
|-------|--------|---------|--------|
| GAOT | MAGNO | Transformer | MAGNO |
| RANO | MP | Transformer | MP |
| RFNO | MP | FNO | MP |
| GINO | GNO | FNO | GNO |

Figure F.1 shows the *median relative $L^1$ error* for each model on six PDE datasets (4 time-dependent PDEs, 2 time-independent PDEs). All models are trained for 500 epochs under the same data splits and hyperparameter conditions. We can see that GAOT consistently achieves strong performance and robustness across all six datasets. Its errors remain low, highlighting the effectiveness of combining MAGNO for local geometric encoding with transformer-based global attention. GINO ranks second in overall accuracy, yet exhibits noticeable difficulties on NS-Gauss and Poisson-C-Sines. RANO and RFNO perform moderately well on simpler datasets (e.g. Elasticity), but show instability on more challenging tasks (e.g. NS-SVS or NS-Gauss). This indicates that reliance on message-passing or FNO-based processors alone may not be sufficient to handle diverse PDE and geometry conditions with the same level of robustness.

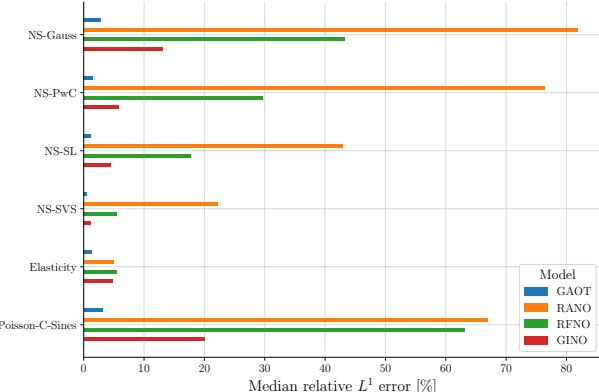

Figure F.1: Median relative $L^1$ errors (%) of GAOT, RANO, RFNO, and GINO on six PDE benchmarks.

Overall, these results reinforce GAOT's stability across multiple PDE settings. Even with a fixed training protocol (500 epochs for each dataset), GAOT consistently converges faster and more reliably, underscoring the advantage of geometry-aware tokens, multiscale attention, and the flexible transformer backbone.

## F.2 Tokenization Strategies

In Section B.1, we have introduced three tokenization methods. Here, we compare these strategies on two datasets, Elasticity and Poisson-C-Sines. Figure F.2 shows the final median relative $L^1$ errors for each approach. The Strategy I consistently achieves the best performance on both unstructured datasets. Strategy II & III perform similarly to Strategy I on simpler datasets (elasticity), but can fail to converge on the more challenging one, Poisson-C-Sines. Similar situations also happen on models like UPT and GNOT in Tab. 1 of main text. Overall, Strategy I emerges as the most robust approach in our current experiments. While Strategies II and III show promise, they require more careful optimization to match Strategy I's reliability.

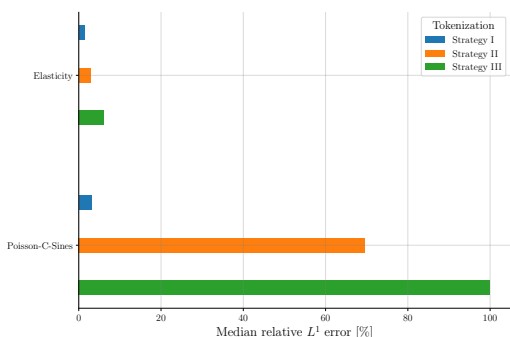

Figure F.2: Median relative $L^1$ errors (%) comparing three tokenization strategies on Elasticity, and Poisson-C-Sines. The Strategy I, II, III corresponds to the methods discussed in Section B.1.

Next, we focus on Strategy I and study how varying the *number of latent tokens (LT)*, *patch size (PS)*, and *radius (GR)* affect performance. Note that here we do not use multiscale radii; each token has a single radius. Table F.2 summarizes experiments on the elasticity and Poisson-Gauss datasets. Results show that fewer tokens (e.g., [32,32]) can degrade performance in some cases

Table F.2: Median relative $L^1$ errors (%), parameter counts, and training time with different numbers of latent tokens (LT), patch sizes (PS), and radii (GR).

| Model Size | | Hyperparameters | | | Median Relative $L^1$ Error [%] | |
|---|---|---|---|---|---|---|
| Params [M] | Time [s/it] | LT | PS | GR | Elasticity | Poisson-Gauss |
| 5.60 | 10.1 | [64, 64] | 2 | 0.033 | 1.80 | 1.05 |
| 6.00 | 3.06 | [64, 64] | 4 | 0.033 | 1.71 | 1.57 |
| 10.7 | 2.20 | [64, 64] | 8 | 0.033 | 1.60 | 1.65 |
| 5.56 | 10.1 | [32, 32] | 1 | 0.066 | 3.41 | 1.22 |
| 5.60 | 3.00 | [32, 32] | 2 | 0.066 | 2.25 | 1.22 |
| 6.00 | 11.8 | [128, 128] | 4 | 0.033 | 1.67 | 1.72 |
| 10.7 | 5.02 | [128, 128] | 8 | 0.033 | 1.62 | 1.22 |

(elasticity), presumably because the domain coverage becomes coarser, making it harder to capture local variations. More tokens ([64,64] or [128,128]) typically improve accuracy and stabilize convergence. Nevertheless, computational costs rise when the number of tokens grows, as transformer attention scales quadratically with token count. Increasing the patch size (PS) reduces the number of tokens entering the transformer, lowering the training time. Encouragingly, performance does not degrade sharply with larger patches. For instance, going from PS = 2 to 8 is fairly stable across datasets. Note that the overall parameter count can increase if each token aggregates larger local features, but in practice, training runs faster due to fewer tokens in self-attention. Radius (GR) grows if we reduce the number of latent tokens because we need to ensure coverage of the entire physical domain by enlarging the receptive field. This is critical for unstructured or irregular samples, especially if tokens must capture a bigger subregion.

## F.3   Time-Stepping Method

We now investigate how different *time-stepping* formulations (see Section B.6) affect performance on time-dependent PDEs. Specifically, we compare the output, residual, and derivative stepping strategies. Table F.3 reports the median relative $L^1$ errors for six representative fluid dynamics benchmarks on regular grids.

As shown, modeling the operator as a time derivative (derivative column) often yields the lowest final-time errors on all but one dataset (CE-RP, where the *Output* strategy slightly outperforms the others). We hypothesize that treating the operator as $\partial_t u$ naturally enforces a continuous dependence on time, analogous to neural ODEs or residual networks [21, 11], which can improve stability and accuracy over multiple steps. In experiments involving time-dependent PDEs, we therefore use

Table F.3: Median relative $L^1$ errors (%) at final time $t_{14}$ for GAOT with three different time-stepping methods.

| Dataset | Median relative $L^1$ error [%] | | |
|---|---|---|---|
| | Output | Residual | Derivative |
| NS-Gauss | 3.57 | 3.60 | 2.52 |
| NS-PwC | 1.95 | 1.70 | 1.23 |
| NS-SL | 1.78 | 1.49 | 1.29 |
| NS-SVS | 0.60 | 0.60 | 0.56 |
| CE-Gauss | 8.80 | 8.93 | 7.97 |
| CE-RP | 5.17 | 6.12 | 5.94 |

derivative time stepping as the default unless stated otherwise. This approach not only achieves strong final-time accuracy, but also aligns with our design goal of a differentiable, time-continuous operator.

## F.4 Geometric Embedding

As discussed in Section B.3, our framework incorporates a *geometric embedding network* to encode shape and domain information separately from the physical (PDE) state. Table F.4 compares these geometric embedding approaches against a baseline "original" (i.e., no additional geometry embedding) on two original unstructured datasets (Wave-C-Sines, Poisson-C-Sines).

Table F.4: Median relative $L^1$ errors (%) for various geometry embedding approaches. Original omits geometric embedding, while Statistical and PointNet follow Section B.3.

| Dataset | Median relative $L^1$ error [%] | | |
|---|---|---|---|
| | original | statistical | pointnet |
| Wave-C-Sines | 6.50 | 5.69 | 6.07 |
| Poisson-C-Sines | 6.60 | 4.66 | 23.7 |

In the unstructured datasets, including Wave-C-Sines, and Poisson-C-Sines, explicitly encoding domain geometry yields a more pronounced benefit. In particular, the statistical strategy consistently outperforms PointNet on these irregular meshes, and in Poisson-C-Sines, training with the PointNet approach appears unstable (23.7% error). Based on these observations, we use statistical embedding by default for unstructured dataset given its stable and superior performance in most cases.

We further conducted ablation studies on DrivaerNet++ and DrivaerML to examine the effect of statistical geometric embeddings. Three configurations were considered: (1) without geometric embedding (nGEmb-nGEmb), (2) applying geometric embedding only in the encoder (GEmb-nGEmb), and (3) applying geometric embedding in both the encoder and decoder (GEmb-GEmb).

Table F.5: Ablation study on the effect of geometric embedding configurations across DrivAerNet++ and DrivAerML datasets.

| Dataset | GEmb-nGEmb | | GEmb-GEmb | | nGEmb-nGEmb | |
|---|---|---|---|---|---|---|
| | MSE | Mean AE | MSE | Mean AE | MSE | Mean AE |
| DrivAerNet++(p) | 4.2694 | 1.0699 | 4.3119 | 1.0818 | 4.5278 | 1.1036 |
| DrivAerNet++(wss) | 8.6878 | 1.5429 | 8.7783 | 1.5690 | 9.3192 | 1.6125 |
| DrivAerML(p) | 5.1729 | 1.2352 | 10.8591 | 1.7344 | 8.6625 | 1.5693 |
| DrivAerML(wss) | 16.9818 | 2.1640 | 41.0027 | 3.0822 | 24.8614 | 2.4965 |

The results show that introducing geometric embeddings consistently improves model performance compared to not using them. Interestingly, we observed that applying the geometric embedding only in the encoder yields even better performance, but less computational effort. Moreover, for the DrivAerML dataset, adding a geometric embedding in the decoder actually degrades performance, producing results even worse than the model without any geometric embedding. Our analysis indicates that while the model's training loss continues to decrease, its validation loss quickly saturates. We think that incorporating geometric embedding in the decoder, where operations are very close to the

final predictions, makes the model more prone to overfitting. Especially on the 3D industrial dataset, where we used more complex graph-building techniques. These complex graph-building tricks lack the unified principles for the model to learn and likely amplify the overfitting tendency when the decoder also encodes geometric information. Therefore, we only use geometric embeddings in the encoder for our 3D industrial benchmarks.

## F.5 Multiscale Features

As introduced in Section B.2, our encoder can capture multiscale local information by aggregating neighborhood features across multiple radii. Specifically, we compare:

- Single-scale: using a single fixed radius of $0.033$ for each point.
- multiscale: using three radii $[0.022, 0.033, 0.044]$ for each point.

Table F.6: Median relative $L^1$ errors (%) comparing single-scale vs. multiscale features.

| Dataset | Median relative $L^1$ error [%] | |
| --- | --- | --- |
| | Single-scale | multiscale |
| Wave-C-Sines | 5.69 | 4.6 |
| Poisson-C-Sines | 4.66 | 3.04 |

Table F.6 reports the mean relative $L^1$ errors on unstructured datasets including Wave-C-Sines and Poisson-C-Sines. Results show that multiscale neighbors yield a clear reduction in error. For instance, in Poisson-C-Sines, the error decreases from $4.66\%$ to $3.04\%$. This contrast reflects the fact that a single, fixed receptive field on a regularly spaced grid is often sufficient. However, on unstructured domains where the mesh density can vary, using multiple radii helps the network capture both fine and coarse local structures.

## G  Visualizations of Datasets

Estimates produced by trained models are visualized in this section for different datasets.

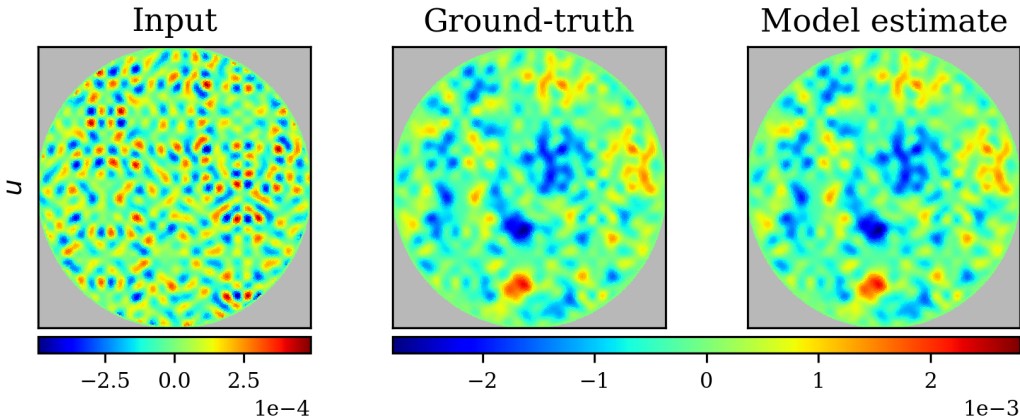

Figure G.1: Model input, ground-truth solution, and model estimate of a test sample of the Poisson-C-Sines dataset.

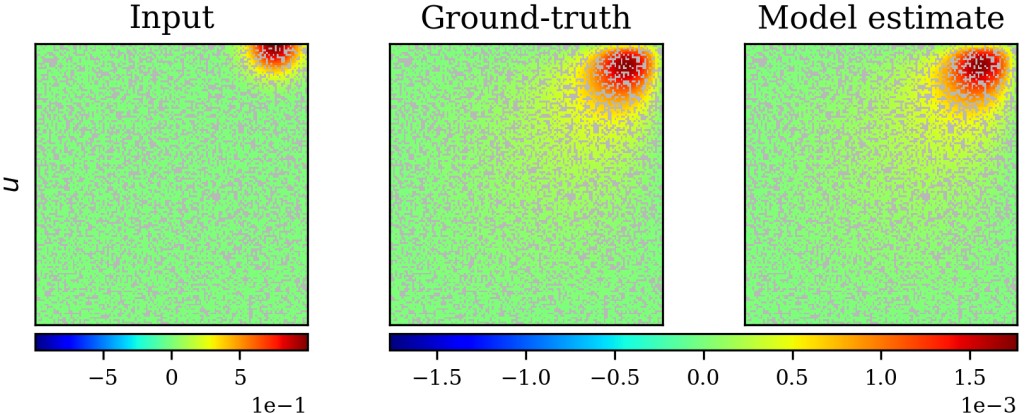

Figure G.2: Model input, ground-truth solution, and model estimate of a test sample of the Poisson-Gauss dataset.

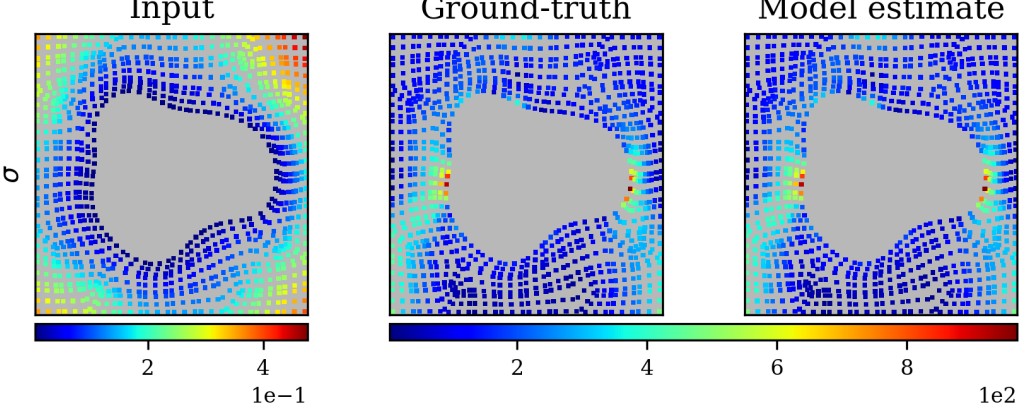

Figure G.3: Model input, ground-truth solution, and model estimate of a test sample of the Elasticity dataset.

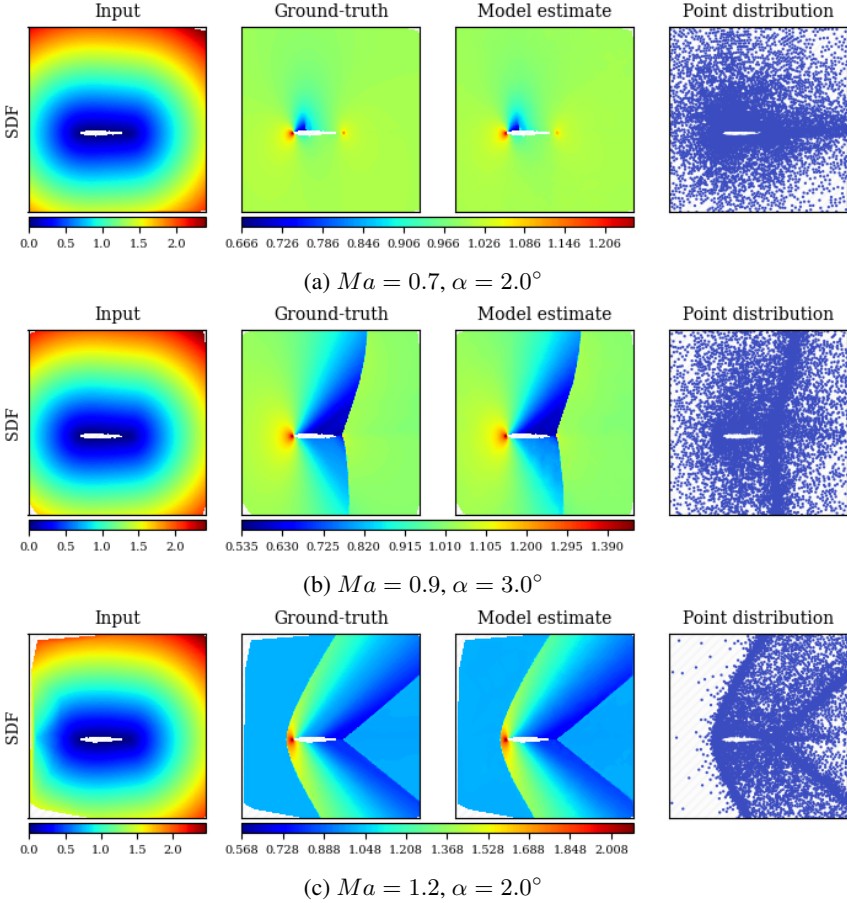

(a) $Ma = 0.7$, $\alpha = 2.0°$

(b) $Ma = 0.9$, $\alpha = 3.0°$

(c) $Ma = 1.2$, $\alpha = 2.0°$

Figure G.4: Model input, ground-truth solution, model estimate and point distribution of test samples of the NACA0012 dataset.

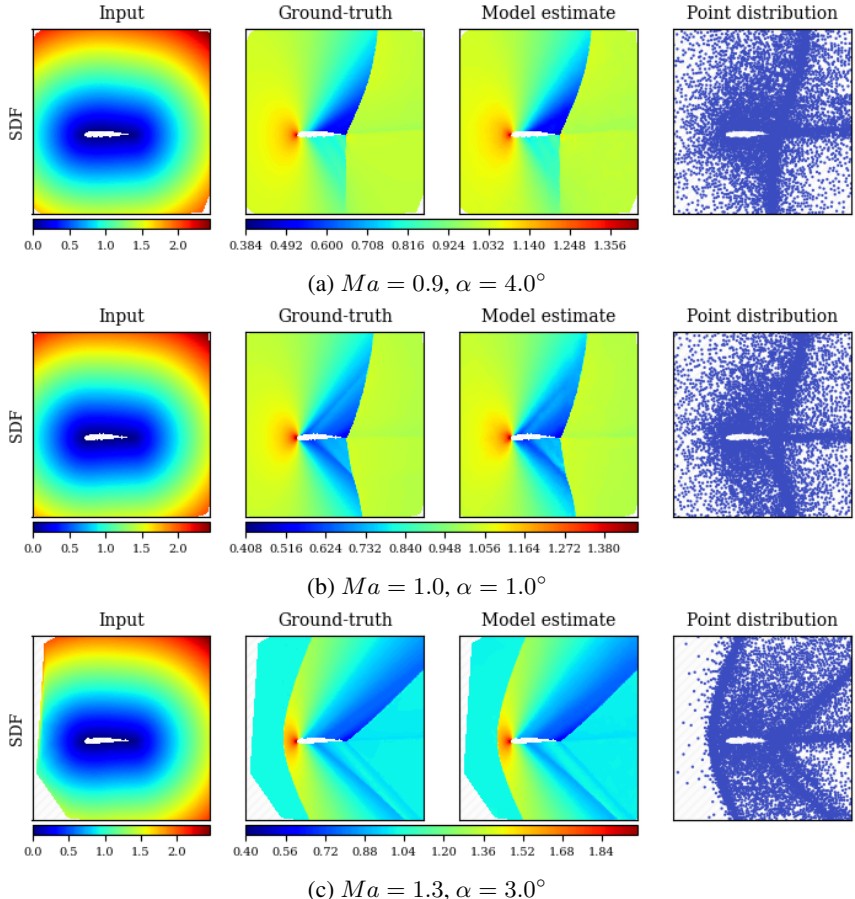

Figure G.5: Model input, ground-truth solution, model estimate and point distribution of test samples of the NACA2412 dataset.

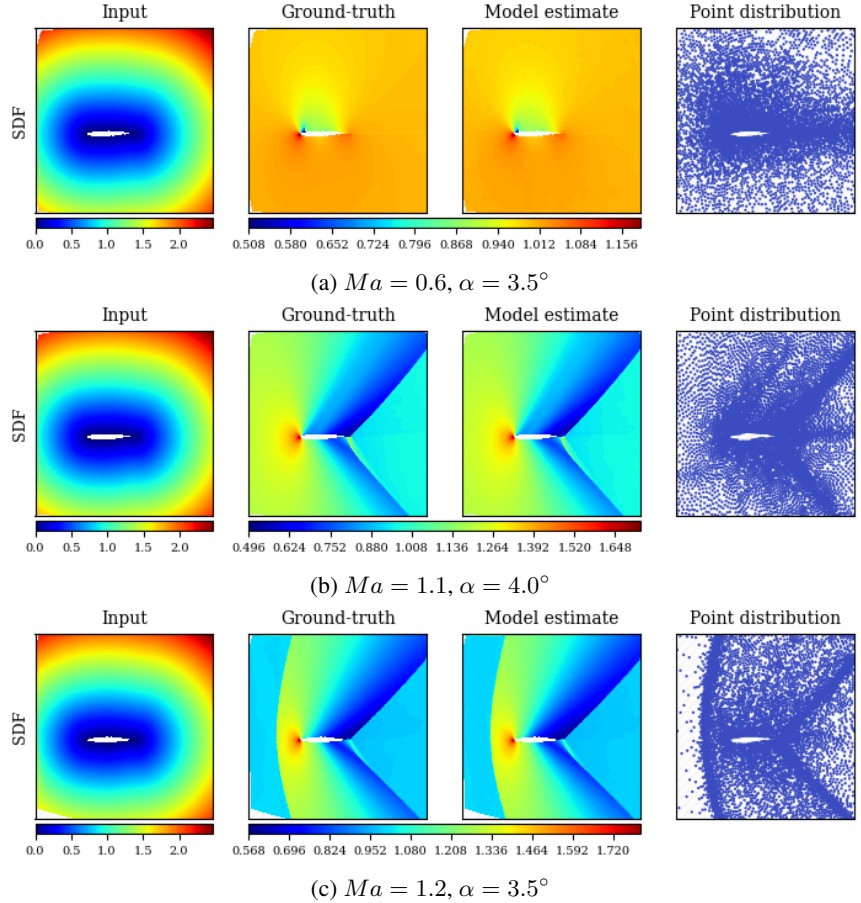

(a) $Ma = 0.6$, $\alpha = 3.5°$

(b) $Ma = 1.1$, $\alpha = 4.0°$

(c) $Ma = 1.2$, $\alpha = 3.5°$

Figure G.6: Model input, ground-truth solution, model estimate and point distribution of test samples of the RAE2822 dataset.

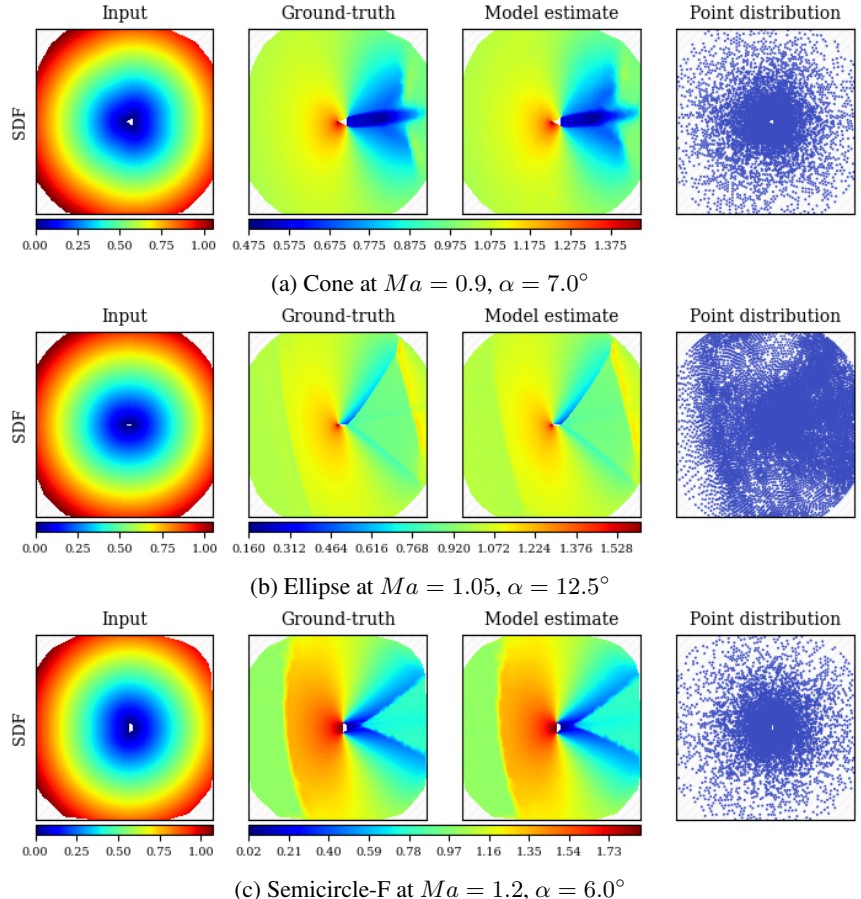

(a) Cone at $Ma = 0.9$, $\alpha = 7.0°$

(b) Ellipse at $Ma = 1.05$, $\alpha = 12.5°$

(c) Semicircle-F at $Ma = 1.2$, $\alpha = 6.0°$

Figure G.7: Model input, ground-truth solution, model estimate and point distribution of test samples of the Bluff-Body dataset.

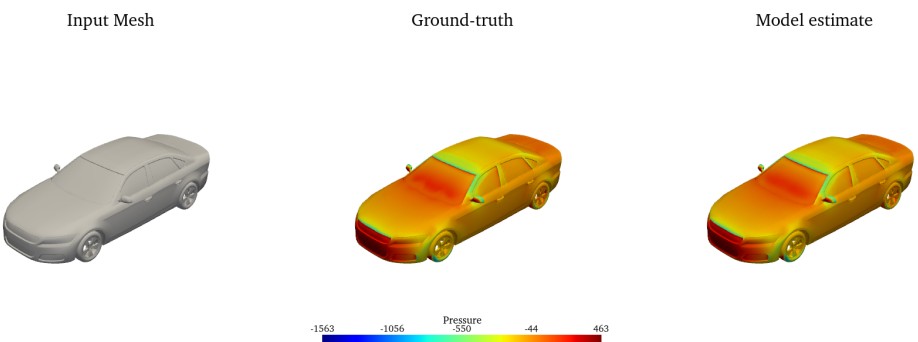

Figure G.8: Model input, ground-truth solution, model estimate of a test sample `N_S_WWS_WM_172` of the surface pressure on the DrivAerNet++.

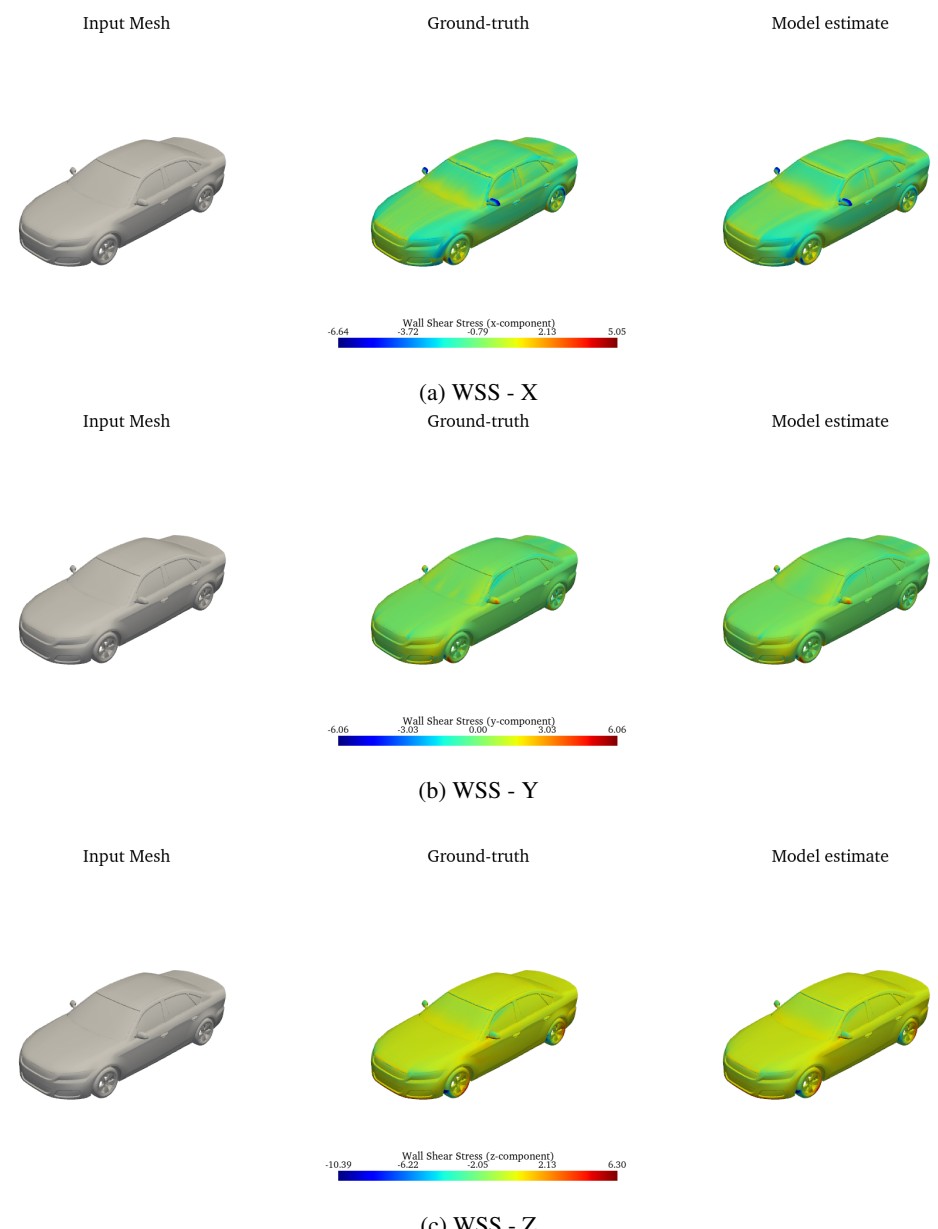

Figure G.9: Model input, ground-truth solution, model estimate of a test sample `N_S_WWS_WM_172` of the surface wall shear stress on the DrivAerNet++.

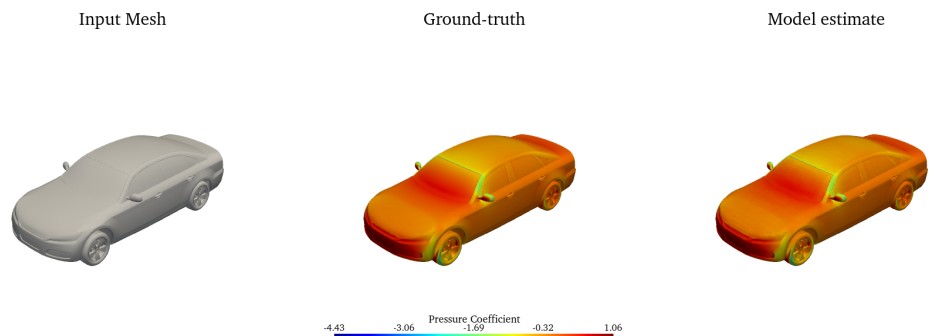

Figure G.10: Model input, ground-truth solution, model estimate of a test sample `Boundary_78` of the surface pressure coefficient on the DrivAerML.

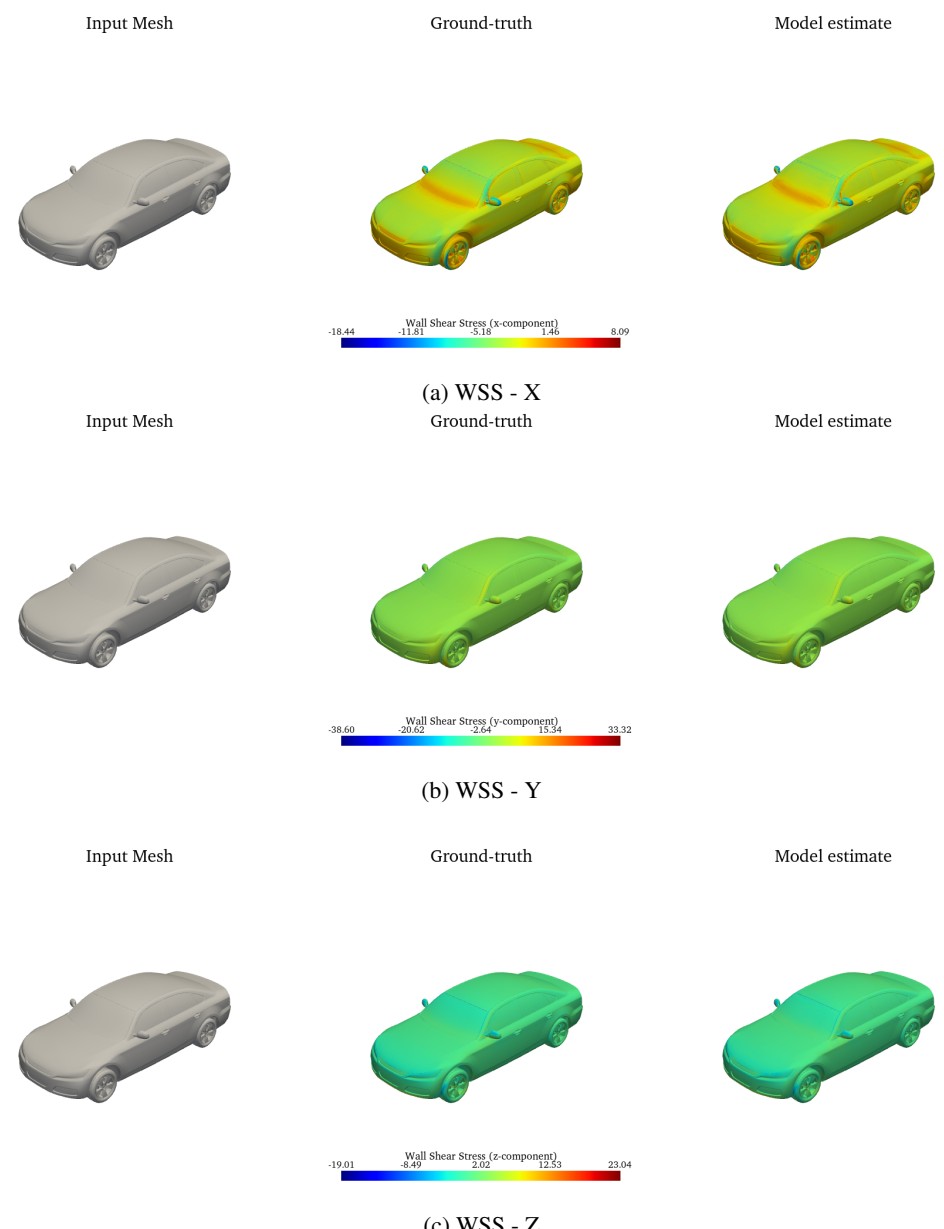

Figure G.11: Model input, ground-truth solution, model estimate of a test sample `Boundary_78` of the surface wall shear stress on the DrivAerML.

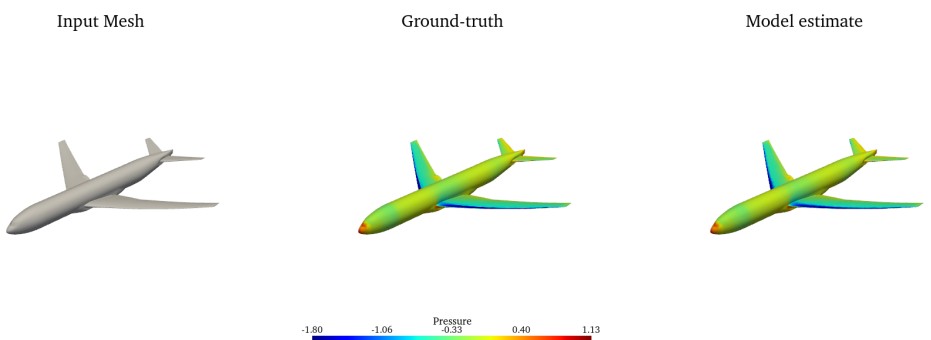

Figure G.12: Model input, ground-truth solution, model estimate of a test sample of the surface pressure on the NASA-CRM.

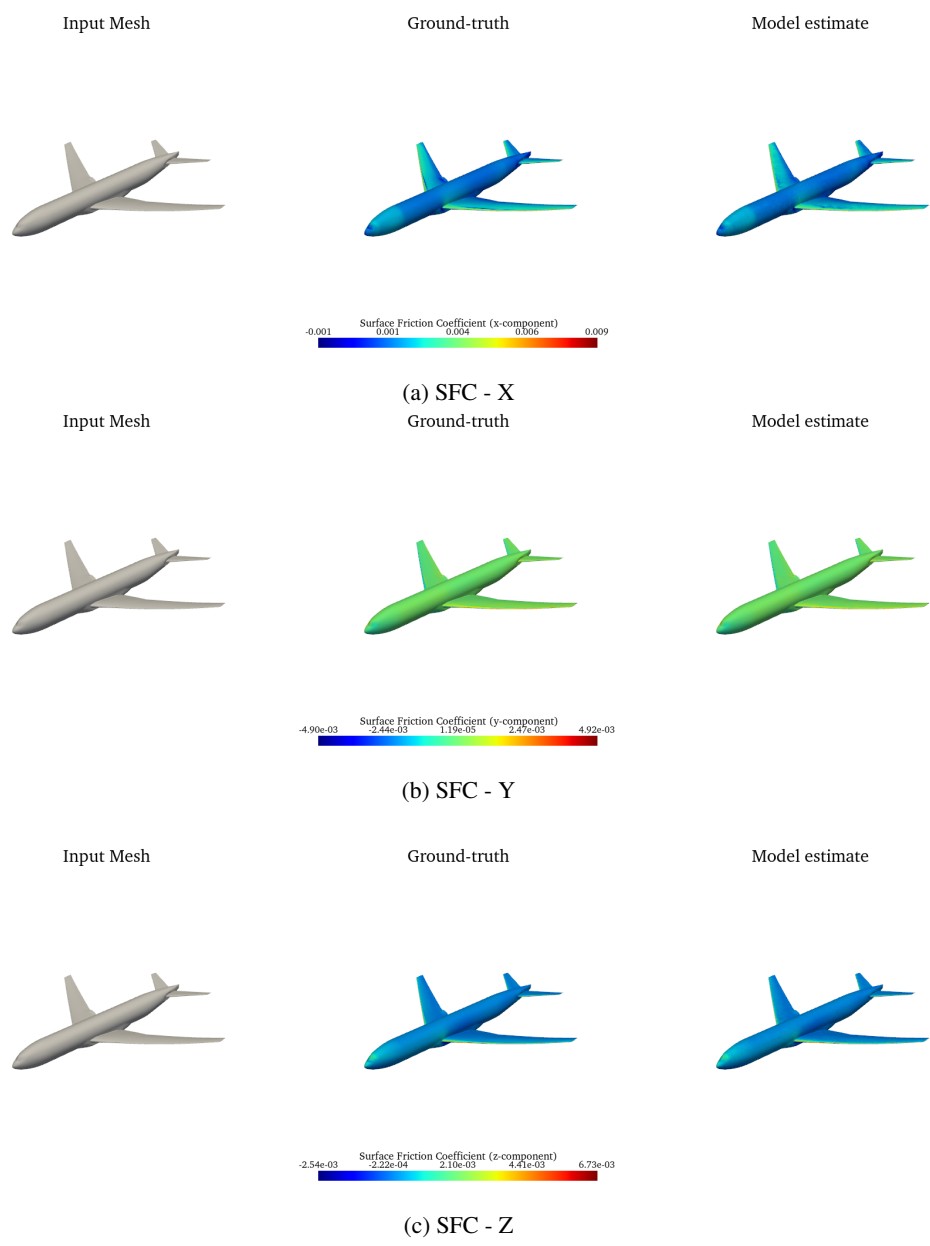

Figure G.13: Model input, ground-truth solution, model estimate of a test sample of the surface friction coefficient on the NASA-CRM.

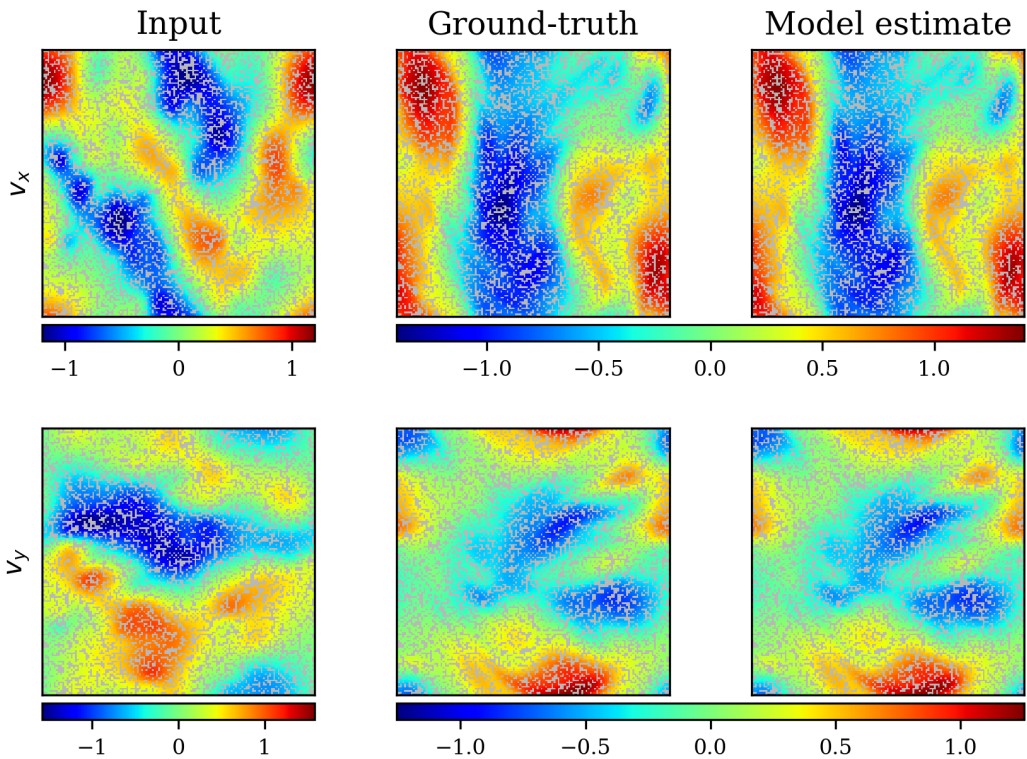

Figure G.14: Model input at $t = t_0$, ground-truth solution and model estimate at $t = t_{14}$ of a test sample unstructured NS-Gauss dataset.

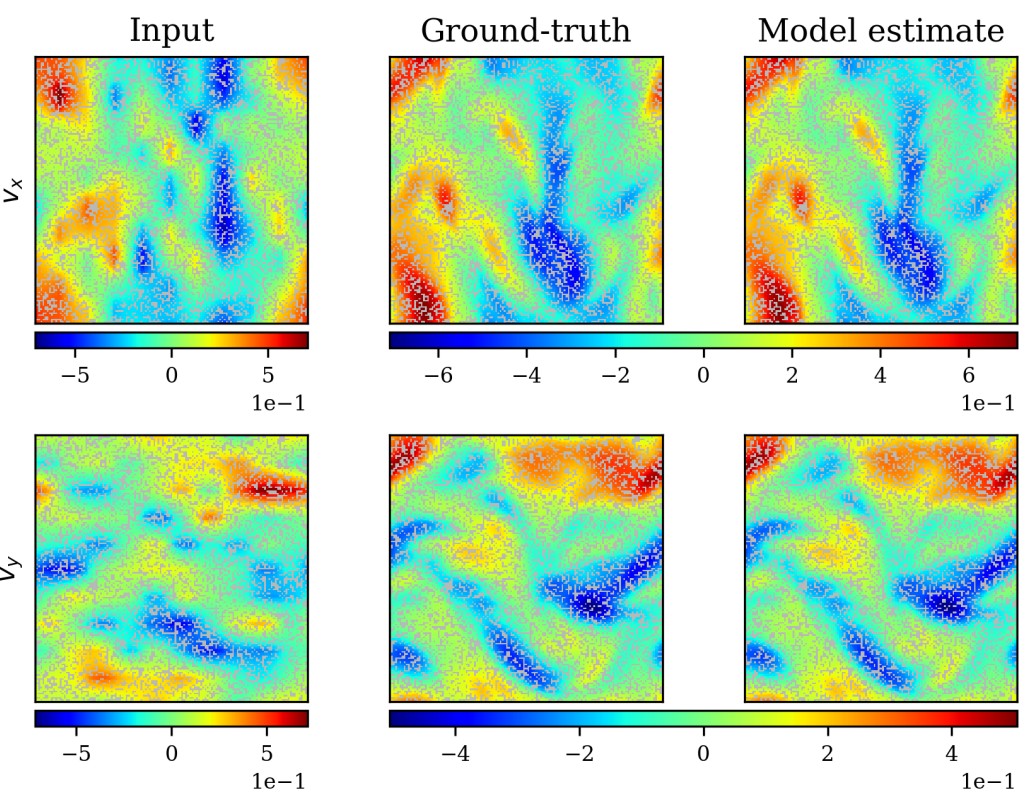

Figure G.15: Model input at $t = t_0$, ground-truth solution and model estimate at $t = t_{14}$ of a test sample unstructured NS-PwC dataset.

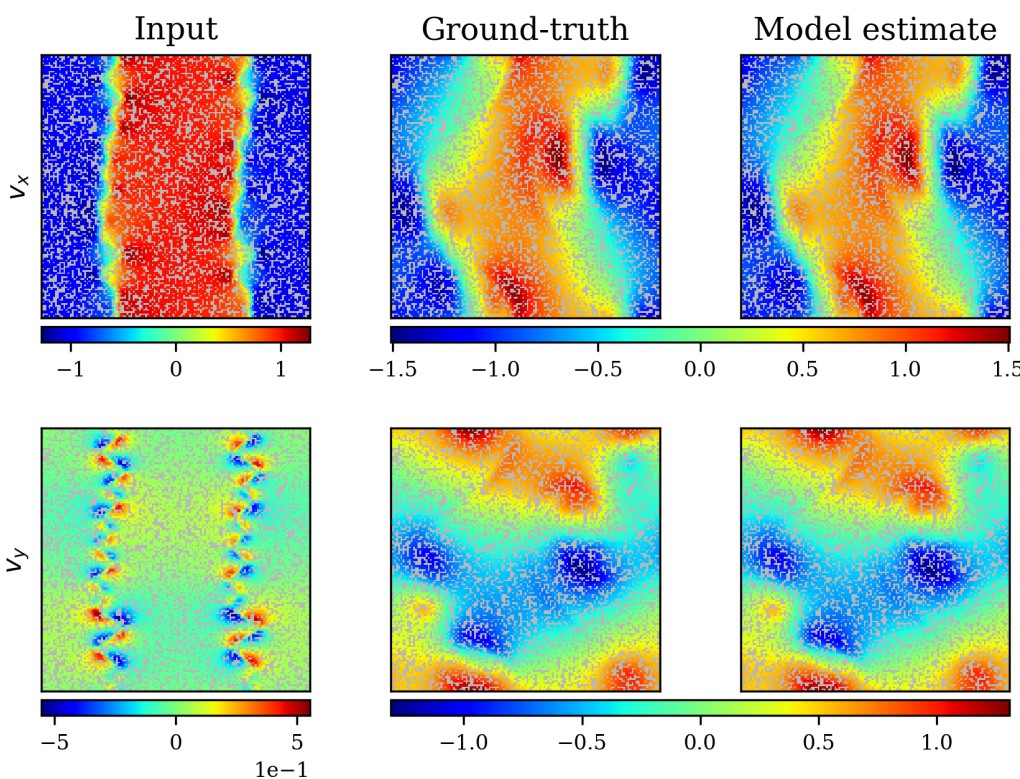

Figure G.16: Model input at $t = t_0$, ground-truth solution and model estimate at $t = t_{14}$ of a test sample unstructured NS-SL dataset.

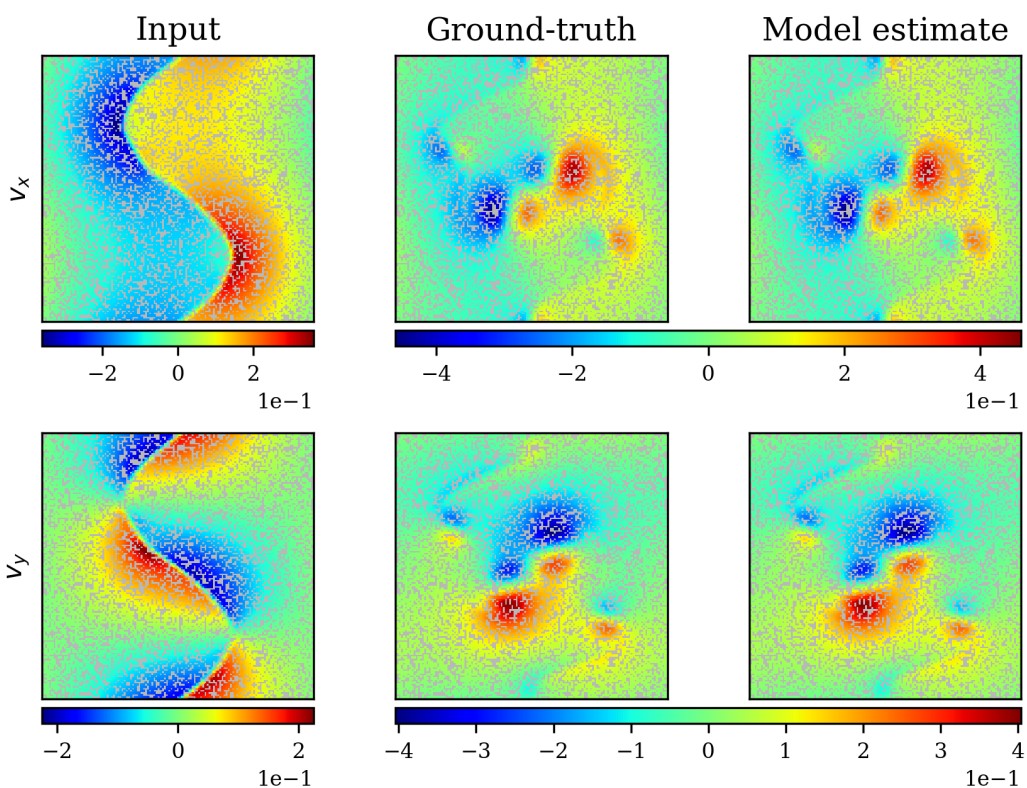

Figure G.17: Model input at $t = t_0$, ground-truth solution and model estimate at $t = t_{14}$ of a test sample unstructured NS-SVS dataset.

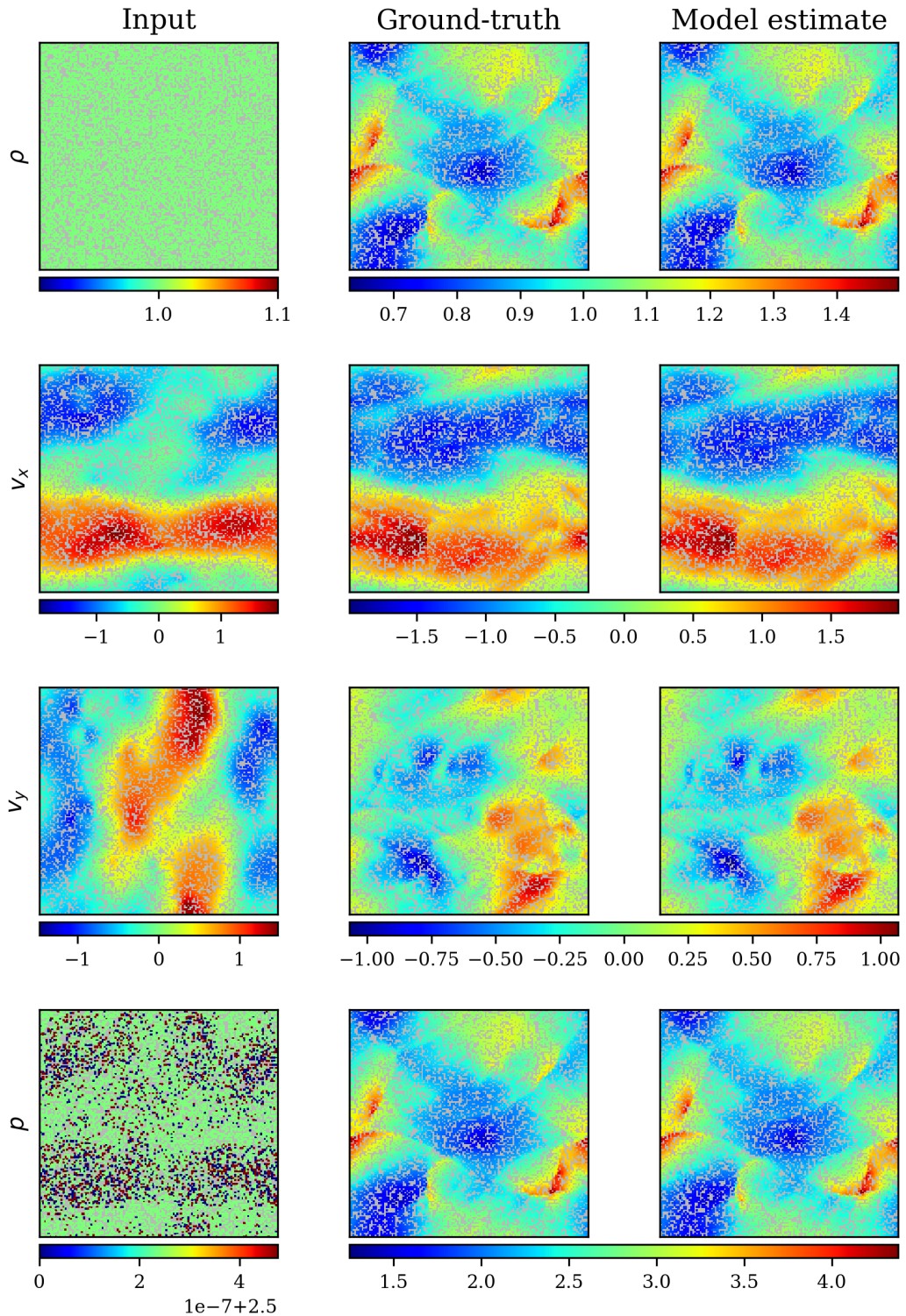

Figure G.18: Model input at $t = t_0$, ground-truth solution and model estimate at $t = t_{14}$ of a test sample unstructured CE-Gauss dataset.

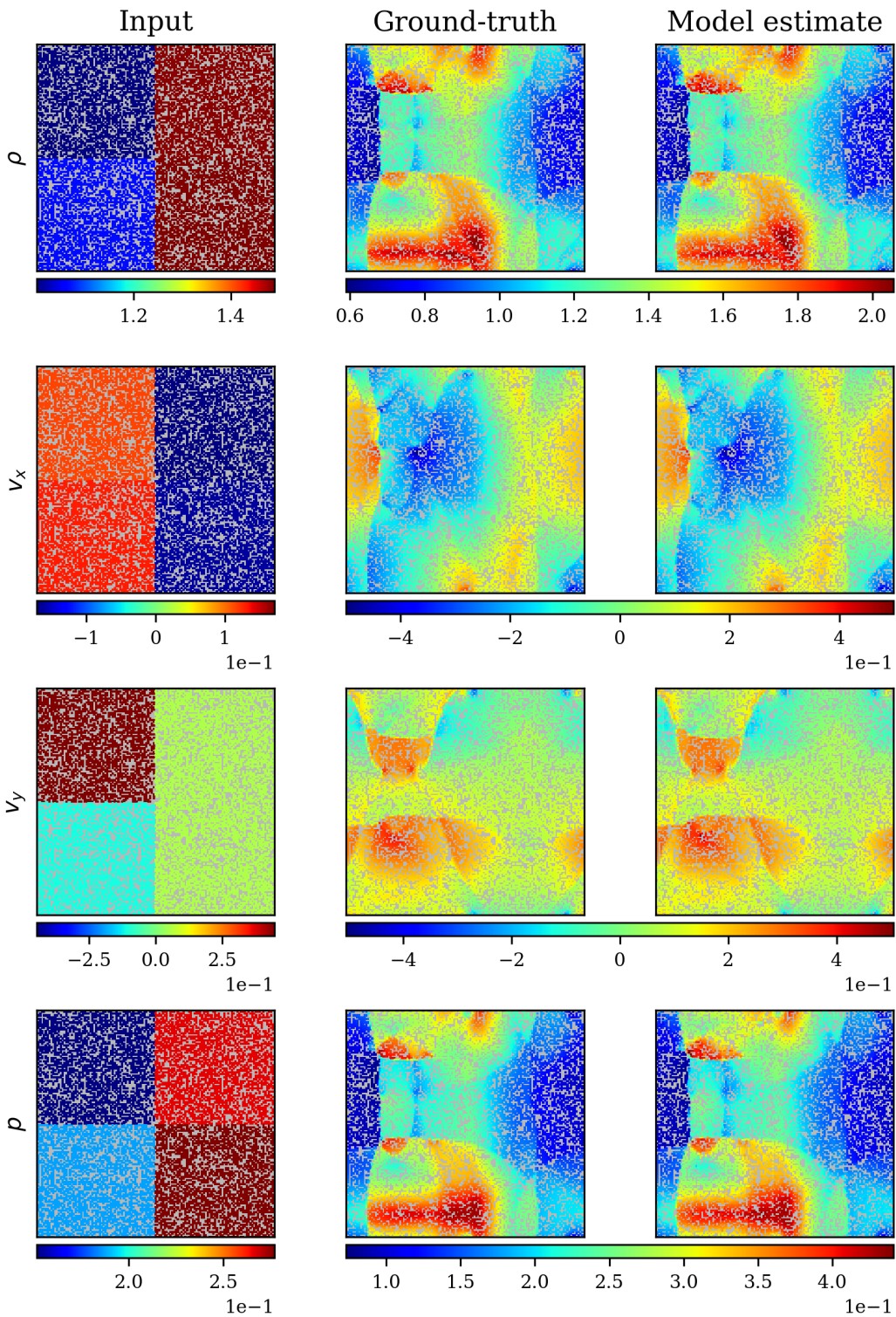

Figure G.19: Model input at $t = t_0$, ground-truth solution and model estimate at $t = t_{14}$ of a test sample unstructured CE-RP dataset.

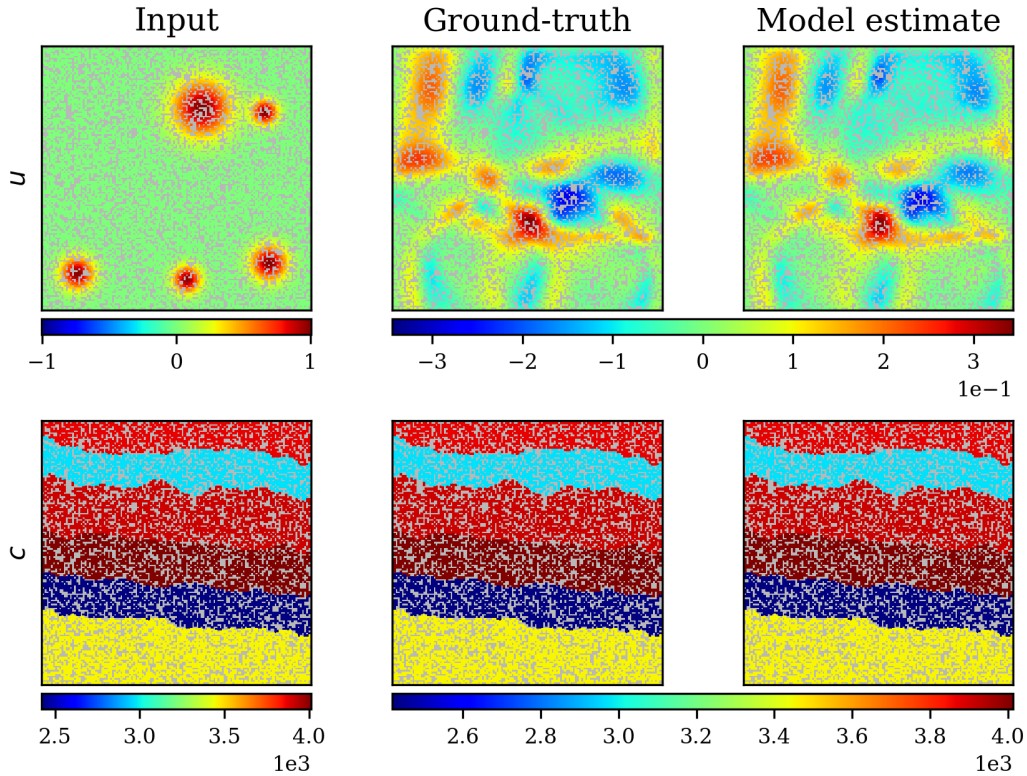

Figure G.20: Model input at $t = t_0$, ground-truth solution and model estimate at $t = t_{14}$ of a test sample unstructured Wave-Layer dataset.

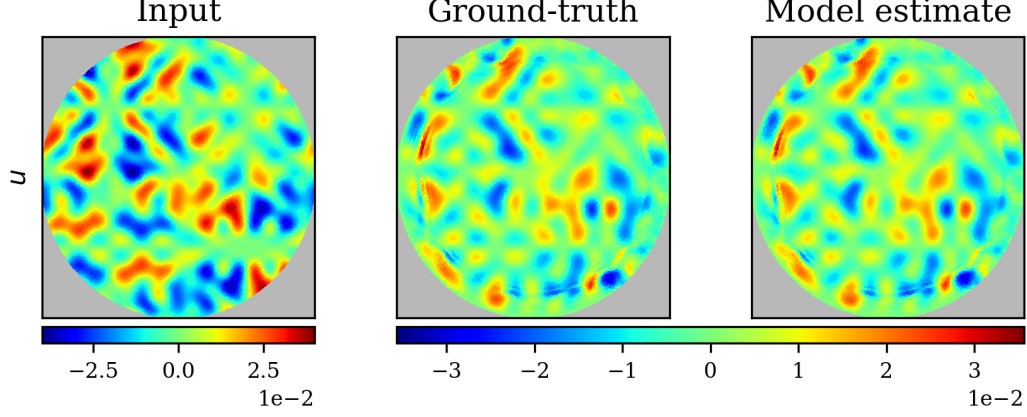

Figure G.21: Model input at $t = t_0$, ground-truth solution and model estimate at $t = t_{14}$ of a test sample Wave-C-Sines dataset.

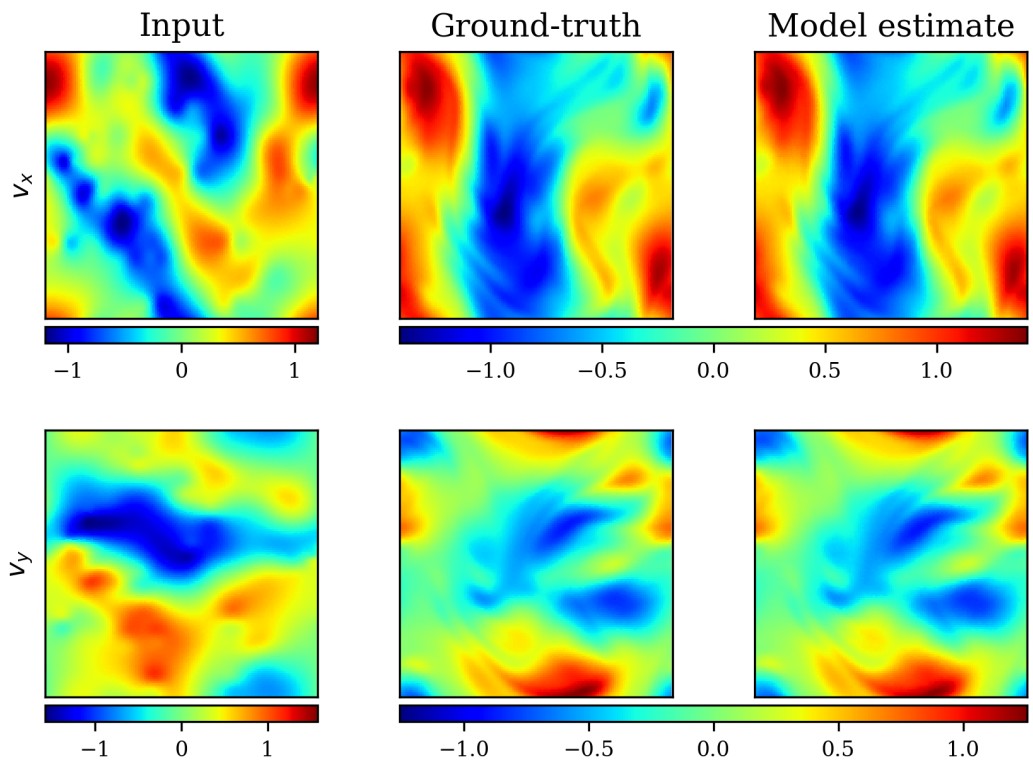

Figure G.22: Model input at $t = t_0$, ground-truth solution and model estimate at $t = t_{14}$ of a test sample NS-Gauss dataset.

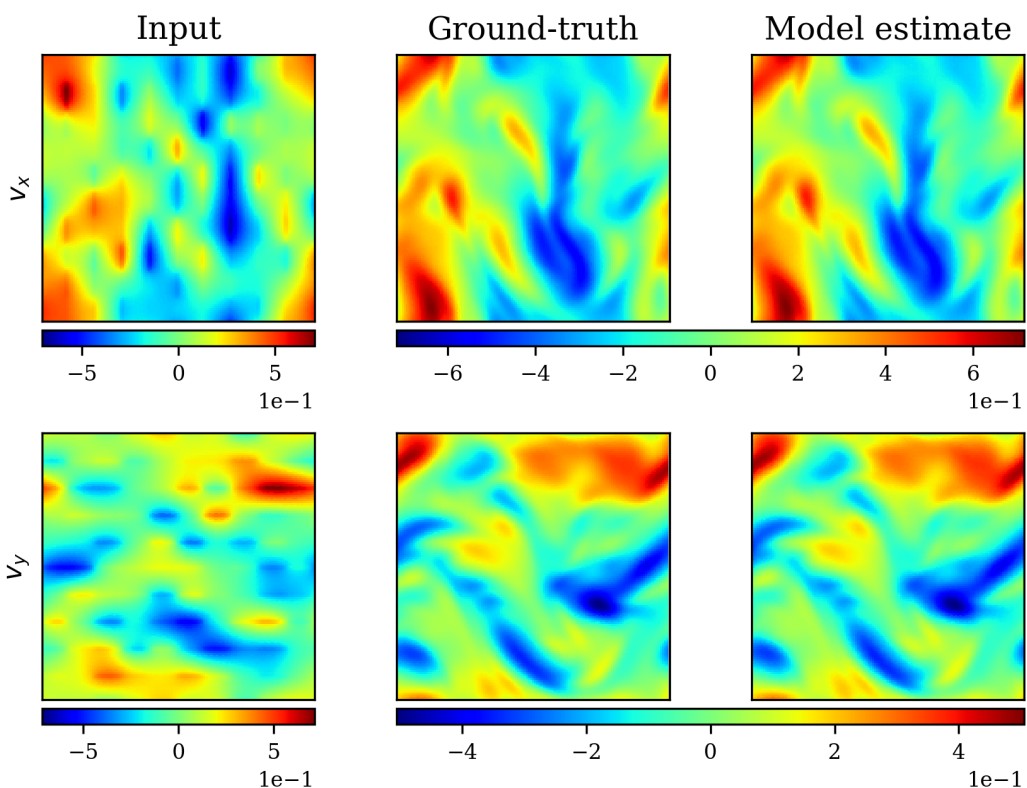

Figure G.23: Model input at $t = t_0$, ground-truth solution and model estimate at $t = t_{14}$ of a test sample NS-PwC dataset.

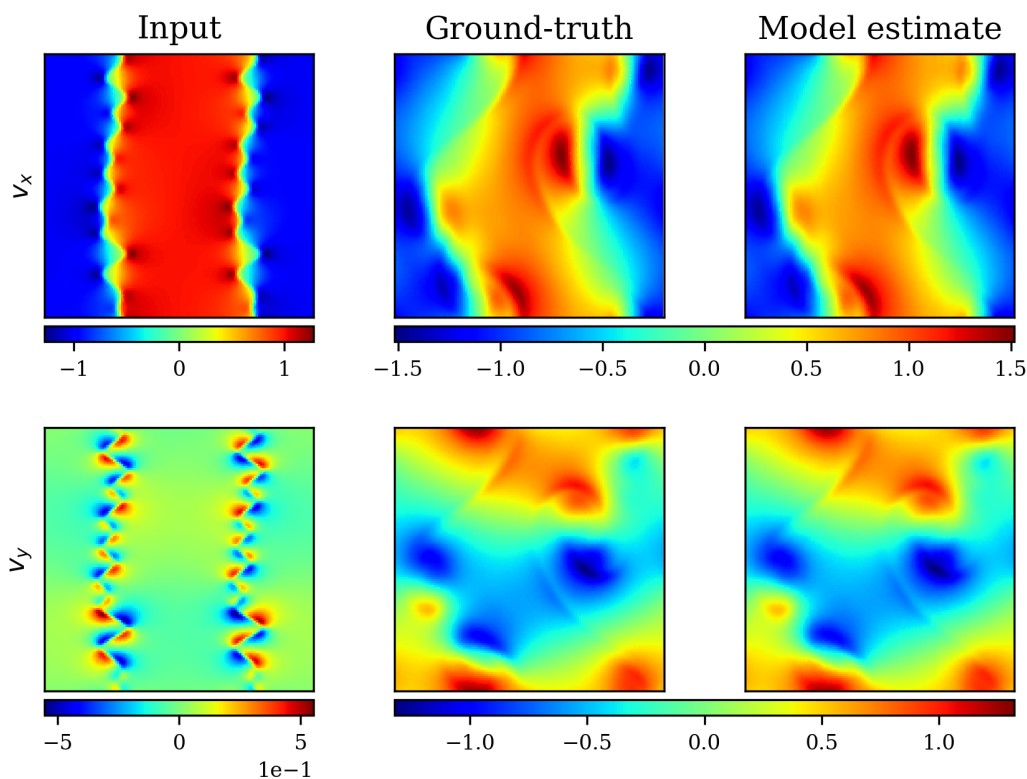

Figure G.24: Model input at $t = t_0$, ground-truth solution and model estimate at $t = t_{14}$ of a test sample NS-SL dataset.

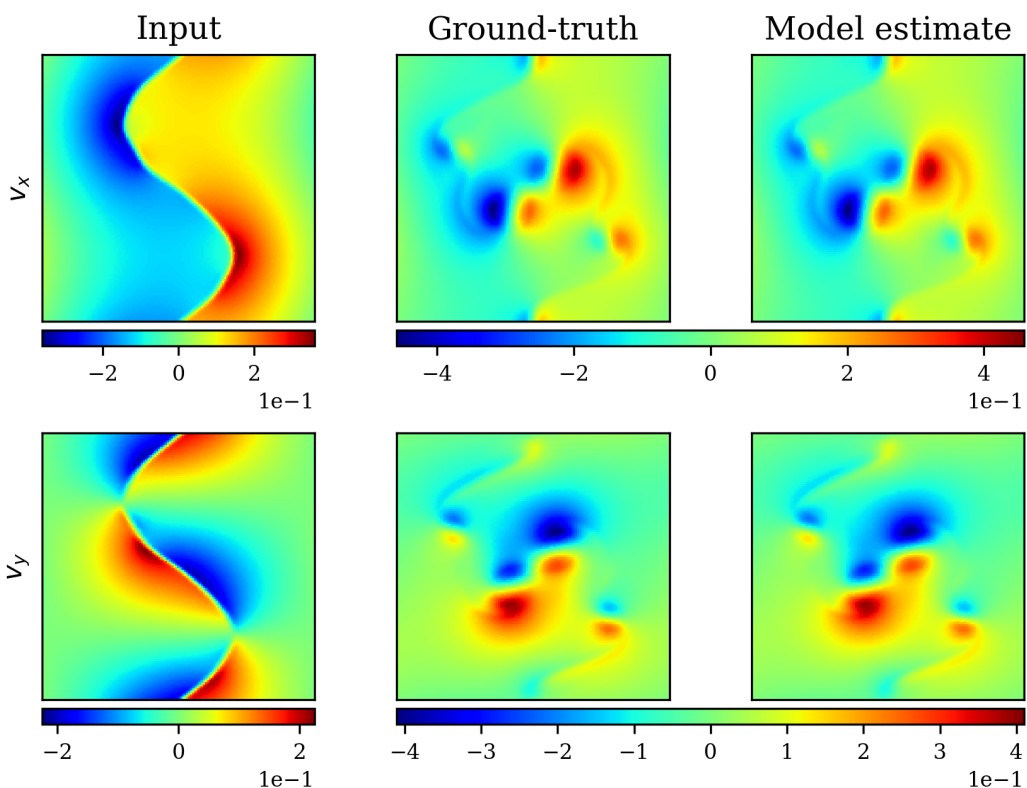

Figure G.25: Model input at $t = t_0$, ground-truth solution and model estimate at $t = t_{14}$ of a test sample NS-SVS dataset.

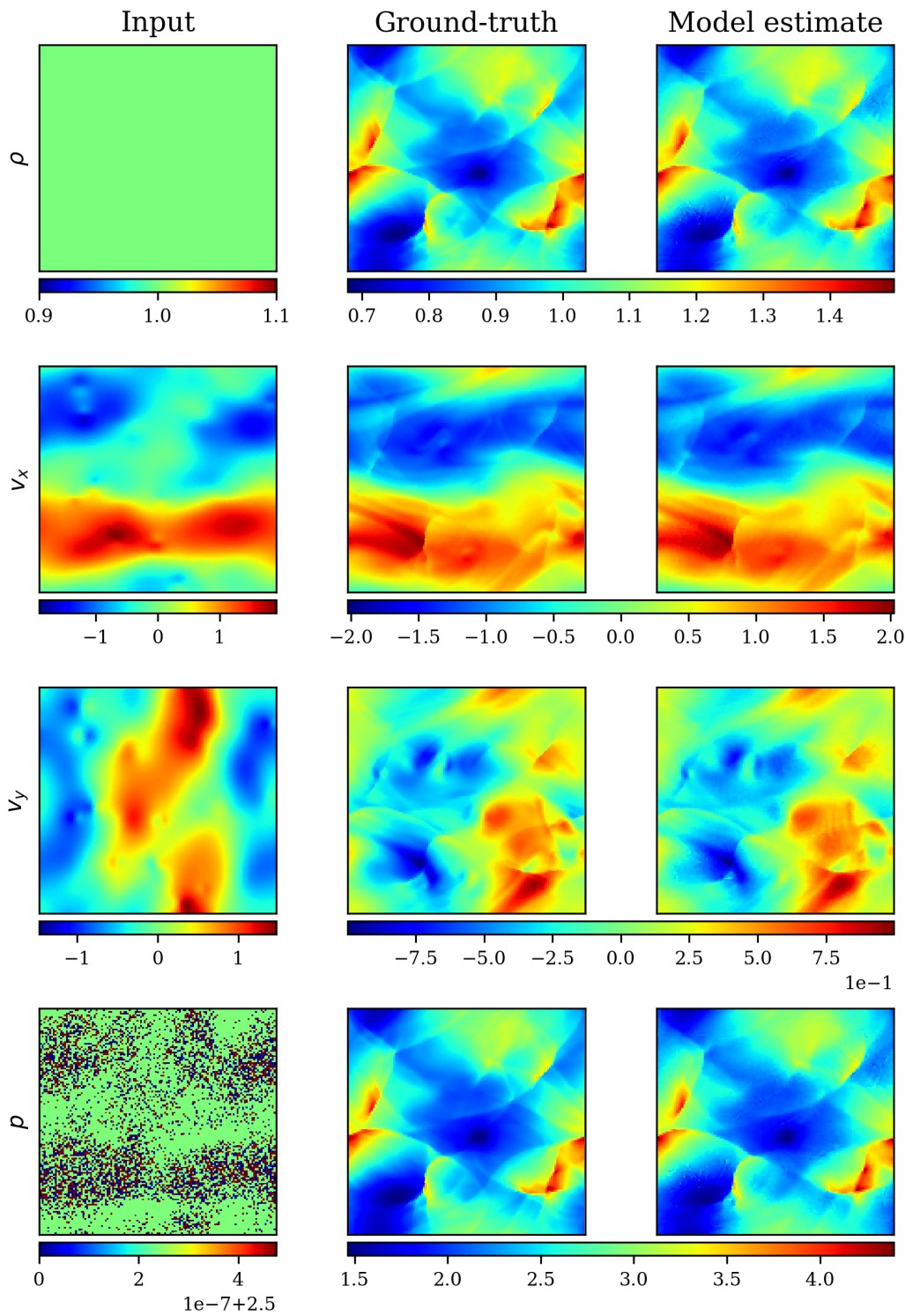

Figure G.26: Model input at $t = t_0$, ground-truth solution and model estimate at $t = t_{14}$ of a test sample CE-Gauss dataset.

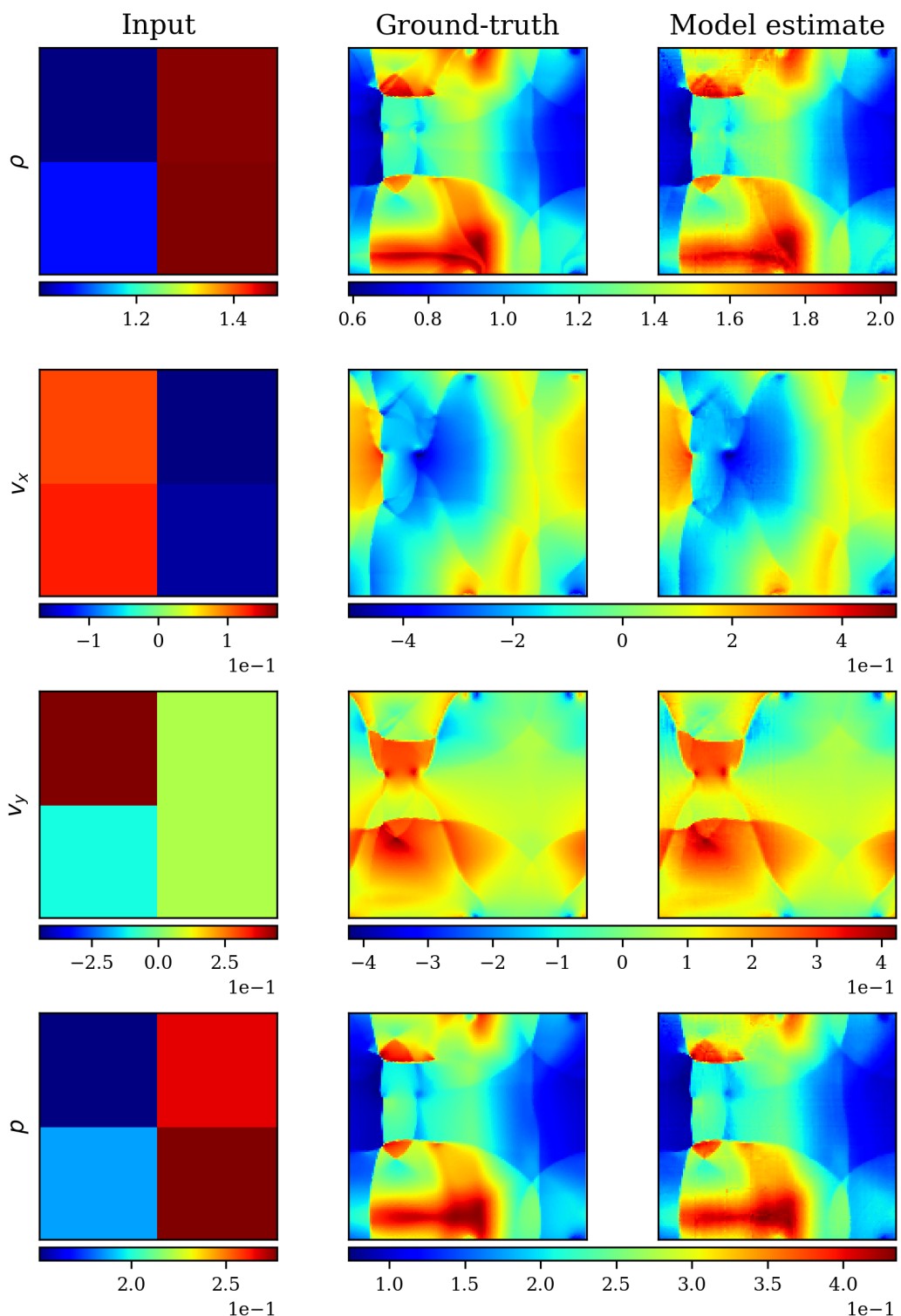

Figure G.27: Model input at $t = t_0$, ground-truth solution and model estimate at $t = t_{14}$ of a test sample CE-RP dataset.

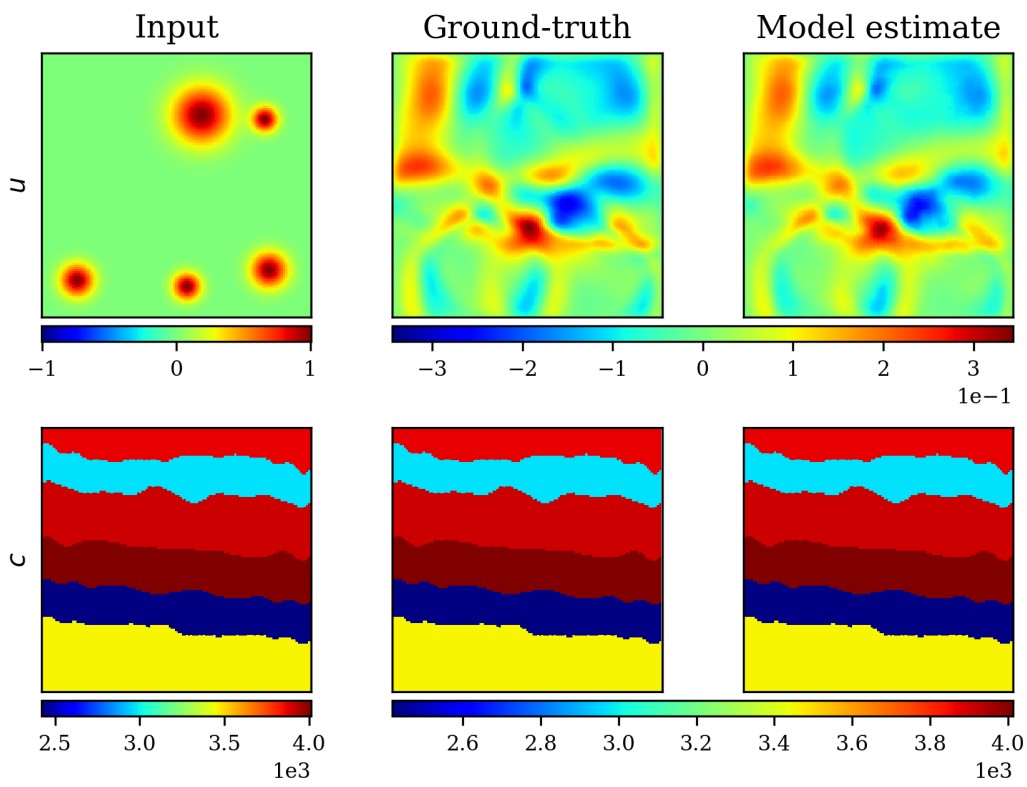

Figure G.28: Model input at $t = t_0$, ground-truth solution and model estimate at $t = t_{14}$ of a test sample Wave-Layer dataset.

