# OpenReview forum: "Geometry Aware Operator Transformer as an efficient and accurate neural surrogate for PDEs on arbitrary domains"
_NeurIPS.cc/2025/Conference — NeurIPS 2025 poster_

### Official Review · Reviewer_Kj85 · 2025-06-26

**Clarity:** 3
**Significance:** 3
**Originality:** 2
**Rating:** 4
**Confidence:** 4

**Summary:**

The proposed paper presents a novel architecture for PDE solving on arbitrary geometries. The model is built-upon on a combination of several architectures (GINO, ViT…). The novelty lies in an improved multiscale GINO encoder with geometry embeddings. The model is evaluated on a lot of datasets and compared to several baselines.

**Questions:**

-	As the claim of the model is its scalability, I couldn’t find a comparison of the training time required for the proposed method wrt the baselines. This is also an important indicator. I found the training throughput (baseline) and the training time for one batch (but only for GAOT).
-	Does your time stepping strategy allows for irregular queries in the time dimension ?
-	Line 203  is stated that « the computational burden should fall on the processor ». Could you elaborate on this aspect? Do you have experiments/explaination about this ? If so, why using encoder/decoder ? Isn’t it usefull to compress information to make model more compact and thus more scalable for real-world applications?
-	How would behave the ViT processor (let’s say on a regular grid dataset) with and without the encoder/decoder proposed ? My point here is to isolate the improvement from the encoder/decoder.
-	Have you considered to evaluate the performances of the AE alone ? since the contribution relies strongly on this part of the model, I think you should evaluate reconstruction performances compared to other baselines.

**Ethical Concerns:**

["NO or VERY MINOR ethics concerns only"]

**Final Justification:**

The main contribution of the paper lies in the cost–accuracy trade-off achieved by GAOT, which results from the careful combination and optimization/engineering of several existing architectures. The method is evaluated on a wide range of standard datasets, and ablation studies are included to empirically demonstrate the properties and effectiveness of GAOT.
Several concerns were addressed during the rebuttal, more specifically adjusting some comparisons with existing methods [1, 2, 3, 9], as well as additional ablations to isolate the contribution of GAOT's components [5, 7, 8].
To conclude, with the suggestions and clarifications provided in the rebuttal, I decided to raise my score to 4.

**Limitations:**

Yes a limitation section is added. However, I only see discussions about extensions and future works in this part. What limitations has GAOT?

**Quality:**

3

**Strengths And Weaknesses:**

### Strength
 -	The numerous experiments: the model is evaluated on a lot of datasets, in several settings (regular/irregular grids, time dependant and independant PDEs…)
 -	The paper is well-writen and despite the rich content, the reader still understands the method
 -	The paper proposes several interesting studies and ablations on the proposed model (time stepping strategies, discretisation…).

### Weaknesses
 -	Novelty : the architecture is built upon several well-known components : GINO+ViT.
 -	Limited related work : the related work is only a paragraph at the end of the paper. Due to the numerous existing references, I think this part is important in a paper. The related work can be added in the appendix, when limited space is available in the main part. This could help the reader to clearly see the similarities and differences with other existing methods. Some references are provided in the introduction, but no clear positionnig of the paper is provided.
 -	Missing references : a few references are missing in my opinion eg [1, 2, 3]. [1] : While you discuss about UPT and transolver, you are missing AROMA which is also using transformers+neural field model. [2] : As the purpose of the model is to scale on large geometries, I think that you should consider baselines with similar objectives, which is the case of Transolver ++, an extension of transolver that is design to handle large geometries (and compares on DrivAernet ++). [3] : Matching irregular to regular grids for the encoder has been studied is several models such as Text2PDE, that makes use of a GINO + CNN encoder/decoder architecture.
- Finally, I couldn't identify which part of the model greatly helps in reducing the computation cost. Compared to GINO, what explains that GAOT is faster?


[1] AROMA: Preserving Spatial Structure for Latent PDE Modeling with Local Neural Fields, Louis Serrano, Thomas X Wang, Etienne Le Naour, Jean-Noël Vittaut, Patrick Gallinari, NeurIPS 2024

[2] Transolver++: An Accurate Neural Solver for PDEs on Million-Scale Geometries, Huakun Luo, Haixu Wu, Hang Zhou, Lanxiang Xing, Yichen Di, Jianmin Wang, Mingsheng Long, ICML 2025

[3] Text2PDE: Latent Diffusion Models for Accessible Physics Simulation, Anthony Zhou, Zijie Li, Michael Schneier, John R Buchanan Jr, Amir Barati Farimani, ICLR 2024

---

> ### Author Rebuttal · Authors · 2025-07-30
>
> We start by thanking the reviewer for their careful reading of our paper, their appreciation of its merits and their suggestions for improving it. We provide a detailed response to their points of concern below.
>
> [1.] Regarding *novelty of our model*, while the reviewer is correct that the ViT processor in GAOT is not novel nor is the encode-process-decode paradigm, we reiterate from l41-55 (Main Text)  and Fig. 1 that there is an **accuracy-efficiency** trade-off for Neural PDE surrogates and accurate GNN type models (RIGNO is state of the art in this family) are not efficient while efficient models (GINO) are not as accurate. So, our main rationale for this paper was to provide a model that could be efficient and accurate at the same time. While combining available elements, we did need to come up with 3 key *novel features* to do this: i) multiscale attentional GNO for encoding/decoding ii) geometric embeddings based on statistical descriptors to enhance geometric inputs to the model and iii) several tricks allowing for an optimized implementation (see pt. [3] below). To the best of our knowledge, these are original contributions and they clearly enhance model performance. See Tab. F.5 and reply pt [8] below, for results on how multiscale information in the encoder/decoder significantly improves upon a single-scale version, Similarly, see Tab F.4. to observe how statistical geometric  embeddings improve accuracy over just inputting coordinates and SDFs. Finally, the role of efficient implementation is described below in pt [3].
>
> [2.] We acknowledge the reviewer's point that *related work* is rather succinct, primarily due to page limits for the main text. Following the reviewer's suggestion, we are happy to elaborate on the related work section in a camera-ready version (CRV) esp. SM, if accepted. In particular, we thank the reviewer for bringing Refs. [1,3] to our attention and we will gladly add them to related work. Regarding Transolver ++, we will certainly add it with the context described in our reply pt [6.] to reviewer diAB above.
>
> [3.] To answer the reviewer's pertinent question on **why is GAOT faster than GINO** ?, we ran GINO (6.07 M params) for a single sample of the NACA0012 dataset on a NVIDIA-4090 GPU to obtain the following breakdown of compute time in ms for GINO:  Encoder (Graph  building: 1.984, GNO: 2.793, Total: 4.777), Processor (5-Layer FNO): 5.89, Decoder (Graph Build: 1.984, GNO: 1.140, Total: 3.124). Total GINO run time per sample = 13.791 ms. As it happens, **GINO does not support batch processing** when the underlying graphs change in a batch (as in general for PDEs on arbitrary domains). Thus to process a batch of 32, GINO has to loop over the entire batch, leading to a runtime of 441.32 ms ! Thus, we observe 2 computational bottlenecks here: i) Graph building in the encoder/decoder is as (or more) expensive than the GNO module and ii) lack of batch processing severely impedes computational throughput. Our challenge was to overcome these bottlenecks.
>
> To this end, we came up with the following tricks: a) we take graph building out of the GAOT model evaluation loop to reduce computational wastage by precomputing the graphs and storing them as cache files and loading with efficient data loaders and b) batch processing the transformer processor while looping over the encoder/decoder sequentially. Please see our reply pt [4] to reviewer diAB above for further details on these tricks.
>
> With these implementation tricks, we have the following compute breakdown for a 5.62M GAOT  (same setting as in GINO) for a batch size of 1: Encoder (AGNO: 2.296, GeoEmb: 3.412, Total: 5.708), 5-Layer ViT processor with patch size 2: 10.1, Decoder (AGNO:1.356, GeoEmb:1.865, Total: 3.221). Total GAOT run time per sample=19.029 ms. Hence, GAOT is slower than GINO for a batch size of 1 as the multi-scale GNO and GeoEmb embeddings add cost although Graph building has been taken out of the loop. However, the results for a batch size of 32 are: Encoder (AGNO+GeoEmb) = 126 ms, Transformer processor = 30.1 ms, Decoder (AGNO+GeoEmb) = 107 ms, leading to an overall batch compute time of 263.1 ms, which is *178 ms* faster than GINO !!. The payoff happens in the **batch processing of the transformer** as this module only increased from 10.1 ms (batch size 1) to 30.1 ms (batch size = 32). It was precisely this novel trick that allows GAOT to be more efficient than GINO, while being much more accurate. As a final trick for very large (particularly 3-D) graphs, we also introduce an edge masking procedure to further reduce memory bottlenecks in the encoder/decoder (l214). We will add this quantitative computational breakdown in the CRV to better inform the readers on how our model achieves computational efficiency.
>
> [4.] As total number of training samples in each dataset is exactly the same for all models, training time per epoch = (Total Number of Samples)/Training throughput, scales exactly as the training throughput for each dataset. Hence, GOAT is 1.6 times faster to train than GINO from Tab E.1. We can also provide the exact training times for each model in a CRV, if accepted.
>
> [5.] Regarding the reviewer's question about *irregular queries* in time, our model provides the current time t and lead time $\tau$ as inputs (l191), allowing it to be evaluated at any query point in time and at any query point in space (neural field property). To demonstrate this feature, we consider the NS-PwC dataset, where we had used training data at time stamps $t=0.0,0.1,0.2,0.3,0.4.0.5,0.6,0.7$. At test time, we output the velocity fields at times $t=0.35$ and $t=0.45$, which are not included in the training set to obtain relative % errors of 1.4 and 1.91, respectively. This interpolates very nicely with errors of 1.2 ($t=0.3$), 1.65 ($t=0.4$) and 2.2 ($t=0.5$). We thank this reviewer for this suggestion and will add the results in a CRV.
>
> [6.] Perhaps l203 was not clearly articulating our point, which was to say that we would like the compute of the processor to scale very favorably with increasing number of layers as we clearly show in the argument presented in pt [3] above. Clearly the encoder-decoder are necessary to compress information but we want them to be as lightweight computationally as possible as they don't scale as well as the processors. We will adjust this sentence accordingly in a CRV, if accepted.
>
> [7.] The reviewer's aim to understand the precise role of encoder/decoder vs. processor in our model is very much appreciated. We had infact presented results on a Cartesian grid in SM Tab E.2, where we compared GOAT to standard Cartesian grids models such FNO, CNO and the SWIN transformer based scOT, to observe that GAOT was clearly more (or as) accurate as the dedicated cartesian grid architectures on problems on Cartesian grids. Following the reviewer's excellent suggestion, we ablated GAOT further by just running the ViT processor (without the encoder/decoder) on these Cartesian datasets to obtain the following relative % errors: NS-Gauss (ViT (3.16) vs GAOT (2.29)), NS-PwC (ViT (3.89) vs. GAOT (1.23)), NS-SL (ViT (0.73) vs. GAOT (0.98)), NS-SVS (ViT (0.4) vs. GAOT (0.46)), CE-Gauss (ViT (6.81) vs. GAOT (5.28)), Wave-Layer (ViT (5.48) vs. GAOT (5.40)), Poisson Gauss (ViT (2.47) vs. GAOT (0.79)). From these results, it is clear that in some cases, ViT is comparable to or slightly better than GAOT, indicating that the power of GAOT in these cases stemmed from a good processor. However in many more cases, **GAOT is significantly superior to ViT** also showing that the encoder/decoder contribute significantly to expressivity and together, this combination achieves SOTA performance. This interesting experiment clearly delineates the relative contributions of encoder/decoder vs. processor and we thank the reviewer again for asking us to perform it. Needless to say, we will include it in a CRV, if accepted.
>
> [8.] To answer the reviewer's interesting question about if the GAOT encoder-decoder can work as *autoencoders* (AE), we consider the output distribution of the NACA0012 compressible flow dataset. As shown in SM Fig. G.4, second column from left, this is a challenging distribution to learn as it contains sharp shocks separated by smooth regions in the flow and there is considerable variation when the increasing Mach number transitions the flow from sub-sonic to supersonic. Thus, it is worthwhile to see how the MAGNO + geometric embeddings based encoder-decoder of GAOT fairs as an AE. As a baseline, we chose GINO with its GNO encoder/decoder as the only point of difference now is the multi-scale attentional features in MAGNO compared to the single-scale GNO and the geometric embeddings. The resulting relative L1 reconstruction errors are 0.08 (GAOT) vs 0,11 (GINO) showing that the MAGNO encoder-decoder+ geometric embeddings are clearly more powerful than a vanilla GNO. We had already shown the advantage of MAGNO in SM Tab F.5 and geometric embeddings in SM Tab F.4 and this experiments further reinforces these advantages. We thank the reviewer for this interesting suggestion and will be happy to add this point to a CRV.
>
> [9.] We don't see any empirical limitations with GAOT given its very broad scope, scalability and excellent performance. However,, as we wrote on l364, we have not shown any rigorous theoretical results such as an universal approximation property etc. This is a limitation that we hope to address in future work.
>
> We take this opportunity to thank the reviewer again and hope that we have satisfactorily addressed their concerns, particularly regarding the computational efficiency of GAOT and the contribution of individual elements (encoder/decoder vs. processor). If so, we request the reviewer to kindly upgrade their assessment accordingly, while remaining at your disposal to answer any further questions about our paper.

---

> > ### Comment · Reviewer_Kj85 · 2025-08-04
> > **Answer to Authors' Rebuttal**
> >
> > Thanks for the answer to my questions, and the additional experiments, especially [4.] to [8.] introduce new experiments and explanations that helps identifying the benefits of GAOT. After reading the rebuttal, I now have the following additional questions and remarks:
> >
> > -	[3.] : Thanks for this detailed explanation regarding the computationnal efficiency of GAOT. This helps clarify which parts effectively improve this aspect. However, I am not fully convinced by the decision to exclude the graph computation time from GAOT’s inference time, while keeping it for GINO. While I understand that removing graph construction during training can significantly speed up computation and is an interesting trick of GAOT, at inference time, the situation is different. In a real-world use case, one typically does not have access to a precomputed graph for a new instance. Therefore, the graph computation would likely need to be performed on the fly, and should be included in the reported inference time. Alternatively, if this step is excluded for GAOT, it should also be excluded for GINO to allow a fair comparison - especially since graph construction appears to be a computational bottleneck for both methods.
> > -	[9.] : I am a bit surprised to read that no limitations exists for GAOT. I believe that in any model there is still room for improvement in terms of performance or generalization… Here are some ideas that are generally discussed : What are the potential directions for future work? For instance, does GAOT need to be retrained for each new task, such as forward or inverse problems? Can it handle multi-physics scenarios? These are high-level questions, but I think addressing them, even briefly, would help positioning the paper wrt current research directions.

---

> > > ### Author Response · Authors · 2025-08-04
> > > **Response to the Reviewer**
> > >
> > > We start by thanking the reviewer for your prompt response and for appreciating some points in our rebuttal. We provide a detailed reply to your open questions and comments below:
> > >
> > > [1.] The reviewer certainly has a point about being consistent with reporting computational time in terms of adding/not adding the graph building modules. We added graph building for GINO as it is an intrinsic part of its source code. However, we had already provided you with a detailed break-up of the computational cost of GINO earlier which includes a graph building time of 3.97 ms of graph building per sample. So in a batch of 32, the total graph building time will be 127 ms. We can subtract it from the total GINO runtime of 441ms leading to an adjusted runtime of 314 ms for GINO. We recall that the GAOT runtime for the batch of 32 is 263 ms, which is still faster than GINO. In fact, this calculation further showcases the benefit of batch processing. Our main point remains that GAOT can be competitive with respect to GINO, when it comes to computational efficiency and scalability. At the same time, this fact has to be viewed in conjunction with the significant gain in accuracy that GAOT provides vis a vis GINO (Table 1). Form this table, we observe that GAOT is 3.27x more accurate than GINO on an average over the 15 datasets recorded there. So, even matching the computational efficiency of GINO provides a big payoff for GAOT. Nevertheless, we will be consistent in our reportage of the computational breakdown of GAOT vs. GINO in a CRV, if accepted.
> > >
> > > [2.] We apologize to the reviewer for possibly misunderstanding your question about *limitations* as we interpreted it as asking for empirical limitations of our model. Clearly, it was not intended to say that GAOT has no intrinsic limitations. In fact, we had already acknowledged as such in the Main Text l356-365 where we have clearly stated that our current work is restricted to learning the *forward* solution operator of PDEs on arbitrary domains. It does not cover Inverse problems (l362) for instance nor have we tried with physics informed loss functions as in Physics-informed neural operators. Regarding the reviewer's point about multiphysics, we have also acknowledged this limitation in l357-358 by saying that GAOT can be used as the backbone of a Foundation model such as Poseidon or DPOT, which deal with multiphysics training. Our results on a transfer learning experiment (Fig. 3c and SM E.6) indicate the potential for GAOT in such a setting. We have clearly sketched out that these current limitations will be basis of future work (l365) besides also acknowledging the lack of theoretical results in our paper. We are happy to expand on these themes further in a a CRV, if accepted.
> > >
> > > We thank the reviewers again for your questions and comments and hope to have addressed them to your satisfaction. If so, we would kindly request the reviewer to upgrade their assessment accordingly. We remain at your disposal for further questions.

---

> ### Comment · Reviewer_Kj85 · 2025-08-05
> **Answer to Authors' Rebuttal (2)**
>
> I thank the authors for their answer.
>
> -	Regarding [1.] : Thanks for the updated results. These should be consistently reported in the manuscript to avoid any misleading for the reader. In my opinion, inference time should be reported in an end-to-end manner, reflecting the realistic scenario of deploying the model on new samples. The detailed inference time is also interesting as it illustrates some of the claims of the paper.
> -	Regarding limitations, thanks for your answer. I think the section l356-365 reads more like a discussion of future work rather than a clear articulation of the method’s limitations, which is why I raised the question.
>
> To conclude, with the suggestions and clarifications provided in the rebuttal, I will raise my score to 4. I am particularly refering to points 1, 2, 3, 9, which contribute to an honest and rigorous comparison of the proposed method with existing ones (in the related work as well as in the experiments). Additionally, the new experiments presented (e.g., points 5 and 7) help illustrate and support some of the properties and claims of the model.

---

> > ### Author Response · Authors · 2025-08-05
> > **Thanking the Reviewer**
> >
> > We thank the reviewer for your comments and really appreciate increasing our score. If accepted, we will certainly present consistent inference and training times as well as clearly delineate limitations as we have presented in our last response. The other points mentioned in our detailed rebuttal will be added to a CRV, if accepted.

---

### Official Review · Reviewer_diAB · 2025-06-30

**Clarity:** 2
**Significance:** 3
**Originality:** 2
**Rating:** 4
**Confidence:** 4

**Summary:**

This paper presents GAOT, a novel neural neural operator approach for solving time-dependent and independent PDEs on arbitrary domains. The authors proposed a new transformer approach for solving PDEs in an efficient way, when dealing with PDEs discretized over dense grids. Notably, they propose a new multi-scale approach inspired by GINO for encoding arbitrary point clouds into a regular (or not) latent grid. Their encoding and geometry embedding approach allows them to patch and thus reduce the number of physical tokens, reducing the computational cost for their transformer that does the time-stepping operation, trained in a all2all fashion. They notably evaluated their approach on a large number of challenging datasets, with time-dependent and independent problems with regular and irregular grids. Through their experiments, they exhibit competitive or sota performance while remaining computationally effective.

**Questions:**

In the contributions, you mention that you use a *coarser* latent grid. In the appendix, it is mentioned that the size is $64 \times 64$. It is therefore a relevant compression rate when dealing with PDEs discretized over a $128 \times 128$ grid, but for denser grids, how restrictive is this choice? In other words, how does the latent grid size influence the performance of the model?

i am a bit concerned by the training throughput results you have presented. From my knowledge, GINO encoding is particularly costly and needs to proceed each sample observation one by one. This seems to be costly as you cannot vectorize the operation. In addition, you also different scales, which means you have to also add a new loop depending on the number of scales considered. I don't really see how you still manage to be more efficient to other NO approaches.

For the Transolver and UPT, you mentioned that it was too expensive to train it an all2all fashion. I don't really see why it is that expensive? For instance, for UPT, you can control the number of super-nodes to consider and adjust it such that the number of tokens match the number of tokens used for GAOT no?

For GAOT, if I am correct, you embed physical fields into a 2D latent grid of shape $64 \times 64$ so this means after the patchify operation (2 according to appendix) you get 1024 tokens. In UPT, you said you use 2048 super-nodes. Decreasing the number of super-nodes wouldn't make it work for time-dependent data?

The same could apply for Transolver where we could decrease the number of slices, I guess. Also an extension has been proposed to tackle problems with denser grids [3]. I think it could be mentioned somewhere as it relates to neural solver efficiency.

[3] Luo et al, Transolver++: An Accurate Neural Solver for PDEs on Million-Scale Geometries, arXiv preprint arXiv:2405.13998

**Ethical Concerns:**

["NO or VERY MINOR ethics concerns only"]

**Final Justification:**

I think the authors provided an interesting approach that is both computationally efficient and very accurate. The authors answered some of concerns and recognized that the computational efficiency of their method is not specifically due to the model architecture but to relatively straightforward implementation tricks that could be used on GINO.

In its current form, the paper's contributions are a bit over-claimed to me, preventing me to put a higher score and only recommend a borderline accept.

Still, I think the paper could be accepted, because the experiments are very consequent and the method very effective. If they provide the revision needed, it can be a good paper valuable for the neural operator literature.

**Limitations:**

Limitations have been addressed. I might have miss the information as they mentioned they addressed negative social impacts but I did not find any broader impact section.

**Paper Formatting Concerns:**

No formatting issues.

**Quality:**

3

**Strengths And Weaknesses:**

The paper's quality is good and the experiments clearly highlight the benefits of GAOT over existing baselines. The number of experiments, datasets and baselines considered is huge. They proposed extensive ablations in the appendix and provide a lot of technical details to explain their approach in the SM.

While learning neural operators is a well-known research direction, GAOT seems to be a new effective solution for solving PDEs in a cost-effective way while still being more accurate. They notably compared the computational costs of their method with respect to existing baselines when increasing the number of model parameters, and provided experimental results showing that GAOT seems faster to be trained and to infer new PDE solutions.

This might not be considered as an inconvenient to some readers/reviewers, but I feel like the main paper is just a succession of small descriptions of the method, but it is very difficult to get a correct understanding from it. Most of the technical details are actually in the SM which is very large.

Regarding the novelty of the paper, it comes from the multi-scale approach that extends GINO and the statistical embeddings used for better encoding geometries. The processor is a classical vision transformer with a patchify operation as in [1] with an all2all approach as done in Poseidon [2].

[1] Wang et al, CViT: Continuous Vision Transformer for Operator Learning, ICLR 2025.
[2] Herde et al, Poseidon: Efficient Foundation Models for PDEs, NeurIPs 2024.

---

> ### Author Rebuttal · Authors · 2025-07-30
>
> We start by thanking the reviewer for their careful reading of our paper, their appreciation of its merits and their suggestions for improving it. We provide a detailed response to their points of concern below.
>
> [1.] Regarding the reviewer's point about *clarity of exposition*, we would be happy to significantly elaborate on the overview section (l102-109) in the main text in a camera-ready version (CRV), if accepted. This will allow us to better inform the reader on how our model is designed.
>
> [2.] We will add a reference to CVIT paper in a CRV as it is indeed relevant. We agree with the reviewer that the *multiscale encoder-decoder* and *statistical geometric embeddings* are our main novel contributions at the level of model design.
>
> [3.] On the reviewer's question about *different choices of the latent grid*, we would like to point to SM F.2 (l1344-1358), in particular Tab. F.2, where we have precisely explored this question. From this table, we observe that reducing the latent grid size does affect the overall accuracy to a minor extent, but not in any significant manner, particularly for the correct patch sizes. This is consistent with our heuristic that the main point of the latent grid should be to provide adequate coverage of the physical domain. As long as it holds, even coarser latent grids are admissible. Summarizing, GAOT seems to be very robust to varying latent grid size, radii of aggregation and patch size. We will highlight this point further in a CRV, if accepted.
>
> [4.]  The reviewer is absolutely spot on questioning the *computational costs of GINO*. To further quantify your concerns, we ran GINO (6.07 M params) for the NACA0012 dataset on a NVIDIA-4090 GPU to obtain the following breakdown of compute time in ms for GINO for a single sample:  Encoder (Graph  building: 1.984, GNO: 2.793, Total: 4.777), Processor (5-Layer FNO): 5.89, Decoder (Graph Build: 1.984, GNO: 1.140, Total: 3.124). Total GINO run time per sample = 13.791 ms. However, as the reviewer very astutely pointed out, **GINO does not support batch processing** when the underlying graphs change in a batch (as in general for PDEs on arbitrary domains). Thus to process a batch of 32, GINO has to loop over the entire batch, leading to a runtime of 441.32 ms ! Thus, we observe 2 computational bottlenecks here: i) Graph building in the encoder/decoder is as (or more) expensive than the GNO module and ii) lack of batch processing severely impedes computational throughput. Our challenge was to overcome these bottlenecks.
>
> To this end, we came up with the following tricks: First, we take graph building out of the GAOT model evaluation loop to reduce computational wastage by precomputing the graphs and storing them as cache files and loading with efficient data loaders. Next to enable batch processing, initially we explored vectorizing the encoder and decoder operations. However, this approach proved impractical for two key reasons. First, the grid sizes within a batch may exhibit significant variations, complicating efficient vectorization. Second, for relatively large graphs, vectorization introduces an operation that triggers a transient spike in memory consumption, which can cause the overflow of GPU memory. Consequently, straightforward vectorization is still neither memory-efficient nor scalable. To address this, we adopted a sequential processing strategy for the encoder and decoder while batch processing the transformer. Importantly, this only introduces a lightweight loop whose overhead is negligible compared to the transformer processor, ensuring that it does not become a computational bottleneck when the processor is scaled.
>
> With these implementation tricks, we have the following compute breakdown for a 5.62M GAOT  (same setting as in GINO) for a batch size of 1: Encoder (AGNO: 2.296, GeoEmb: 3.412, Total: 5.708), 5-Layer ViT processor with patch size 2: 10.1, Decoder (AGNO:1.356, GeoEmb:1.865, Total: 3.221). Total GAOT run time per sample=19.029 ms. Thus, as the reviewer anticipated GAOT is slower than GINO for a batch size of 1 as the multi-scale GNO and GeoEmb embeddings add cost although Graph building has been taken out of the loop. However, the results for a batch size of 32 are: Encoder (AGNO+GeoEmb) = 126 ms, Transformer processor = 30.1 ms, Decoder (AGNO+GeoEmb) = 107 ms, leading to an overall batch compute time of 263.1 ms, which is *178 ms* faster than GINO !!. The payoff happens in the **batch processing of the transformer** as this module only increased from 10.1 ms (batch size 1) to 30.1 ms (batch size = 32). It was precisely this novel trick that allows GAOT to be more efficient than GINO, while being much more accurate.
>
> Moreover, this set of implementation tricks become even more impactful when the input size and model size is scaled up. As this happens, the memory bottleneck in the encoder/decoder and the compute usage in the processor grows proportionately. Consequently, moving graph building outside the encoder/decoder and batch processing the transformer processor makes GAOT scale significantly better than GINO in both input size and model size as we observe from Figs. 3a and 3b.
>
> As a final trick for very large (particularly 3-D) graphs, we also introduce an edge masking procedure to further reduce memory bottlenecks in the encoder/decoder (l214).
>
> Following the reviewer's excellent suggestion, we will elaborate on this point in a CRV and provide the above detailed computational breakdown such that readers can better appreciate the effort we put into optimizing our implementation for efficiency and scalability.
>
> [5.] As the reviewer correctly claims, UPT is a scalable model. Our problem with UPT is that it lacked stability while training, making convergence very difficult, if not impossible. Despite considerable efforts, we could not make UPT (particularly for larger model sizes) converge for many for our time-dependent datasets. Hence, we did not present these results in the main text. However, we obtained the following relative % errors for the time-dependent datasets of Tab. 1 : NS-Gauss (did not converge (UPT) vs. 2.91 (GAOT)), NS-PwC (did not converge (UPT) vs. 1.5 (GAOT)), NS-SL (51.5 (UPT) vs. 1.21 (GAOT)), NS-SVS (4.20 (UPT) vs. 0.46 (GAOT)), CE-Gauss (64.2 (UPT) vs. 6.4 (GAOT)), CE-RP (26.8 (UPT) vs. 5.97 (GAOT)), Wave-Layer (19.6 (UPT) vs. 5.78 (GAOT)), Wave-C-Sines (12.7 (UPT) vs. 4.65 (GAOT)). Together with the time-independent results from Tab. 1, we see that GOAT is often one to two orders of magnitude more accurate than UPT.  We are happy to add these UPT results in a CRV, if accepted.
>
> [6.] Regarding Transolver, we found this model to be very slow computationally. In our understanding, this is because the Transolver model, although it also adopts an encoder–processor–decoder architecture, this architecture is applied at the level of a single Transolver block and there is no sequential processing of information in the latent space. As a result, when stacking many Transolver blocks, the model repeatedly executes both the encoder and the decoder. Since the encoder and decoder each time need to establish mappings between the input grid size and the latent tokens, this leads to a substantial amount of computation (as input grids are typically large) and significant memory overhead. Moreover, the computations in the encoder and decoder cannot fully exploit the hardware’s throughput capabilities, and the large memory footprint forces us to use very small batch sizes as soon as the input grid size becomes slightly larger. Consequently, the model struggles to achieve high throughput as we observe from Figs 3a and 3b. Using the original setup described in the Transolver paper, we found it nearly infeasible to train time-dependent datasets (each with around 20,000 samples) in an all-to-all manner on our hardware.
>
> Transolver++ was published concurrently with the submission of our article, preventing us from using it as a baseline. Although Transolver++ also benchmarks on the Drivaernet++ dataset, their model is trained with only 200 of the total of 8000 samples, making it impossible for us to compare with their results as more data will surely lead to more accuracy. Instead, we have compared with models that sit on top of the Drivaernet++ leaderboard to emerge as the best performing model for this entire dataset.  Nevertheless, we do agree with the reviewer that Transolver++ is a contribution to neural PDE surrogate efficiency and we will refer it in a  CRV, if accepted.
>
> We take this opportunity to thank the reviewer again and hope that we have satisfactorily addressed their concerns, particularly regarding the computational efficiency of GAOT and its comparison with UPT and Transolver. If so, we request the reviewer to kindly upgrade their assessment accordingly, while remaining at your disposal to answer any further questions about our paper.

---

> > ### Comment · Reviewer_diAB · 2025-08-03
> >
> > I would like to thanks the authors for addressing my questions and providing a detailed rebuttal response.
> >
> > Regarding clarity, indeed, it would be nice to clearly describe in the main section the complete architecture considered in the principal experiments. For instance, for the choice of latent domain, if I am correct main results have been tested with a regular latent domain; maybe it would be valuable to explain this strategy in more detail (and refer to SM for alternative strategies).
> >
> > Indeed CVIT is a relevant related work in my opinion. Maybe in (l65), you could change your contribution and be more precise, as CVIT is the first (to my knowledge) to patch physical systems. Their method is however restricted to regular grids, unlike yours.
> >
> > Thanks for pointing out to F.2. Maybe my question wasn’t clear enough, but I was actually referring to denser grids. You tested it on cases with $128 \times 128$ points, thus the latent grids you considered are still reasonable (in terms of compression ratio). On cases with higher number of input points in the physical space, does GAOT need to encode physical data into a denser latent representation, or a coarse representation still yield good performance?
> >
> > Regarding the computational efficiency, I am not sure to fully understand. Building the graph is part of the drawbacks of considering a graph approach and should be considered in the computational cost, and if not, it could also be removed from GINO right? Then for the batch processing with the transformer, this is very basic property shared by all neural network architectures no? GINO also processes data by batches in their process (FNO). Can you elaborate a bit more on that?
> >
> > For UPT, I might have missed the information in the main text, but if it not written, maybe add a small explanation somewhere about the convergence issues you faced when evaluating their model.
> >
> > Thanks for the clarification on the Transolver approach. For Transolver++, I was not expecting you to test it as a baseline as it is indeed a very recent paper, a reference is enough to me.

---

> > > ### Author Response · Authors · 2025-08-04
> > > **Response to the Reviewer**
> > >
> > > We starting by thanking the reviewer for your prompt response and reply below to your remaining questions and comments:
> > >
> > > [1.] If accepted, in the CRV, we will have an extra page and will reorganize the main sections of the paper, including a discussion on results for regular grids.
> > >
> > > [2.] We will cite the CVIT paper and provide appropriate context for it by adjusting the statement on l65 accordingly.
> > >
> > > [3.] Regarding the reviewer's question about *dense grids and compression ratios*, in SM Table F.2, for a (original) physical input grid of $128^2$, we have tested on a latent grid of $32^2$, leading to a compression ratio of 6.25% to obtain very good results, even with this compression. However, we concede the point that, in practice, one could encounter much denser
> > > input grids. As it happened, the input grid for the Drivaernet++ benchmark has 500K points (approximately) per input sample. We compress this very dense and unstructured input into a regular latent grid of size $64 \times 32 \times 32$ or 65K points, leading to a compression ratio of 13%. As Table 3 shows, GAOT is the best performing model on this challenging benchmark, when compared to the SOTA baselines from the leaderboard. That this happens, despite a very dense input grid and large compression ratio, further demonstrates the ability of GAOT to process complex learning tasks. We will provide further clarification on this issue in a CRV, if accepted.
> > >
> > > [4.] We start by again acknowledging that GINO is a very efficient architecture and we have mentioned that multiple times in our paper (see Figure 1 and discussion next to it). We also agree that efficient techniques such as pre-cached graph building, batch processing and edge masking could also be applied to GINO (and other baselines) to further improve its efficiency. However, we maintain that these techniques have not been used for instance in GINO. For instance,  the current GINO code repository (https://github.com/neuraloperator/neuraloperator/blob/main/neuralop/models/gino.py, lines 403–406), explicitly states that *the current GINO implementation does not support batch training when the geometries differ across samples within a batch*. Therefore, when we run our experiments with the latest publicly available GINO code, we obtained the scalability results shown in Figures 3a and 3b, which demonstrate that GAOT is more efficient and scales better than GINO.
> > >
> > > Although our efficient implementations may appear to be an obvious trick, our approach was determined after a detailed profiling of efficiency and memory bottlenecks, and it proved to be the optimal solution. As mentioned in our previous response, we initially attempted vectorization on both the encoder and decoder, but found that the efficiency remained limited. Therefore, we consider this as a contribution worth highlighting in the paper. Given the extensive feedback from the reviewer, we will properly contextualize our contribution and highlight the fact that the efficient implementation tricks can be extended to other baselines, including GINO, in a CRV, if accepted.
> > >
> > > [5.] Following your feedback, we will discuss the limitations of the UPT model training in the CRV, if accepted.
> > >
> > > [6.] We will include a citation to Transolver++ in the CRV.
> > >
> > > We take this opportunity to thank the reviewer again for their response. We hope to have answered your latest round of questions to your satisfaction. If so, please consider upgrading your assessment of our paper accordingly.

---

> > > > ### Comment · Reviewer_diAB · 2025-08-05
> > > >
> > > > I would like to thanks the authors for their response. Some of my concerns have been addressed, but my main concern regarding computational efficiency remain.
> > > >
> > > > I understand your point and I also agree that finding these bottlenecks and providing an efficient solution is worth to be noted as a valuable contribution. But as it is, the contribution seems a bit over-claimed to me, as I indeed feel the proposed tricks are relatively straightforward. I believe GINO could potentially be faster than GAOT by applying the same tricks. The method still leads to sota performance on a lot of datasets, so I think GAOT is a strong and effective model for PDE solving.
> > > >
> > > > The paper's novelty being relatively modest, but the experiments consequent, I will keep my score to 4, corresponding to a borderline accept.
> > > >
> > > > I am still open to engage in further discussion with the authors if needed.

---

> ### Author Response · Authors · 2025-08-06
> **Further Response to the Reviewer Part #1 (Computational Efficiency and Accuracy)**
>
> We start by thanking the reviewer for your comments and sincerely appreciate the possibility to engage in further discussion. We thoroughly respect the reviewer's opinion and score but take this opportunity to reply to your comment as it pertains to the very core of our contributions, particularly in terms of novelty. We take the liberty to divide this response into two parts, the first one focussed on computational efficiency and the second on novelty of design. We apologize in advance for the long response and request the reviewer's patience in reading it.
>
> [1.] We agree that applying **our** implementation tricks to GINO is going to increase its speed and possibly make it faster than GAOT although it is not the case with current implementations. Moreover, we did not find these tricks elsewhere in the literature.
>
> [2.] However, what is undeniable is the striking gain in accuracy of GAOT over GINO. This is clearly demonstrated in Table 1 where we have compared GAOT and GINO over 15 PDE learning tasks to find that the **relative error with GAOT is, on an average, 3.27x lower than GINO**. Furthermore, GAOT is more accurate than GINO in **every single benchmark that we tested on**. These include not only datasets generated by us (compressible flows past objects and Poisson-C-Sines), but also publicly available datasets released in other papers, including the elasticity benchmark which was proposed in Ref. [26] by  many of the authors of the GINO paper. On this dataset, GAOT's error is 3.26x times lower than GINO's, which is exactly the average gain in accuracy of GAOT over GINO. Even if someone makes GINO faster than GAOT, it is very unlikely that the *significant gains in accuracy with GAOT can be compensated by a proportionate gain in computational efficiency of GINO over GAOT.*
>
> [3.] Moreover, we still have degrees of freedom to **further increase the computational efficiency of GAOT**. One of them is to increase the patch size of the VIT processor. The current default patch size is 2 (SM Table B.2) and all the quantitative metrics that we provided about GAOT (GAOT-2 hereon) are with this patch size. However, we have also considered a version of GAOT with a patch size of 8, calling it GAOT-8. The corresponding scaling plots comparing training throughput and peak memory to input grid size and model size are provided in Figure E.1. To provide some concrete numbers, in the training time example that we detailed earlier, while the encoder and decoder runtimes with GAOT-8 are identical as that of GAOT-2, the processor is faster. For a batch size of 1, it takes 8.2 ms  (compared to 10.1 ms for GAOT-2), but for a batch size of 32, the ViT processor of GAOT-8 takes only 9.8 ms to run (compared to 30.1 ms for GAOT-2), so total runtime for GAOT-8 is 20ms faster than GAOT-2 in this case.
>
> However, as you have pointed out, it is imperative to fairly compare GAOT and GINO for computational efficiency. As GOAT-8 is our fastest model, we do this comparison in terms of the training throughout for a dataset (Poisson-Gauss) where the underlying graphs do not change in a batch and GINO supports batch processing in this case. We have also implemented storing graphs on Cache for GINO and taking graph building out of the runtime loop. So, this is a totally fair apples to apples comparison. We observe that GAOT-8 has a training throughput of 957 samples/sec compared to GINO's throughput of 753 samples/sec in this setting.
>
> Hence, we observe that GAOT-8 is clearly more computationally efficient than GINO in this metric, without even resorting to our implementation tricks. Of course, the key question remains: is GAOT-8 accurate enough ? We have answered this question in the ablation study of SM Table F.2 where we report the following relative errors for the Elasticity benchmark: GAOT-2 (1.80), GAOT-8 (1.60) and GINO (4.38) and for the Poisson-Gauss benchmark (no change in underlying graphs): GAOT-2 (1.05), GAOT-8 (1.65) and GINO (7.57). Note that, as this was an ablation study, multiscale radii in the MAGNO encoder/decoder were not used here. These numbers clearly show that the larger patch size in GAOT still leads to a model significantly more accurate than GINO, while being more efficient, without resorting to any novelties in implementation. Viewed in this light, our implementation tricks were aimed at making our most expressive model competitive with the fastest models around, while still retaining the superior accuracy of GAOT.
>
> (See the second part of the response below)

---

> ### Author Response · Authors · 2025-08-06
> **Further Response to the Reviewer Part #2 (Points of Novelty)**
>
> [4.] Given the above clarification about novelty of implementation, we would like to take this opportunity to reiterate the key elements of novelty in our paper.
>
> In our opinion, we did not posit the novelty in implementation as *our key original contribution* in the main text but as one of the many novel contributions (l68-69). We still claim that the originality in our model design, concretized in terms of multiscale encoders/decoders and the use of statistical geometric embeddings contributes significantly to the performance of GAOT and have not been used in the literature to the best of our knowledge.
>
> The role of these design elements has been clearly demonstrated with ablation studies in SM Table F.5 (for multiscale encoders/decoders) and Table F.4 (for statistical embeddings).  Following the suggestion of another reviewer, we ablated GAOT further by just running the ViT processor (without the encoder/decoder of GAOT) on the Regular grid datasets from Table E.2 where ViT can be run without any need of encoding/decoding, to obtain the following relative % errors: NS-Gauss (ViT (3.16) vs GAOT (2.29)), NS-PwC (ViT (3.89) vs. GAOT (1.23)), NS-SL (ViT (0.73) vs. GAOT (0.98)), NS-SVS (ViT (0.4) vs. GAOT (0.46)), CE-Gauss (ViT (6.81) vs. GAOT (5.28)), Wave-Layer (ViT (5.48) vs. GAOT (5.40)), Poisson Gauss (ViT (2.47) vs. GAOT (0.79)). From these results, it is clear that in some cases, ViT is comparable to or slightly better than GAOT, indicating that the power of GAOT in these cases stemmed from a good processor. However in many more cases, GAOT is significantly superior to ViT also showing that our novel design of the encoder/decoder contribute significantly to expressivity and together, this combination achieves SOTA performance, even on regular grids, let alone on arbitrary domains.
>
> Summarizing, we believe that the design novelties in GAOT are more important as contributions than its efficient implementation and these underpin its wide-ranging SOTA performance, which we have demonstrated over a wide range of PDE solution operators on different types of domains, including 3D industrial use cases.
>
> We sincerely hope that this (admittedly long) discussion has further clarified the reviewer's concern about the relative importance of contributions of our paper and will play a role in their final assessment of our paper. We thank the reviewer again for providing us with a very detailed feedback and with multiple suggestions that will help us improve the quality and clarity of a CRV, if accepted.

---

> > ### Comment · Reviewer_diAB · 2025-08-06
> >
> > I appreciate that you recognize that GINO could be faster with these tricks. This is not a concern to me, as your approach is particularly more accurate, as you mentioned.
> >
> > I agree that GAOT's contribution are multiple: 1) the encoder/decoder architecture (multi-scale, geometry embeddings) and 2) its efficiency. Still, I think it can also been as an incremental improvement of GINO by proposing:
> > - a multi-scale version of GINO encoded to a coarser regular grid.
> > - the novel geometry embeddings.
> > - instead of a FNO, a ViT that operates on patches
> >
> > I still believe the work is consequent and interesting, and with a correct revision of the paper where it clearly states that the paper provides tricks to speed up GINO and make it more efficient. Then, to make it more accurate, the authors propose multi-scale and geometry embeddings at a small efficiency cost. This would help to better highlight the real contributions.
> >
> > I thanks the authors again for the added details that will be taken in consideration for the final justification section.

---

> > > ### Author Response · Authors · 2025-08-07
> > > **Thanking the Reviewer**
> > >
> > > We sincerely thank the reviewer for engaging in this very extensive back and forth discussion with us. Your detailed feedback and suggestions, particularly about framing our contributions, will be incorporated into a CRV, if accepted. Moreover, we believe that this extensive discussion, by itself, will also be beneficial to the potential readers of this paper, if accepted,  in positioning its contributions in the right context.

---

### Official Review · Reviewer_uofs · 2025-07-02

**Clarity:** 4
**Significance:** 3
**Originality:** 4
**Rating:** 5
**Confidence:** 4

**Summary:**

The paper introduces Geometry-Aware Operator Transformer (GAOT) as a surrogate model that learns solution operators for PDEs defined on arbitrary, non-Cartesian domains. It combines a multiscale attentional graph neural operator (MAGNO) encoder, geometry embeddings, and a ViT-style token processor/decoder to map irregular point-cloud inputs into geometry-aware latent tokens and accurately reconstruct the physical field at any query point. Across 24 benchmarks it shows SOTA results while substantially improving training throughput and inference speed.

**Questions:**

None.

**Ethical Concerns:**

["NO or VERY MINOR ethics concerns only"]

**Limitations:**

None.

**Paper Formatting Concerns:**

No.

**Quality:**

4

**Strengths And Weaknesses:**

Strengths

1. It is a solid work. Addresses the problem learning PDE operators on arbitrary (non-Cartesian) domains while remaining computationally tractable.
2. The experiments are convincing as it introduces 24 different benchmarks and multiple popular SOTA methods as baselines.
3. The pictures of the paper are clear and the paper is easy to read or follow.

Weakness
1. Is it well defined to be neural operators which are resolution-invariant?
2. Caption of figures are too small.

---

> ### Author Rebuttal · Authors · 2025-07-30
>
> We start by thanking the reviewer for their careful reading, appreciation of the merits of our paper and their comments and suggestions for improving it. Below we address their points of concern.
>
> [1.] Regarding the reviewer's question about **neural operators**, our claim is based on the fact that GOAT is endowed with a *neural field* property (l173-175) that allows it to process inputs on arbitrary point clouds and provide outputs on *any point* in the underlying domain. This enables GAOT to possess a key attribute for a neural operator, i.e., the ability to process inputs and outputs at arbitrary underlying grid resolutions. Moreover, works such as Refs. [22,4] argue that this attribute does not suffice for a neural operator and it should possess some (approximate) *resolution invariance* for genuine operator learning. We test this attribute for GAOT in SM E.5. In this context, Fig E.5 clearly demonstrates how GAOT can generalize seamlessly to test resolutions, different from the training resolution. Thus, our labelling of GAOT as a neural operator is justified in this sense.
>
> [2.] We will gladly increase the size of the figures in a camera-ready version (CRV), if accepted. This will be aided by the additional page that the CRV is allowed to have.
>
> We take this opportunity to thank the reviewer again for their constructive comments and suggestions.

---

### Official Review · Reviewer_SKSS · 2025-07-03

**Clarity:** 2
**Significance:** 3
**Originality:** 2
**Rating:** 5
**Confidence:** 3

**Summary:**

The work proposes the geometry-aware operator transformer (GAOT) for predicting PDE solutions as a neural operator. The authors incorporated the multiscale attentional graph neural operator (MAGNO) to capture multiscale information, utilizing statistical information regarding geometry. Also, they used the patching strategies as in ViTs for faster prediction. The experimental evaluation suggests that the proposed method has the best or second-best accuracy among the selected baselines.

**Questions:**

See the weaknesses above.

**Ethical Concerns:**

["NO or VERY MINOR ethics concerns only"]

**Final Justification:**

The authors addressed all the concerns raised by the reviewer. Therefore, I would recommend acceptance.

**Limitations:**

yes

**Quality:**

3

**Strengths And Weaknesses:**

Strengths:
* The method showed good accuracy and computational efficiency compared to other neural operators.

Weaknesses:
* The reason behind the choice of formulations is not clear, making the method look ad hoc. The author needs to specify the concrete problems persisting in existing methods and explain the strategy to overcome them. For instance, more justification would be necessary for the choice of statistical descriptors, as well as the need for multiscale computation.
* The writing is not clear. For instance, the Efficient implementation section is written in a qualitative manner, but almost no quantitative information is provided. The reviewer recommends adding some quantitative evaluation of the algorithm, such as computational complexity.
* The choice of baseline is not complete. Judging from the motivation, there needs to be a comparison against pure-GNN-based PDE solvers such as meshgraphnet [Pfaff+ 2020] and MP-PDE [Brandstetter+ 2022] because these methods are also geometry aware. In addition, the comparison of the speed-accuracy tradeoff against classical solvers (with various spatiotemporal resolutions to control the tradeoff) is necessary.
* The evaluation of the neural field property of GOAT (Table 3) is not clear because there is no other method with the same setting. The reviewer recommends adding baselines with the neural field setting.

Minor points:
* Equation 2 is confusing, since there is no x in PDEs. The author recommends using x in PDEs or rewriting the condition, e.g., writing $\mathrm{in} \ D$ instead of $\forall x \in D$.
* $t$ should be an open set in Equation 2 because we cannot define a proper derivative on the boundary (that's why we need initial and boundary conditions).
* SM is not clear at first glance. Using Appendix would be more understandable.

---

> ### Author Rebuttal · Authors · 2025-07-29
>
> We start by thanking the reviewer for their comments and suggestions. Below, we provide a detailed response to all of the reviewer's points of criticism.
>
> [1.]  Regarding the point about *reasons behind formulations*: we start by reiterating the main motivation for our paper: as clearly stated in the introduction (l41-50 and Fig.1), we observed an **accuracy-efficiency tradeoff** for all existing ML models for PDEs on arbitrary domains. In particular, accurate models such as GNNs (RIGNO) were not efficient and efficient models such as GINO were not as accurate. This motivates the design of a model that can be both efficient and accurate at the same time -- this is main premise of our paper. To do so, we observe that end-to-end GNNs such as RIGNO involve repeated sparse computations and memory accesses (l204-205) and will not necessarily be efficient and scalable on modern GPU hardware. Hence, we would like to minimize sparse computations. To this end, we focus on a paradigm, where the inputs are encoded to a latent grid, where they can be batch processed efficiently with transformers and then decoded to the desired output. Although our processor is a standard ViT, the encoder and decoder are designed to enhance accuracy by adding 2 novel features: *multiscale attentional GNO* and *geometric embeddings* based on statistical descriptors.
>
> The reason behind a *multiscale GNO encoder-decoder* lies in the fact that solutions of PDEs contain a whole range of spatial scales (SM l764). Almost all the benchmarks that we consider in Tab. 1 contain multiscale solutions (see SM Eq D.5 for an explicit example in the Poisson-C-Sines dataset). Hence, adding multiscale features to the encoder-decoder equips the model with the *ability to sense different perceptive fields*, making it a reasonable choice in this regard. This is also justified by the ablation study presented in SM F.5 (Tab F.5), where we show that using a multiscale encoder-decoder is significantly more accurate than a single-scale version.
>
> We have already discussed the reasons behind our novel *geometry embeddings* briefly in l134-140 and more extensively in SM l675-692. Repeating the main points, existing models either provide coordinates of input points or SDFs to encode geometric information. We argue that this does not suffice for problems where domain geometry itself plays an important role in the solution operator (e.g flow past airfoils) and hence additional geometric information needs to be provided to the model as done with the statistical embeddings based on neighborhood information (l140-149 and SM l693-719). The addition of these embeddings is also justified with the ablation study presented in SM F.4 (Tab F.4) where adding statistical embedding significantly increases accuracy over just inputting coordinates and SDFs.  We would be happy to elaborate on the above points further in a camera ready version (CRV) if accepted.
>
> [2.] Regarding the reviewer's concern about *lack of clarity regarding efficient implementation*, we begin by pointing out that our implementation of GAOT is the **key to achieving efficiency**, which is the second axis in our rationale for this paper. Following the reviewer's suggestion, we can readily include theoretical complexity estimates for GAOT and baselines in a CRV. Given E edges, C channels, T Latent tokens, p Patch size and L Layers, compute for GAOT scales as $O((E+L(T/p)^2)C)$, GINO as $O((E+LT\log T)C)$ and GNN-based RIGNO as $O(LEC)$. Clearly, GAOT and GINO will outperform RIGNO when $E>>T$ (which can be ensured by design) as we see from Fig 3 (a). This also suggests GAOT will be competitive wrt GINO for large patch sizes. Yet, we show in Fig. 3(a) that GAOT is **even more efficient** than GINO. To achieve this was the key objective of our efficient implementation and we outline the main steps in l209-216. Following a detailed analysis of the compute cost of each individual component of GINO (pl. see the exact numbers in pt. [4] in the reply to Reviewer diAB below), we observed that for GINO: i) graph building is almost as expensive as encoding/decoding and ii) there is no batch processing due to the heterogeneity in the underlying graphs for PDEs on arbitrary domains, implying that the realized computational cost of GINO is not as low as the theoretical scaling would suggest. To alleviate these issues, as described in l209-216, we were able to i) move graph construction for GAOT outside the model evaluation and ii) implement batch processing in the GAOT transformer processor while sequentially processing inputs for the encoder/decoder. It is precisely these implementation tricks (which can only be described qualitatively and realized in the submitted code) which allow GAOT to be more efficient than GINO (pl. see exact numbers in pt [4] in the reply to Reviewer diAB below) and provide the context for the scaling results in Fig 3a and 3b. We are happy to add theoretical complexity estimates as well as a detailed compute breakdown in a CRV, if accepted and thank the reviewer for suggesting it.
>
> [3.] RIGNO (Ref. [39]) is a state of the art end-to-end GNN that *greatly improves upon MeshGraphNets* (MGN). In fact, MGN can be realized as a special case of RIGNO. As we considered RIGNO as a baseline, we did not include results with MGN. However, following the suggestions of the reviewer, we ran MGN on several of our benchmarks to obtain the following % relative errors: NS-Gauss (76.4 (MGN) vs 2.91 (GAOT)), NS-PwC (33.1 (MGN) vs 1.5 (GAOT)), NS-SL (18.8 (MGN) vs 1.21 (GAOT)), NS-SVS (76.4 (MGN) vs 6.4 (GAOT)), CE-RP (13.9 (MGN) vs 5.97 (GAOT)), Wave-Layer (84.8 (MGN) vs 5.78 (GAOT)), Wave-C-Sines (17.3 (MGN) vs 4.65 (GAOT)), Poisson-Gauss (30.9 (MGN) vs. 0.83 (GAOT)), Elasticity (11.9 (MGN) vs. 1.34 (GAOT)). All the experiments were run with 2M parameter MGN. Given the limited time of the rebuttal, we could only benchmark MGN on 10 of the 15 tasks of Tab. 1. But, the trend is crystal clear. GOAT is an order (sometime two orders) of magnitude more accurate than MGN. Moreover, one can see from Tab. 1 that RIGNO is also much more accurate than MGN. This justifies our choice of benchmarking against a state of the art GNN such as RIGNO rather than a more traditional MGN baseline. Nevertheless, we will include MGN results in a CRV as an additional baseline. MP-PDE has mostly been run on simple 1-D PDEs and was already shown to be significantly inferior to UPT, which have already benchmarked against (Tab. 1).
>
> [4.] The *neural field property* is a desirable feature for any ML surrogate for PDEs as it enables evaluation of the input and output fields at any point in the underlying spatial domain. In particular, it allows the model to be trained and tested at different resolutions, a highly desirable feature in PDE operator learning (Ref [22]). For example, with a neural field model, one can train at a coarser resolution (for training efficiency) but test at a higher resolution (for potentially more accuracy). It is this feature of GAOT that we wanted to demonstrate in Tab. 3 for the industrial scale 3-D Drivaernet++ dataset. However, as described in l296-301, having the neural field property does not imply that the model is more accurate. We can observe from Tab. 3 that the version of GAOT trained with only 10% of the total input points but tested at the full dataset resolution of 500K surface points using the neural field property is nowhere as accurate as the version of GAOT trained and tested at full resolution, although it is still better than some of the baselines. Thus, comparing with a neural field model on this dataset will not lead to more accurate results. Nevertheless, to further explore the consequences of model endowed with the neural field property, we observe that the UPT baseline provides a neural field. As testing UPT on an industrial scale dataset in the limited time of rebuttal is not possible, we considered the 2-D NACA0012 airfoil dataset with 2 different settings: i) training and testing GAOT and UPT at full resolution leading to relative % errors of 6.81 (GOAT) vs. 16.1 (UPT) and ii) training on 1/4th of the resolution and testing at full resolution leading to errors of 9.54 (GOAT) vs. 16.3 (UPT). This experiment reinforces that a) training at full resolution is simply more accurate than training at coarser resolutions b) GOAT is significantly more accurate than UPT in both settings and iii) UPT does not improve much with full resolution training, indicating lack of convergence with increasing resolution. We will gladly add these results together with a more detailed discussion of the neural field property in a CRV.
>
> [5.] We will address the pertinent minor points raised by the reviewer in a CRV, if accepted.
>
> Finally, in addition to the very low overall score, the reviewer has given our article the lowest possible score of 1 for originality and clarity. We take this opportunity to bring to your attention the fact that i) multiscale attentional encoder/decoder ii) statistical geometric embeddings and iii) multiple tricks for efficient implementation as described in pt [2.] of our reply above are completely novel to the best of our knowledge and are acknowledged as such by the other reviewers. These original contributions have enabled us to propose a model that is highly accurate, very efficient and scalable. We have demonstrated its state of the art performance on 24 different datasets, corresponding to a wide spectrum of challenging PDEs, including industrial scale 3-D benchmarks. To the best of our knowledge, no existing ML PDE surrogate model has the scalability, range and performance of GAOT. We hope that this rebuttal has allowed us to clarify our contributions accordingly and to reiterate the novelty, scope and performance of our model. Therefore, we kindly request the reviewer to upgrade their assessment accordingly and we are at your disposal to answer any further questions about our article.

---

> > ### Comment · Reviewer_SKSS · 2025-08-04
> >
> > Thank you for the response.
> > Although my concerns are partially addressed, the comparison to a classical solver, which is crucial for justifying the use of machine learning over classical solvers, remains unaddressed.

---

> ### Author Response · Authors · 2025-08-04
> **Response to the Reviewer**
>
> We start by thanking the reviewer for your response and for stating that your concerns are partially addressed.
>
> Regarding your question about the runtime of classical solvers, we sincerely apologize for not having responded in our original rebuttal due to space limitations. The comparison of runtime of classical solvers and that of ML surrogates is the **very premise of ML for PDEs** and is certainly an important issue. We did not discuss it in detail in our paper as a lot of the datasets that we compare GAOT and the baselines on are not generated by us. Rather, they are publicly available datasets.
>
> To be more specific, for the results in Table 1, Poisson-Gauss, NS-Gauss, NS-PwC, NS-SL, NS-SVS, CE-Gauss, CE-RP, and Wave-Layer datasets are taken from Ref. [20], with the specific random point cloud inputs from Ref. [39], the elasticity dataset is from Ref. [26] and Wave-C-Sines is from Ref. [39]. In Page 72 of [20], the authors of [20] state that runtime of a highly optimized GPU spectral viscosity solver that they use for the Navier-Stokes problems (NS-Gauss, NS-PwC, NS-SL, NS-SVS) at the resolution of $128^2$ is 0.1 sec and that of the FEM solver for Poisson-Gauss is 10 sec.,The authors of [20] also state that CE-Gauss, CE-RP are generated with highly optimized GPU-based finite volume solvers for the compressible Euler equations but do not provide the runtimes. As these solvers are more expensive than spectral solvers and the compressible Euler equations have more variables (4), we expect that the runtime of these solvers for the CE-RP and CE-Gauss datasets are between 0.1 sec and 1 sec. Unfortunately, the runtimes of Wave-Layer, elasticity and Wave-C-Sines are not available in the above references. But given that either finite difference or finite element solvers were used, we would expect runtimes between 1 sec and 10 secs for these 2D problems.
>
> We have generated the data for the NACA0012, NACA2412, RAE2822 and Bluff-body datasets for compressible flow past objects as well as the Poisson-C-Sines dataset. The compressible flow datasets took between 283-419 seconds (ca. 5 to 7 min) per sample to generate as the underlying code was CPU based and highly-adaptive grids with suitable iterative solvers were needed to obtain accurate results, please SM D.2 for details. The FEM simulation for Poisson-C-Sines took 10 secs to simulate.
>
> Finally, the industrial scale 3D Drivaernet++ dataset has been taken from Ref. [12]. From Page 9 of [12], we observe that *on an average, generating each sample of Drivaernet++ took 375 hours* of CPU compute !!. This is not unexpected for this challenging 3-D problem with a RANS model on a dense volumetric grid.
>
> Summarizing, the runtime of classical solvers ranged from 0.1 secs to 7 min in 2D and a massive 375 hours in 3D.
>
> Next, we find that inference runtime of GAOT per sample for the 2D datasets ranged from 8.95 to 10.14 milli-seconds (we will provide the exact numbers for each dataset in a CRV, if accepted). On the other hand, for 3D Drivaernet++ dataset, the GAOT inference runtime was ca. 365 milliseconds per sample.
>
> Thus, for the 2D datasets, the runtime gain for GAOT over a classical solver ranged from 1 to 5 orders of magnitude, depending on the dataset considered. On the other hand for the industrial scale 3D dataset, **the gain was a massive factor of 37M (7 orders of magnitude)**, while still being the SOTA model on this benchmark.
>
> One can argue that these gains come at cost of errors with respect to the numerical solvers. However, these errors are small. For instance, in the Poisson-Gauss, the error with GAOT is less than 1% (with a runtime gain of a factor of 10000) and for Bluff body, the error is ca 2% (with a runtime gain of 29000). For the 3D benchmark, the gain is very significant. We also refer the reviewer to an earlier study on the accuracy-runtime efficiency performed in Ref. [45] (Page 58) for the NS-SL problem where the authors of Ref. [45] found that their model (CNO) was able to provide accuracy comparable to a spectral solver on a $100^2$ grid. We use exactly the same setup as [45] and find that GAOT has even lower errors than CNO on this benchmark (SM Table E.2). Hence, for this NS-SL dataset, we can show that GAOT has comparable errors to the spectral GPU solver, while still being a factor of 10 faster to run.
>
> We are happy to add a discussion on runtime (and accuracy) comparisons of GAOT and classical solvers in the appendix of a CRV, if accepted and thank the reviewer for pointing out this avenue to further highlight the merits of our model, especially on industrial scale 3D simulations.
>
> We thank the reviewer again for your questions and hope that we have addressed your remaining concerns satisfactorily. If so, we request the reviewer to kindly upgrade your assessment accordingly, while remaining at your disposal for further questions.

---

> > ### Comment · Reviewer_SKSS · 2025-08-06
> >
> > Thank you for the detailed response.
> >
> > > One can argue that these gains come at cost of errors with respect to the numerical solvers.
> >
> > Yes, indeed, this is the point because classical solvers have options for speedup by changing resolution or convergence threshold when some errors are accepted. However, I understand that obtaining such an evaluation of tradeoff is sometimes complex. Therefore, if the authors cannot show the comparison for the tradeoff, I would recommend mentioning it in e.g., the Limitation section, if accepted.
> >
> > Nevertheless, reflecting the detailed response from the authors, I will increase the score to the borderline accept.

---

> ### Author Response · Authors · 2025-08-06
> **Thanking the Reviewer.**
>
> We sincerely thank the reviewer for raising your score to *borderline accept* and for your comments and suggestions which can help us improve the quality and clarity of our paper.
>
> We are happy to discuss the cost-accuracy tradeoff of numerical solvers for PDEs vs. ML surrogates in a CRV, if accepted. To this end, we will mention and contextualize the study performed in Ref. [45], where in page 58, the authors of [45] precisely present such a study for the NS-SL benchmark on regular grids and convincingly show that the accuracy of their model (CNO) was comparable to (error of 2%) with accuracy of a spectral viscosity solver at a resolution of $100^2$ (also error of 2%), when compared to a high-resolution spectral solver for this problem (Ground Truth). So for this dataset, [45] has already shown that the error of $100^2$ grid with a spectral-viscosity solver is 2%, with a run-time of 0.1 sec. As it happens, we had run this dataset with GAOT and presented the results in SM Table E.2 to obtain an error of less than 1%, with an inference time of 10 ms. Thus, our ML surrogate is 10 times faster than a state of the art GPU optimized numerical solver, even for the same accuracy. Nevertheless, we cannot perform this analysis of every dataset that we test on as we have not generated the underlying data in most cases. We will acknowledge this fact as a a limitation, not just of our ML model but most ML models for PDEs as this cost-benefit accuracy with respect to classical solvers is rarely discussed in the literature.
>
> We thank the reviewer again for your comments.

---

> ### Author Response · Authors · 2025-08-06
> **Additional Results on Numerical Solvers vs. ML Surrogates.**
>
> Motivated by the reviewer's comments about the cost-accuracy comparison of classical numerical solvers and ML surrogates, we wanted to explore if the cost-accuracy results of Ref. [45] (as extensively discussed above), which were obtained on regular grids, can be replicated for the case of arbitrary point clouds that we have considered as the main use case of GAOT.
>
> To this end, we focussed on one of the NACA airfoils dataset, where the input and output grids are unstructured. Using a highly refined grid (similar to the one shown in SM Figure D.2) to generate the ground truth, we coarsened this mesh by 2 levels of refinement. Preliminary results indicate that the runtime came down to 1 min (from 7 min for the full mesh) and the resulting relative error was approximately 5%. As reported in Table. 1, the GOAT relative error is approximately 6.5%. Thus, GAOT has comparable errors to a lower resolution numerical solver. Yet, its runtime of 10 ms is **6K times faster**. This experiment clearly justifies the use of an accurate and efficient ML surrogate like GOAT, especially on problems on unstructured grids and 3D, even more than regular grids as considered in [45].
>
> We will include these results in a CRV, if accepted and thank the reviewer for pressing us to provide such a quantitative comparison. We hope that these results have further clarified the reviewer's remaining concern and play a role in their final assessment of our paper.

---

> > ### Comment · Reviewer_SKSS · 2025-08-07
> >
> > Thank you very much. Now I have the ground to increase my score to accept.

---

> > > ### Author Response · Authors · 2025-08-07
> > > **Thanking the Reviewer**
> > >
> > > We sincerely thank the reviewer for reading our response and for raising our score to *Accept*. Your constructive suggestions, particularly on, the cost-accuracy trade off of classical numerical solvers vs. ML surrogates have been very useful and adding them to a CRV, if accepted, will greatly enhance the quality of our paper.

---

### Decision · Program_Chairs · 2025-09-17

**Decision:**

Accept (poster)

**Comment:**

This paper introduces GAOT (geometry-aware operator transformer) to combine an encoder that can handle arbitrary geometry in the gridded data and ViT-based layers for the latent tensors (with a user-chosen topology for the latent grid). Experiemental-wise, this is a very good paper. Please acknowledge the references using similar approaches that the reviewers mentioned such as CViT, Poseidon, and Transolver++. I also suggest adding some earlier works on Transformer neural operators that exploit lower latent dimensions, such as Galerkin or OFormer. In the meantime, multilevel feature aggregation (formula (4)) has been proposed in the Multigrid NO paper (ICLR 2024) using multigrid prolongation and restriction.

BTW: personally I am not a big fan of using the radar chart that is commonly seen in LLM papers to illustrate the performance of different neural operators.